# Secondary organic aerosol formation from OH-initiated oxidation of *m*-xylene: effects of relative humidity on yield and chemical composition

Qun Zhang[1,2], Yongfu Xu[1,2,*], Long Jia[1,2]

[1]State Key Laboratory of Atmospheric Boundary Layer Physics and Atmospheric Chemistry, Institute of Atmospheric Physics, Chinese Academy of Sciences, Beijing 100029, China,
[2]Department of Atmospheric Chemistry and Environmental Sciences, College of Earth Sciences, University of Chinese Academy of Sciences, Beijing 100049, China

*Correspondence to*: Yongfu Xu (xyf@mail.iap.ac.cn)

**Abstract.** The effect of relative humidity (RH) on the secondary organic aerosol (SOA) formation from the photooxidation of *m*-xylene initiated by OH radicals in the absence of seed particles was investigated in a Teflon reactor. The SOA yields were determined based on the particle mass concentrations measured with a scanning mobility particle sizer (SMPS) and reacted *m*-xylene concentrations measured with a gas chromatograph-mass spectrometer (GC-MS). The SOA components were analysed using Fourier transform infrared spectrometer (FTIR) and ultrahigh performance liquid chromatograph-electrospray ionization-high-resolution mass spectrometer (UPLC-ESI-HRMS). A significant decrease was observed in SOA mass concentration and yield variation with the increasing RH conditions. The SOA yield is 14.6% and 0.8% at low RH (14%) and high RH (79%), respectively, with the difference being over an order of magnitude. The chemical mechanism for explaining the RH effects on SOA formation from *m*-xylene-OH system is proposed based on the analysis of both FTIR and HRMS measurements, and the Master Chemical Mechanism (MCM) prediction is used as the assistant. The FTIR analysis shows that the proportion of oligomers with C-O-C groups from carbonyl compounds in SOA at high RH is higher than that at low RH, but further information cannot be provided by the FTIR results to well explain the negative RH effect on SOA formation. In the HRMS spectra, it is found that $C_2H_2O$ is one of the most frequent mass difference at low and high RHs, that the compounds with lower carbon number in the formula at low RH account for a larger proportion than those at high RH, and that the compounds at high RH have higher O:C ratios than those at low RH. The HRMS results suggest that the RH may suppress the oligomerization where water is involved as a by-product and may influence the further particle-phase reaction of high oxygenated organic molecules (HOMs) formed in the gas phase. In addition, the negative RH effect on SOA formation is enlarged based on the gas to particle partitioning rule and the different wall losses of SOA precursors.

## 1 Introduction

Secondary organic aerosol (SOA) is a significant component of atmospheric fine particulate matter in the troposphere (Hallquist et al., 2009; Spracklen et al., 2011; Huang et al., 2014), leading to serious concerns as it has a significant influence

on the air quality, oxidative capacity of the troposphere, global climate change and human health (Jacobson et al., 2000; Hansen and Sato, 2001; Kanakidou et al., 2005; Zhang et al., 2014). In a previous study from a global model simulation, it has been found that SOA represents a large fraction, approximately 80% of the total organic aerosol sources (Spracklen et al., 2011).

The formation of SOA in the atmosphere is principally via the oxidation of volatile organic compounds (VOCs) by common atmospheric oxidants such as $O_3$, OH and $NO_3$ radicals (Seinfeld and Pandis, 2016). Aromatic compounds mainly from anthropogenic sources, including solvent usage, oil-fired vehicles and industrial emissions, contribute 20-30% to the total VOCs in urban atmosphere, which play a significant role in the formation of ozone and SOA (Forstner et al., 1997; Odum et al., 1997; Calvert et al., 2002; Bloss et al., 2005; Offenberg et al., 2007; Ding et al., 2012; Zhao et al., 2017). Amongst aromatics, *m*-xylene is significant, of which mean concentration together with *p*-xylene in daytime was determined up to 140.8 $\mu g\ m^{-3}$ in atmosphere of urban areas in developing countries (Khoder, 2007).

The oxidation of aromatics in the troposphere is mainly initiated through OH radicals, which is affected by many chemical and physical factors. The concentrations of oxidant species, VOCs and $NO_x$ concentrations, as well as the ratio of VOCs to $NO_x$ (Ge et al., 2017b) determine the main chemical mechanism. Light intensity (Warren et al., 2008), temperature (Qi et al., 2010) and relative humidity (RH) are the most significant physical parameters that affect the chemical process. RH governs the water concentration in the gas phase and the liquid water content (LWC) in the particle phase. Water plays a significant role that can serve as reactant, product and solvent to directly participate in chemistry (Finlayson-Pitts and Pitts Jr., 2000) and indirectly affect the reaction environment such as acidity of particles (Jang et al., 2002). Acid-catalysis of heterogeneous reactions of atmospheric organic carbonyl species in particle phase can lead to a large increase of SOA mass, while this process can be suppressed by the lower acidity at high RH (Czoschke et al., 2003). In addition, RH can change the viscosity of SOA and further affect the chemical processes of SOA formation (Kidd et al., 2014; Liu et al., 2017).

Investigations of RH effects on aromatics SOA have been conducted in many previous works. In the presence of $NO_x$, it was observed that RH significantly enhanced the yield of SOA from benzene, toluene, ethylbenzene and xylenes photooxidation, which was explained by a higher formation of HONO, particle water, aqueous radical reactions and the hydration from glyoxal (Healy et al., 2009; Kamens et al., 2011; Zhou et al., 2011; Jia and Xu, 2014, 2018; Wang et al., 2016). Meanwhile, under low $NO_x$ condition, wherein no $NO_x$ were introduced artificially and photolysis of $H_2O_2$ was as an OH radical source, it has been observed that deliquesced seed contributed to the enhancement of SOA yield from toluene (Faust et al., 2017; Liu et al., 2018). However, under low $NO_x$ level, it has been found that, in the study on toluene SOA formation, moderate RH level (48%) leads to a lower SOA yield than low RH level (17-18%) (Cao and Jang, 2010). In a most recent study on SOA formation of toluene (Hinks et al., 2018), high RH led to a much lower SOA yield than low RH under low $NO_x$ level, which is attributed to condensation reactions that remove water, leading to the less oligomerization at high RH. In a study on chemical oxidative potential of SOA (Tuet et al., 2017) under low $NO_x$ conditions, it was observed that the mass concentration of SOA from *m*-xylene irradiation under the dry condition was much larger than that under the humid condition, whereas the study did not focus on the mechanism of the RH effect on *m*-xylene SOA formation. These demonstrate that the RH effects on aromatics

SOA yields, especially *m*-xylene, have not been fully understood and the RH effects are controversial under various $NO_x$ levels and seed particle conditions.

Chemical components of SOA are important, on which climate- and health-relevant properties of particles are dependent. Chemical composition of SOA from aromatics-$NO_x$ photooxidation has been investigated by GC-MS analysis (Forstner et al.,

1997). Nevertheless, this study was only performed at a limited RH range of 15-25%. GC-MS in this study may not be the optimal technique for analysis of SOA components as high temperature at GC injection ports can easily decompose some low-volatile substances in SOA. FTIR was also used to study the chemical composition of SOA from aromatics-$NO_x$ photooxidation under different RH conditions, in which the information of functional groups in SOA was provided (Jia and Xu, 2014, 2018). In these studies, O-H, C=O, C-O, and C-OH were found to be the main functional groups, intensities of

which largely increased with increasing RH. Compounds in SOA with the O-H group mainly contributed to the increasement of SOA, such as polyalcohols formed from aqueous reactions. The recent study on SOA components from toluene-OH system under both dry and humid conditions were analysed via HRMS (Hinks et al., 2018). Although some chemical composition in SOA has been identified, the analysis and the mechanism of RH effects still need to be further studied. RH effects on SOA formation from *m*-xylene under low $NO_x$ condition have not been studied well. In the present study, we present the results

from the experiments about the SOA formation from the OH-initiated oxidation of *m*-xylene in the absence of seed particles in a Teflon bag. The SOA yields at different RHs and the chemical components under both low and high RH conditions will be reported. The underlying mechanism of SOA formation for these different conditions will be also discussed.

## 2 Experimental materials and methods

### 2.1 Equipment and reagents

Experiments on *m*-xylene photooxidation were performed in a 1 m$^3$ air-tight Teflon FEP film reactor (DuPont 500A, USA), which is similar to our previous works (Jia and Xu, 2014, 2016, 2018; Ge et al., 2016, 2017a, b, c). A light source was provided by 96 lamps (F40BLB, GE; UVA-340, Q-Lab, USA) surrounding the Teflon bag to simulate the UV band of solar spectrum in the troposphere. The $NO_2$ photolysis rate was determined to be 0.23 min$^{-1}$, which was used to reflect the light intensity in the reactor. To remove the electric charge on the surface of the FEP reactor, two ionizing air blowers were equipped outside

the Teflon bag and were used throughout each experiment (McMurry and Rader, 2007).

The background gas was zero air, which was generated from Zero Air Supply and CO Reactor (Model 111 and 1150, Thermo Scientific, USA) and further purified by hydrocarbon traps (BHT-4, Agilent, USA). The humid zero air was obtained by bubbling dry zero air through ultrapure water (Milli Q, 18MU, Millipore Ltd., USA). To obtain the different desired RH in the reactor, the different ratio dry and humid zero air was mixed. The RH and temperature in the reactor were measured by a

hygrometer (Model 645, Testo AG, Germany).

Throughout each experiment, the background $NO_x$ concentration in the reactor was lower than 1 ppb and OH radicals were provided from $H_2O_2$ photolysis. Hydrogen peroxide was introduced into the reactor along with the zero air flow over a period

of 30 min via an injection of $H_2O_2$ solution (30 wt %) into a three-way tube using a syringe to the desired concentration of 20 ppm. Though the $H_2O_2$ level was not measured, it was estimated through the measured volume of $H_2O_2$ solution evaporated. *m*-Xylene (99%, Alfa Aesar) was introduced to the reactor subsequently using the same approach. No seed particles were introduced artificially. All reactants were introduced initially and then the lights were turned on and the reaction starts. The

5 experiments were conducted for 4 h. Thus, the "end" of the experiment in this study refers to the experiment at 4 h of reaction time.

## 2.2 Monitoring and analysis

The concentration of *m*-xylene in the reactor was measured with a gas chromatograph-mass spectrometer (GC-MS, Model 7890A GC and Model 5975C mass selective detector, Agilent, USA), which was equipped with a thermal desorber (Master

TD, Dani, Italy). The size distribution and concentrations of particles were monitored with a scanning mobility particle sizer (SMPS, Model 3936, TSI, USA). In our enclosed Teflon bag, our sampling instruments consumed 10% of reactor volume during the photooxidation and we did not use make up air to dilute the bag. The particle wall loss constant has been determined to be $3.0 \times 10^{-5}$ s$^{-1}$ and $6.0 \times 10^{-5}$ s$^{-1}$ at low RH and moist conditions, respectively. Though the particle wall loss constant is size dependent, it is not a strong function of particle size for the relatively narrow size distributions in smog chamber

experiments (Park et al., 2001). Here we approximate the wall loss as size independent. In experiments under moist conditions, particles measured by SMPS consisted of liquid water content (LWC) and SOA. In low RH experiments, as SOA hardly absorbs aerosol water, LWC can be negligible. Thus, the SOA mass can be directly measured by SMPS in low RH experiments. To obtain the SOA mass in high RH experiments, LWC should be excluded from total particle mass. The method for the measurement of LWC has been already described in the previous study (Jia and Xu, 2018), so here a brief introduction is only

provided. During each high RH experiment, the SMPS measured the humid particles. After 4 h from the start of oxidation reaction in each high RH experiment, the SMPS was modified to the dry mode. In the dry mode, a Nafion dryer (Perma Pure MD-700-12F-3) was added to the sampling flow and a Nafion dryer (Perma Pure PD-200T-24MPS) was added to the sheath flow. After the modification of SMPS, the humid air in SMPS was quickly replaced by dry air through venting the sheath air at 5 L min$^{-1}$, so that the RH in the sheath air can decrease to 7%. Then, SMPS at this dry mode measured dry particle

concentrations as the RH in the sample air decreased to 10 % at this time. The LWC was determined by the difference of the particle mass concentrations before and after the SMPS modification to the dry mode.

The chemical composition of SOA originated from *m*-xylene-OH irradiation was investigated using Fourier transform infrared spectrometer (FTIR), which can provide the information of functional groups. The particles were collected on a ZnSe disk using a Dekati low-pressure impactor (DLPI, Dekati Ltd., Finland) at the end of each experiment (Ge et al., 2016; Jia and Xu,

2016). The duration of DLPI for FTIR was 15 min, and this sampling was taken just after 4 h of each experiment. Then, the ZnSe disk was directly put in a FTIR (Nicolet iS10, Thermos Fisher, USA) for the measurement of functional groups of the chemical composition in SOA samples.

To obtain the detailed information of chemical composition, SOA particles were sampled using the Particle into Liquid Sampler (PILS, model 4001, BMI, USA). The PILS samples water-soluble species in particles. As the low-$NO_x$ $m$-xylene SOA composition is almost all water-soluble species, it is reasonable and reliable to use PILS to sample SOA for analysis of chemical composition. The flow rate of sample gas was around 11 L min$^{-1}$, and the output flow rate of liquid sample was 0.05 mL min$^{-1}$. Two denuders were used to remove the VOCs and acids in the sample gas. SOA liquid samples collected by PILS were finally transferred into vials for subsequent analysis of mass spectrometry. The duration of PILS was 5 min, and this sampling was taken just after 4 h of each experiment. Operatively, the blank measurements were obtained by replacing the sample gas with zero air collected in vials. It is well known that the PILS samples water-soluble species in the SOA with high efficiency. In addition, it is reported that the PILS can also samples slightly water-soluble organic compounds with average O:C ratios higher than 0.26 instead of the total SOA composition and the collection efficiency could exceed 0.6 (Zhang et al., 2016). Thus, the PILS can sample the overwhelming majority of the SOA system in our study, though PILS cannot sample water-insoluble species in the SOA.

The accurate mass of organic compounds in SOA was measured by the ultrahigh performance liquid chromatograph (UPLC, Ultimate 3000, Thermo Scientific, USA)-heated-electrospray ionization-high-resolution orbitrap mass spectrometer (HESI-HRMS, Q Exactive, Thermo Scientific, USA). Methanol (Optima™ LC/MS Grade, Fisher Chemical, USA) was used as the eluent in UPLC system. The elution flow rate was 0.2 mL min$^{-1}$, and the overall run time was 5 minutes. The injection volume was 20 µL. In this study, the UPLC was only used as the injection system of HRMS. The acquired mass spectrum of SOA was in the range of 80-1000 Da. The HESI source was conducted in both positive and negative ion modes using the optimum method for characterization of organic compounds. We used the Thermo Scientific Xcalibur software (Thermo Fisher Scientific Inc., USA) to analyse the data from HRMS. To calculate the elemental composition of compounds, the accurate mass measurements were used. The reaction pathways and products of $m$-xylene-OH photooxidation in Master Chemical Mechanism (MCM v3.3.1, the website at http://mcm.leeds.ac.uk/MCM; last accessed October 16, 2017) was used for analysis of the products measured by HRMS (Jenkin et al., 2003; Jia and Xu, 2014).

## 3. Results and discussion

### 3.1 RH effects on SOA yields

Eleven experiments were conducted. Experimental conditions and results at 4 h of experiments in $m$-xylene-$H_2O_2$ photooxidation system are summarized in Table 1. Exps. 1-4 were conducted in dry zero air, which are defined as the low RH experiments. Exps. 8-11 were conducted in humid zero air, which are defined as the high RH experiments. Exps. 5-7 were conducted in the mixed air of dry and humid zero air, which are defined as the intermediate RH experiments. The initial concentrations of $m$-xylene and the consumed $m$-xylene proportion (typically ~40%) were approximate under different RH conditions. Under intermediate and high RH conditions, LWC accounts for a certain proportion of particles (Jia and Xu, 2018). To obtain the time evolution of SOA concentrations, the LWC has to be subtracted during the whole photooxidation period.

Since LWC was only measured at the end of the reaction, the volume growth factor (VGF) was used to estimate the contribution of LWC in particles, which was defined as the ratio of the humid particle volume to the dry particle volume (Engelhart et al., 2011). It was assumed that the VGF did not change during the whole photooxidation period. The removal of aerosol water during the LWC measurement may cause the dissolved species that are probably volatile/ semi-volatile compounds to evaporate back into the gas phase (El-Sayed et al., 2015; El-Sayed et al., 2016). Glyoxal is a typical semi-volatile compound with high Henry's law constant, which is involved in SOA formation in $m$-xylene-OH system of our study. The Henry's law constant of glyoxal in pure water is as high as $4.19 \times 10^5$ M atm$^{-1}$ at 298 K (Ip et al., 2009). Only one in ten thousand of glyoxal can dissolve in the LWC whose concentration was obtained in our study. Thus, SOA concentrations for intermediate and high RH conditions were slightly underestimated, but the underestimation is extremely low and can be neglected. To obtain the mass concentrations of SOA, an SOA density of 1.4 g cm$^{-3}$ was used (Song et al., 2007).

In Fig. 1, the wall-loss-corrected SOA mass concentrations are plotted as a function of photooxidation reaction time for $m$-xylene-OH systems at low (Exps.3-4) and high (Exps.10-11) RHs. It can be clearly observed that there is a large difference in the maximum mass concentration between low and high RHs. In Table 1, the maximum mass concentrations are 95.5-150.3 µg m$^{-3}$ at low RHs, whereas they are 7.5-27.9 µg m$^{-3}$ at high RHs, with the largest difference being over ten times. The RH effect was reproducible when the initial $m$-xylene concentration was slightly changed under similar conditions.

We used the definition of the ratio of the SOA mass to the consumed $m$-xylene mass to calculate the SOA yield at the end of each experiment. In Table 1, the SOA yields at low RH are 14.0-16.5%, while those at high RH are only around 0.8-3.2%. SOA yields at low RH are nearly seven times larger than those at high RH. Though temperatures at high RH are slightly higher than those at low RH as shown in Table 1, which can lead to a higher SOA yield, the difference of temperatures between low and high RH conditions is lower than two degree, which cannot lead to a significantly different SOA yield to affect the result (Qi et al., 2010).

Seed aerosols were not artificially introduced throughout all the experiments, which could lead to the underestimation of SOA, as SOA-forming vapours partly condense to the reactor walls instead of particles (Matsunaga and Ziemann, 2010; Zhang et al., 2014). The extent to which vapor wall deposition affects SOA mass yields depends on the specific parent hydrocarbon system (Zhang et al., 2014; Zhang et al., 2015; Nah et al., 2016; Nah et al., 2017). Zhang et al (2014) have estimated two $m$-xylene systems under low NO$_x$ conditions and concluded that SOA mass yields were underestimated by factors of 1.8 (Ng et al., 2007) and 1.6 (Loza et al., 2012) under low RH conditions. In addition, the excess OH radicals in our experimental system lead to a less underestimation of SOA formation as the losses of SOA-forming vapours can be mitigated via the use of excess oxidant concentrations (Nah et al., 2016). Thus, the underestimation of SOA formation can be limited. In fact, the wall loss of $m$-xylene was not taken into consideration of calculation of mass yields, which generally overestimates the mass yields.

The wall loss of organic compounds that is sensitive to humidity can affect the RH effect on SOA yields, as the reduction of SOA yields at the high humidity can be due to the loss to the wet reactor wall. There are thousands of SOA precursors from $m$-xylene-H$_2$O$_2$ photooxidation. However, as far as we know, there is no previous study that investigates the RH effect on the loss of these organic compounds to the wet reactor wall. Thus, we select four organic compounds that have the relevant

experimental data to estimate the extent of how much the wall loss of chemical species affects the SOA formation at different RHs.

The first two compounds are glyoxal and acetone. Although they cannot directly partition in the particle phase, they can form SOA. Glyoxal can easily dissolve in the aqueous phase due to the large Henry's law constant of $4.19 \times 10^5$ M atm$^{-1}$ at 298 K (Ip et al., 2009), very sensitive to humidity. Loza et al. (2010) found that the wall loss of glyoxal was minimal at 5% RH, with $k_W = 9.6 \times 10^{-7}$ s$^{-1}$, whereas $k_W$ was $4.7 \times 10^{-5}$ s$^{-1}$ at 61% RH. Obviously, there is a large difference in wall loss between low and high RHs. We assume that $k_W$ linearly increases with RH, and the $k_W$ value is estimated to be $6.1 \times 10^{-5}$ s$^{-1}$ at 80% and $7.4 \times 10^{-6}$ at 13% RH, with the difference being 8.2 times. According to the wall loss of glyoxal, glyoxal only decreased by 10% at the end of our experiment at low RH, while glyoxal decreased by 59% at high RH. This means that SOA yield would be underestimated by 59% at high RH and by 10% at low RH if glyoxal lost to the wall was completely transformed to SOA. If this wall effect of SOA precursors was taken into consideration, the SOA yields at high (Exp. 8) and low (Exp. 2) RHs would be 6.1% and 15.5%, respectively, still with a nearly three times difference. Acetone can hardly dissolve in the aqueous phase due to the small Henry's law constant of 29 M atm$^{-1}$ (Poulain et al., 2010), which is 4 orders of magnitude less than that of glyoxal. Ge et al. (2017) obtained that the wall loss of acetone was $5.0 \times 10^{-6}$ s$^{-1}$ at 87% RH and $3.3 \times 10^{-6}$ s$^{-1}$ at 5% RH. Acetone only decreased by 5% at the end of our experiment at low RH, while glyoxal decreased by 7% at high RH. Obviously, the relationship of SOA yields at high (Exp. 8) and low (Exp. 2) RHs would not change largely when this wall effect of SOA precursors was taken into consideration. In addition, the difference of wall loss between glyoxal and acetone at low RH is about 2 times, while it becomes about 12 times at high RH. It can be considered that the wall loss among different species at low RH is less affected by the Henry's law constant, but it is greatly affected at high RH. Thus, the loss of organic vapor with the Henry's law constant as high as glyoxal's to the wet wall cannot completely interpret the large difference of SOA formation at low and high RH in our study.

In a recent study, the signal decay of two compounds ($C_5H_8O_2$ and $C_5H_9O_4N$) generated from isoprene oxidation at RH = 5%, 50% and >90% has been presented (Huang et al., 2018), which is selected as the last two compounds. $C_5H_8O_2$ decreased by 10% after 8.3 h at 5% and 50% RH, while $C_5H_9O_4N$ decreased by 20% and 40% at 5% and 50% RH after 8.3 h, respectively. In our study, the SOA yield at the end of our experiment decrease by 71% at intermediate RH (Exp. 6) relative to that at low RH (Exp. 2). According to the wall loss of $C_5H_8O_2$, the SOA yield would not decrease at intermediate RH, while taking $C_5H_9O_4N$, as it only decreased by 10% at low RH and decreased by 22% at intermediate RH, the SOA yield at intermediate RH (Exp. 6) would decrease by 13% relative to that at low RH (Exp. 2). Obviously, the decay characteristic of both two compounds cannot explain the more than three times difference of SOA formation at 14% and 51% RH in our experiments. If we take an extreme case of >90% RH to estimate the impact of the semi-volatile organic compound (SVOC) wall losses on SOA formation in our experiments, the results are indeed different. $C_5H_8O_2$ and $C_5H_9O_4N$ decreased by 90% and 70% after 2 h ($k_W = 3.2 \times 10^{-4}$ s$^{-1}$ and $1.7 \times 10^{-4}$ s$^{-1}$), respectively and subsequently remained steady, indicating the saturation of wet wall to absorb the organic vapour under humid conditions of >90% RH. Taking the decay characteristics of $C_5H_8O_2$, it is estimated that the SOA yield at the end of 2 h would decrease by 90% and would not further decrease after 2 h. However, our experimental

results show that the yield at the end of 2 h is 1.4%, decreased by 87% relative to that at low RH, and then further decreased by 82% at the end of 4 h. It seems that for VOCs that generate intermediate SVOCs with high RH effect of wall losses the RH effect of their SOA yields can be explained by SVOC wall losses during the first period. However, these SVOCs generally have saturation characteristics, which cannot explain our observed RH effect of SOA formation. In fact, there were many different SOA precursors from the $m$-xylene oxidation system that probably have much smaller Henry's law constant relative to that of glyoxal, $4.19 \times 10^5$ M atm$^{-1}$. Thus, it is considered that there can exist other mechanisms to explain the negative RH effect on SOA formation from $m$-xylene photooxidation.

For comparison and discussion of the results of SOA formation with other previous studies, Fig. 2 was plotted to show the SOA yields as a function of RH for the two aromatic compounds (toluene and $m$-xylene) oxidation under low NO$_x$ conditions with the photolysis of H$_2$O$_2$ as the OH source. In Fig. 2, the hollow circles represent that no seed particles were introduced and the circles with a cross represent that seed particles were introduced, and the size of markers indicates the magnitude of amount of reacted VOC. In the most recent study on toluene SOA formation conducted without seed particles (Hinks et al., 2018), the SOA yield at low NO$_x$ level was 15% under dry conditions (< 2% RH) and 1.9% under humid conditions (89% RH), with the ratio of two yields between dry and humid conditions being over 7.5. The toluene SOA produced under high RH conditions were significantly suppressed, in which the tendency of RH effects on SOA yield was very similar with our study, though the difference of SOA yield in the range of low and high RH conditions in Hinks et al (2018) was slightly smaller than that in this study. The small difference of RH effects between Hinks et al. and our study is likely associated with the difference in experimental conditions, including RHs, initial and reacted VOCs and H$_2$O$_2$ concentrations, in addition to different species. This comparison demonstrates that different species of toluene and $m$-xylene of aromatics pose very similar RH effects under low-NO$_x$ conditions. Hinks et al. attributed the suppression of SOA yields by elevated RH to the lower level of oligomers generated by condensation reactions and the reduced mass loading at high RH.

In a study on an SOA model for toluene oxidation, the negative RH effect on SOA formation was also found in the presence of seed particles (Cao and Jang, 2010). In their study, the SOA yield at low NO$_x$ level was 28-30% under low RH conditions (17-18% RH) and 20-25% under moderate RH conditions (48% RH) (Cao and Jang, 2010), but they did not focus on the RH effect to give an explanation. Furthermore, their RH only changed from 17% to 48%, the reacted parent VOC was smaller and the seed particles were present, so the RH effect on SOA yields was not as significant as those in Hinks et al and our study. Ng et al. have investigated the yields of SOA formed from $m$-xylene-OH system at low RH (4-6%) under low NO$_x$ conditions (Ng et al., 2007). They obtained that the SOA yields were in the range of 35.2-40.4% in the presence of seed particles. The SOA yields were larger than those of our study, as they conducted the experiments under different irradiation time and with inorganic seed particles. These seed particles can provide not only surface for chemical reactions, but also acidic and aqueous environments that can promote the SOA formation (Jang et al., 2002; Liu et al., 2018; Faust et al., 2017). The reacted concentration of parent VOC was close between Cao and Jang and Ng et al. though the species were different. The results from these two studies can be considered together, since their experiments all had seed particles. As shown in Fig. 2, the negative RH effect on SOA yields can be found. In addition to these three previous studies shown in Fig. 2, a study on chemical

oxidative potential of SOA (Tuet et al., 2017) found that the concentration of SOA from *m*-xylene irradiation at low NO$_x$ level under dry condition was much larger than that under humid condition (89.3 μg m$^{-3}$ at < 5% RH and 13.9 μg m$^{-3}$ at 45% RH), but they did not calculate the *m*-xylene SOA yields or give an explanation for the RH effect.

## 3.2 RH effects on functional groups of SOA

Figure 3 shows the FTIR spectra of particles from the photooxidation of *m*-xylene-OH experiments under both low (Exp. 2) and high (Exp. 8) RH conditions. The DLPI sample flow rate was 10 L min$^{-1}$, and the sampling duration was 15 min. We used same sampling flow rate and duration for both RH conditions. DLPI has 13 stages, and it can collect particles in the size range of 30 nm - 10 mm. When we sampled using DLPI, the four plates for stages 4-7 were removed, so that particles in the range of 108-650 nm were collected on the third plate. As shown in Fig. S2 in the supplementary information, the particles in the

range of 108-650 nm can represent the total SOA from *m*-xylene oxidation in this study. The mean collection efficiency of the DLPI was 83% for stages 4-7 (Durand et al., 2014). Thus, the SOA mass collected on the ZnSe window was 10.3 and 3.0 μg at low RH (Exp. 2) and high RH (Exp. 8), based on the SMPS measurement and the DLPI collection efficiency. As shown in Fig. 2, the SOA from *m*-xylene-OH experiments can be obviously observed under both RH conditions. The intensities of all functional groups from the low RH experiment are much higher than those from the high RH experiment, which is consistent

with the reduced SOA yields under elevated RH conditions.

The assignment and the intensity of the FTIR absorption frequencies at low (Exp. 2) and high (Exp. 8) RHs is summarized in Table 2. The broad absorption at 3600-2400 cm$^{-1}$ is O-H stretching vibration in phenol, hydroxyl and carboxyl groups (Stevenson and Goh, 1971; Santos and Duarte, 1998; Duarte et al., 2005). The band at 3000 cm$^{-1}$ is C-H stretching vibration (Stevenson and Goh, 1971; Santos and Duarte, 1998; Duarte et al., 2005). The sharp absorption at 1720 cm$^{-1}$ is the C=O

stretching vibration in carboxylic acids, formate esters, aldehydes and ketones (Stevenson and Goh, 1971; Santos and Duarte, 1998; Duarte et al., 2005). The absorptions at 1605 cm$^{-1}$ match C-C stretching of aromatic rings and the C=O stretching of conjugated carbonyl groups. The absorptions at 1415 cm$^{-1}$ match the deformation of CO-H, phenolic O-H and C-O (Coury and Dillner, 2008; Ofner et al., 2011). The absorptions at 1180 cm$^{-1}$ match the C-O-C stretching of polymers, C-O and OH of COOH groups (Jang and Kamens, 2001; Jang et al., 2002; Duarte et al., 2005). The absorptions at 1080 cm$^{-1}$ match the C-C-

OH stretching of alcohols (Jang and Kamens, 2001; Jang et al., 2002).

The absorption intensity at ~3200 cm$^{-1}$ that is identified as the hydroxyl group is used to be a representative for reflection of the SOA formation. As well, Table 2 gives the ratio of intensities at high RH (Exp. 8) to those at low RH (Exp. 2) to compare the difference of relative intensities of functional groups. The intensities of functional groups are obviously suppressed at high RH, but the extents of the suppression for different functional groups are basically divided into two types. The ratios of O-H,

C-H, C=O and C-C-OH groups are 0.29 to 0.34, which is close to the ratio of SOA mass at high RH to that at low RH collected on the ZnSe disk, whereas the ratios of CO-H, C-O-C, C-O-H in COOH are above 0.48. The relative intensity of the C-O-C group is significantly higher than the C=O group, which can be explained by more oligomerization with the formation of C-O-C than other reactions at high RH. Nevertheless, the FTIR results cannot provide further information to well explain the

differences of SOA yields between low and high RH, which will be further discussed in terms of mass spectra of SOA in the next section.

### 3.3 RH effects on mass spectra of SOA

The blank-corrected mass spectra of SOA sample formed from *m*-xylene-OH photooxidation under low and high RH conditions in both positive and negative ion modes are presented in Fig. 4, which is plotted as a function of the mass-to-charge ratio. It should be noted that the Y-axis scales for low and high RH are largely different, $10^6$ at low RH and $10^5$ at high RH. As shown in Fig. 4, a visible decrease in the overall peak abundance for both positive and negative ion modes can be obviously observed as the RH elevates, which is consistent with the result that the SOA mass concentration is lower at high RH. In addition, it is obvious that the number of peaks is less under the high RH condition. As shown in Fig. 4, where the m/z values of SOA samples are close for both low and high conditions, the absolute and relative peak abundance is much different, indicating that RH significantly affects the concentration of SOA components.

For rough quantification of the RH effect, the blank-corrected mass peaks of SOA samples were selected from whose abundance is larger than $10^5$ under the low RH condition and corresponding mass spectra under the high RH condition, and then assigned with the number of carbon atoms. The peak abundance with the same number of carbon atoms (nC) is summed, which is presented in Figure 5. It should be noted that the Y-axis scales at low and high RHs are largely different, with a label step of $4.0 \times 10^6$ at low RH and $4.0 \times 10^5$ at high RH in the positive ion mode, $5.0 \times 10^6$ at low RH and $1.0 \times 10^5$ at high RH in the negative ion mode. The compounds with nC > 8, larger number of carbon atoms than *m*-xylene, are proposed to be oligomers that account for a large mass fraction of SOA due to their large molecular weights and lower volatilities, though their peak abundance is lower. As a result, the processes for formation of such compounds play an important role in the formation of SOA. It can be obviously observed that the peak abundance is much lower at high RH in the negative ion mode than that in the positive mode, indicating that the decrease of the compounds obtained in the negative ion mode account for a larger decrease at high RH.

### 3.4 Proposed mechanism of RH effects on SOA formation

The large difference of SOA yields and composition between low and high RHs suggests that water is directly involved in the chemical mechanism and further affects the SOA growth. In the particle-phase accretion equilibrium reactions, where water is involved as a by-product, the elevated RH alters the equilibrium of reaction by moving toward reducing the fraction of oligomers with low volatility and increasing the fraction of monomers (Nguyen et al., 2011; Hinks et al., 2018). In this study and the previous study on toluene SOA formation, $C_2H_2O$ was one of the most frequent mass difference at low and high RHs, but the peak abundance of its related compounds was much lower under elevated RH conditions (Hinks et al., 2018). $C_2H_2O$ was proposed to be from the oligomerization reaction of glycolaldehyde ($C_2H_4O_2$), which can react with carbonyl compounds by aldol condensation reactions with water as the by-product. This chemistry may dominantly affect the negative RH effect on the whole process of SOA formation.

Moreover, there may exist other processes that enlarge the difference of SOA formation under various RH conditions. Before we discuss the possible processes, the reaction pathway between *m*-xylene and OH radicals need to go through first. Reactions between *m*-xylene ($C_8H_{10}$) and OH radicals have two pathways, the H-abstraction from the methyl group and OH-addition to the aromatic ring, which generates products such as methylbenzaldehyde ($C_8H_8O$) and methylbenzyl alcohol ($C_8H_{10}O$), as

shown in Scheme 1. OH-addition is the dominant pathway, as the branching ratio of H-abstraction only accounts for 4% based on MCM. OH-addition to the aromatic ring is followed by $O_2$-adduct and isomerization to form a carbon-centered radical, which can form a dimethylphenol ($C_8H_{10}O$) or is adducted by an $O_2$ molecule forming a bicyclic peroxy radical (BPR, $C_8H_{11}O_5$) (Calvert et al., 2002; Birdsall et al., 2010; Wu et al., 2014). The BPR reacts with other $RO_2$ radicals or $HO_2$ forming the bicyclic oxy radical ($C_8H_{11}O_4$). This RO radical can get further reaction and finally form carbonylic products, such as (methyl) glyoxal

and other SOA precursors (Jenkin et al., 2003; Hallquist et al., 2009; Carlton et al., 2010; Carter and Heo, 2013), or react with $HO_2$ radicals forming bicyclic hydroxyhydroperoxides (ROOH, $C_8H_{12}O_5$), or react with other $RO_2$ radicals forming ROH ($C_8H_{12}O_4$) and R-HO ($C_8H_{10}O_4$). The self- and cross-reactions of $RO_2$ radicals also form ROOR ($C_{16}H_{22}O_{10}$) or ROOR' that is the accretion products (Berndt et al., 2018; Molteni et al., 2018). The further $O_2$-adduct of BPR can form a highly-oxygenated $RO_2$ radicals and further get reacted and finally form highly oxygenated organic molecules (HOMs) (Types 1 and 2 in Scheme

1) (Wang et al., 2017; Crounse et al., 2013; Ehn et al., 2014; Jokinen et al., 2015; Berndt et al., 2016). Dimethylphenol ($C_8H_{10}O$) as well as other products from termination reaction with benzene ring or double bond can react with OH radicals and get further reacted to form HOMs as well.

Most of HOMs can fall into extremely low or low volatility organic compounds ((E)LVOC) and a small number of HOMs are semi-volatility organic compounds (SVOC) (Bianchi et al., 2019). ELVOCs can condense onto particles but SVOCs exist in

significant fractions in both the condensed and gas phases at equilibrium. As SMPS measured, at the end of the experiment the number concentrations (not corrected) of Exp. 1 (low RH) and Exp. 7 (high RH) were $1.9 \times 10^3$ and $5.8 \times 10^2$ particles cm$^{-3}$, with a factor of 3; while the mass concentrations (not corrected) of Exp. 1 (low RH) and Exp. 7 (high RH) were 116.9 and 8.7 µg m$^{-3}$, with a factor of 13. This indicates that the size of particles at low RH are higher than that at high RH. The O:C ratios in positive and negative ion modes under low and high RH conditions were roughly calculated using the carbon and

oxygen atom numbers multiplied by the relative abundance obtained by HRMS. The O:C ratio in the positive ion mode was close to each other, 0.56 and 0.58 at low and high RHs, respectively; while the O:C ratio in the negative ion mode was different, 0.66 and 0.77 at low and high RHs, respectively. Based on the gas to particle partitioning rule, the more volatile compounds in the gas phase can condense to the particles with larger size (Li et al., 2018). It can be deduced that the particles with larger size at the reduced RH result in more SVOC in the gas phase to condense, leading to the difference of SOA mass at various

RHs. As shown in Fig. 5, more compounds with less nC (nC < 8) are present under the low RH experiment, also indicating that more SVOCs in the gas phase condense onto the particles. SVOCs tend to escape to the wet reactor wall as we discussed in Sec. 3.1, which interprets a certain proportion of SOA reduction at high RH. The wall process of the reactor enlarges the difference of SOA mass between low and high RH.

The higher O:C ratio in the negative ion mode demonstrates that the compounds in the negative ion mode are much more oxygenated than those in the positive ion mode. As shown in Fig. 5, the peak abundance at high RH is much lower in the negative ion mode than in the positive mode, indicating that the decrease of the more oxygenated compounds accounts for the larger fraction at high RH. These high O:C ratios cannot be explained by any of the formerly known oxidization pathways,

except that the formation of HOMs from $RO_2$ autoxidation is taken into consideration (Crounse et al., 2013; Barsanti et al., 2017). To our knowledge, RH does not directly impact the formation of HOMs (Li et al., 2019). It is possible that HOMs undergo further particle-phase reactions as it has been suggested in a few studies (Bianchi et al., 2019) which may be influenced by RH, but this process need to be further investigated in the future studies.

## 4. Conclusion and atmospheric implication

The current study investigates the effect of RH on SOA formation from the oxidation of $m$-xylene under low $NO_x$ conditions in the absence of seed particles. The elevated RH can significantly obstruct the SOA formation from the $m$-xylene-OH system, so that the SOA yield decrease from 14.6% at low RH to 0.8% at high RH, with a significant discrepancy of higher than one order of magnitude. The FTIR analysis shows that the proportion of oligomers with C-O-C groups from carbonyl compounds in SOA at high RH is higher than that at low RH, but the negative RH effect on SOA formation cannot be well explained as

the FTIR results cannot provide further information. From the analysis of the HRMS spectra, it is found that $C_2H_2O$ is one of the most common mass difference at low and high RHs, that the compounds with lower carbon number in the formula at low RH account for a larger proportion than those at high RH, and that the compounds at high RH have higher O:C ratios than those at low RH. The HRMS results suggest that the RH may suppress the oligomerization where water is involved as a by-product and may influence the further particle-phase reaction of high oxygenated organic molecules (HOMs) formed in the

gas phase. In addition to the chemical processes, the negative RH effect on SOA formation is enlarged based on the gas to particle partitioning rule and the different wall losses of SOA precursors. Together with the previous study on toluene SOA, it is conceivable that the effect of RH on SOA yield is a common feature of SOA formation from monocyclic aromatics oxidation under low $NO_x$ conditions and using $H_2O_2$ as the OH radical source. Our results indicate that the production of SOA from aromatics in low-$NO_x$ environments can be modulated by the ambient RH. Our study highlights the role of water in the SOA

formation, which is particularly related to chemical mechanisms used to explain observed air quality and to predict chemistry in air quality models and climate models. The clear pathway of the influence of $H_2O$ on the particle phase reaction of HOMs formed in the gas phase needs to be further studied in the future.

**Author contribution**

Qun Zhang and Yongfu Xu designed the research. Qun Zhang carried out the experiments and analyzed the data. Long Jia provided valuable advices on the experiment operations. Yongfu Xu and Long Jia provided advices on the analysis of results. Qun Zhang prepared the manuscript with contributions from all co-authors.

**Acknowledgments**

This work was supported by the National Key R&D Program of China (2017YFC0210005) and National Natural Science Foundation of China (No. 41375129).

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

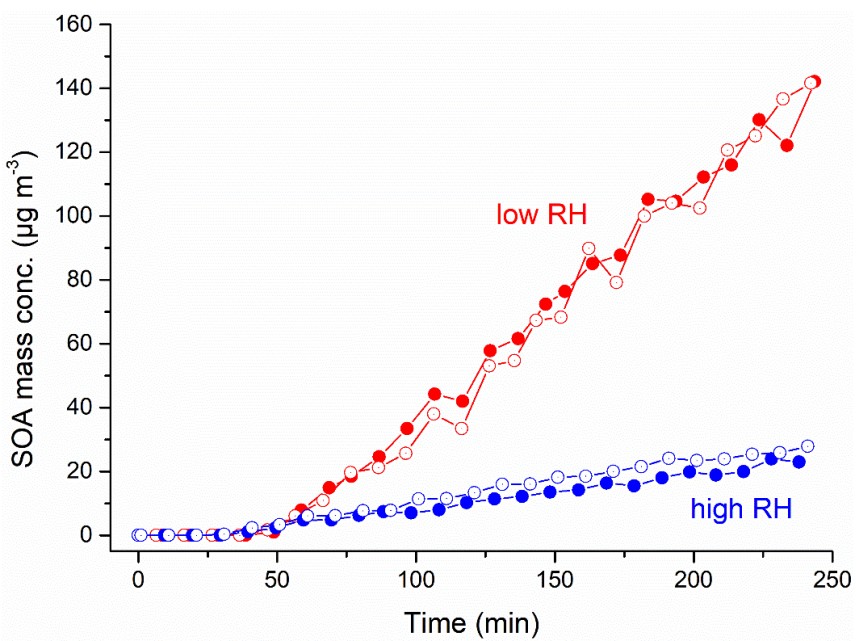

**Figure 1. Mass concentration time profiles of SOA from *m*-xylene-$H_2O_2$ photooxidation at low (Exps. 3-4) and high (Exps. 10-11) RH (corrected by particle wall loss and for the amount of LWC in particles).**

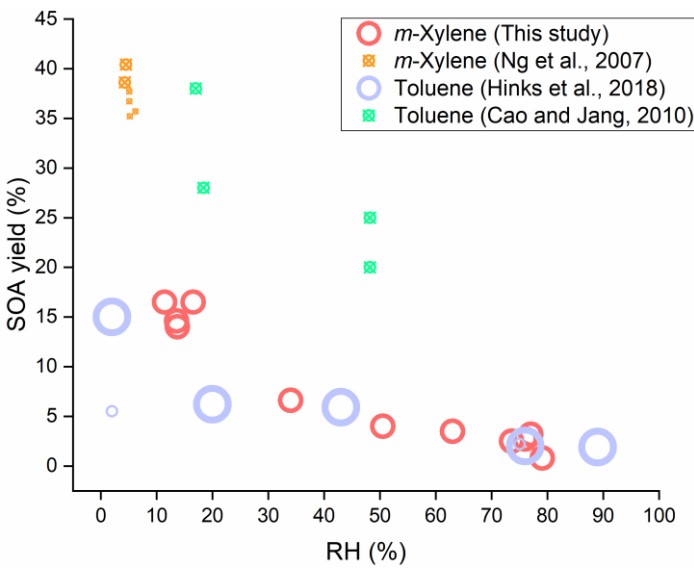

**Figure 2. SOA yields as a function of RH for the different aromatic (toluene and *m*-xylene) oxidation under low NOₓ conditions with the photolysis of H₂O₂ as the OH source. The hollow circles represent that no seed particles were introduced and the circles with a cross represent that seed particles were introduced. The size of markers indicates the magnitude of amount of reacted VOC.**

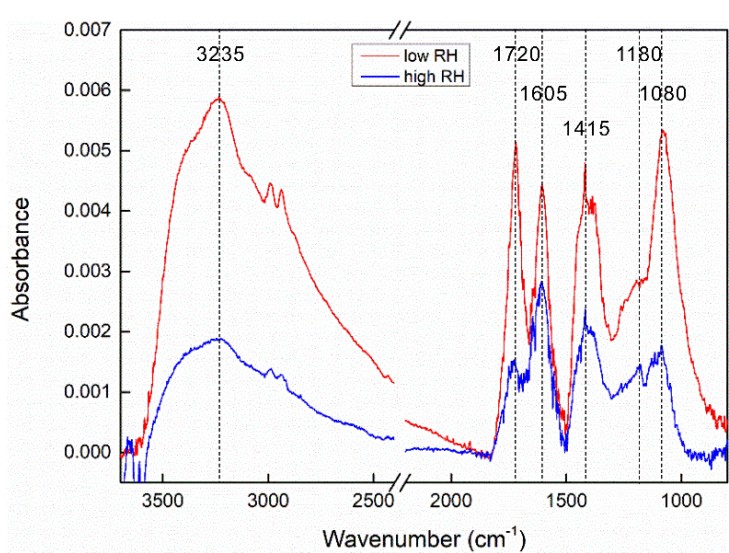

**Figure 3. FTIR spectra of particles from photooxidation of *m*-xylene-OH experiments under low (Exp. 2) and high RH (Exp. 8) conditions.**

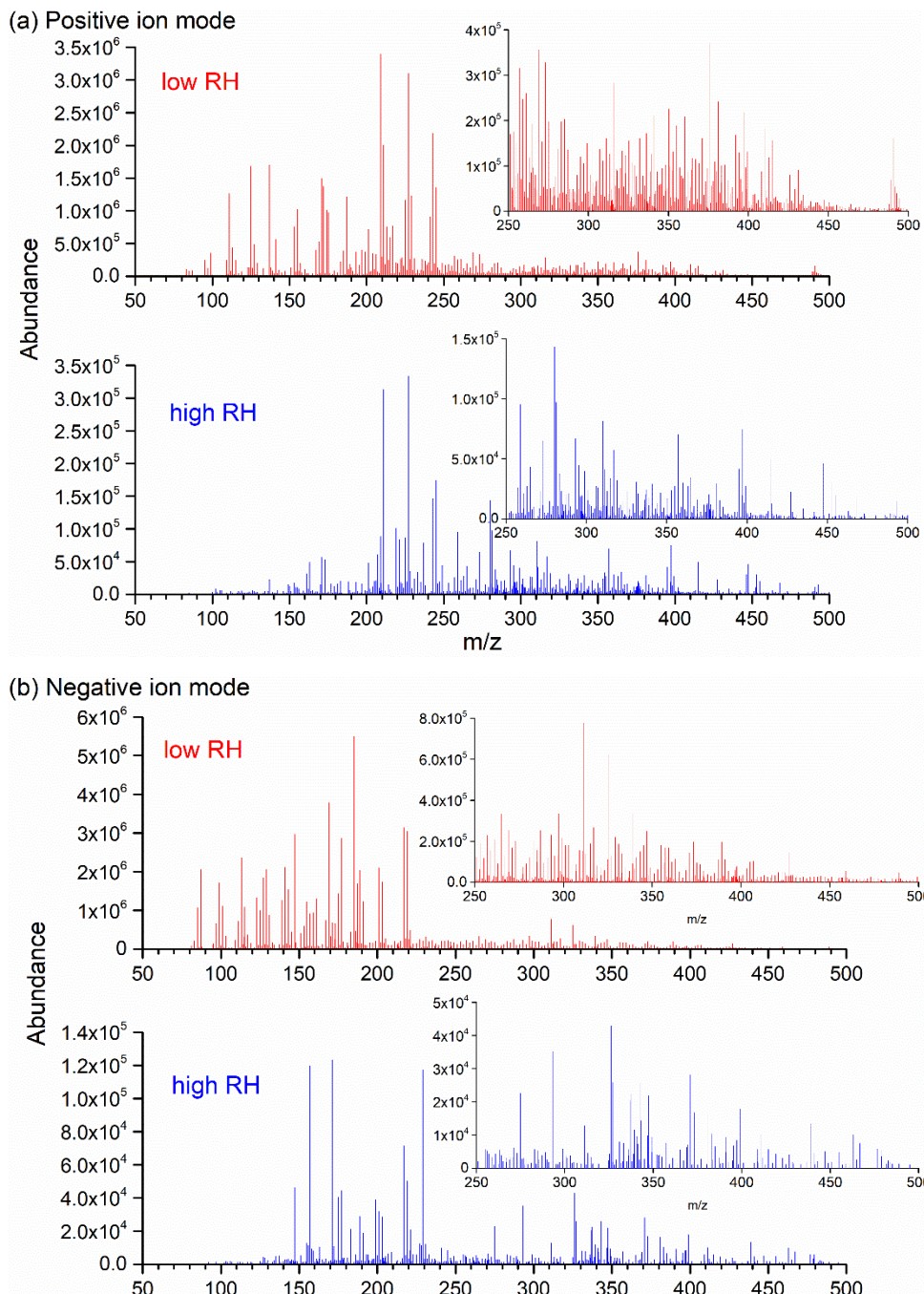

**Figure 4. Background-subtraction HESI-Q Exactive-Orbitrap MS results of SOA in both positive (a) and negative (b) ion modes from the photooxidation of *m*-xylene-OH under both low (Exp. 2) and high (Exp. 8) RH conditions (Note that the Y-axis scales for low and high RH are largely different, $10^6$ at low RH and $10^5$ at high RH).**

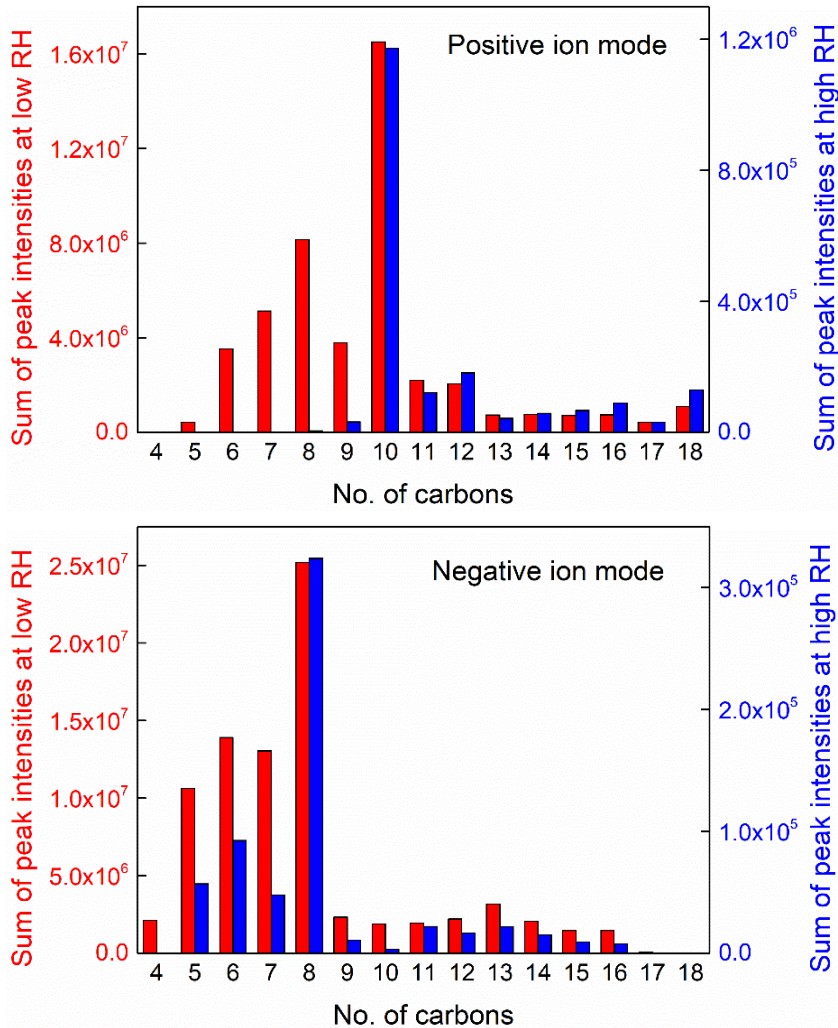

**Figure 5. Sum of peak abundance based on peaks selected in Figure 4 as a function of the number of carbon atoms under the positive ion mode and negative ion mode (Note that the Y-axis scale at low and high RH are largely different, with a label step of $4.0 \times 10^6$ at low RH and $4.0 \times 10^5$ at high RH in the positive ion mode, $5.0 \times 10^6$ at low RH and $1.0 \times 10^5$ at high RH in the negative ion mode).**

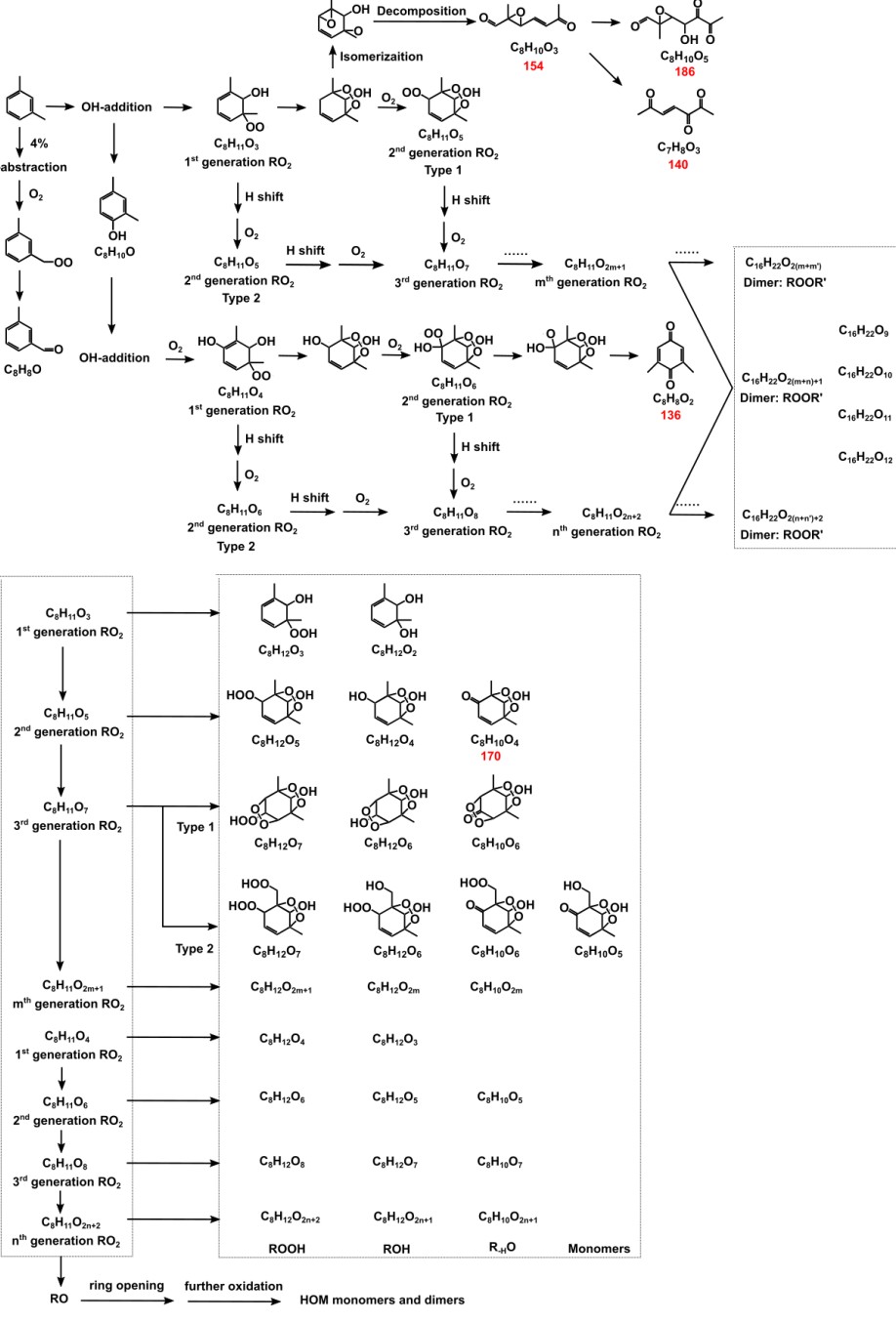

**Scheme 1. The route of OH-initiated *m*-xylene oxidation. The red number below the molecular formula is its molecular weight, which is determined by HRMS to exist in the particle phase.**

**Table 1. Experimental conditions and results at 4 h of experiments in *m*-xylene-H₂O₂ photooxidation system.**

| Exp. No. | $[m\text{-xylene}]_0$ ($\mu g\ m^{-3}$) | $[H_2O_2]_0$[a] (ppm) | RH (%) | T (°C) | $[m\text{-xylene}]_{reacted}$ ($\mu g\ m^{-3}$) | $[LWC]_{4h}$[b] ($\mu g\ m^{-3}$) | $[SOA]_{4h}$[b] ($\mu g\ m^{-3}$) | SOA yield (%) |
|---|---|---|---|---|---|---|---|---|
| 1 | 2288 | 20 | 14 | 26 | 1026 | - | 150.3 ± 15.0 | 14.6 ± 1.5 |
| 2 | 1855 | 20 | 14 | 25 | 682 | - | 95.5 ± 9.5 | 14.0 ±1.4 |
| 3 [c] | 2150 | 20 | 14 | 26 | 860 | - | 142.0 ± 14.2 | 16.5 ± 1.6 |
| 4 [c] | 2150 | 20 | 14 | 26 | 860 | - | 141.5 ± 14.1 | 16.5 ± 1.6 |
| 5 | 2157 | 20 | 34 | 26 | 923 | 3.5 ± 0.3 | 61.1 ± 6.1 | 6.6 ± 0.7 |
| 6 | 2042 | 20 | 51 | 26 | 837 | 2.9 ± 0.3 | 33.3 ± 3.3 | 4.0 ± 0.4 |
| 7 | 2233 | 20 | 63 | 27 | 722 | 5.4 ± 0.5 | 25.0 ± 2.5 | 3.5 ± 0.3 |
| 8 | 2411 | 20 | 74 | 27 | 841 | 7.7 ± 0.8 | 21.0 ± 2.1 | 2.5 ± 0.2 |
| 9 | 2029 | 20 | 79 | 27 | 947 | 4.4 ± 0.4 | 7.5 ± 0.7 | 0.8 ± 0.1 |
| 10[c] | 2150 | 20 | 77 | 26 | 860 | 18.5 ± 1.8 | 27.9 ± 2.8 | 3.2 ± 0.3 |
| 11[c] | 2150 | 20 | 76 | 26 | 860 | 15.3 ±1.5 | 22.9 ± 2.3 | 2.7 ± 0.3 |

[a]Calculated using the density and mass concentration of injected $H_2O_2$ solution, and the volume of the reactor.

[b]The mass concentration at 4 h of reaction time with particle wall loss corrected. An SOA density of 1.4 g cm⁻³ was used to obtain the SOA mass concentrations (Song et al., 2007).

5  [c]In Exps. 3-4 and 10-11, the initial concentrations of m-xylene were calculated using the density and the volume of the injected m-xylene, and the volume of the reactor; the reacted concentrations of *m*-xylene were estimated using 40% of the initial concentrations of *m*-xylene.

**Table 2. Absorbance positions of functional groups and the abundance at low (Exp. 2) and high (Exp. 8) RHs.**

| Absorption frequencies | Functionality | Intensity ($\times 10^{-3}$) | | Ratio [a] |
|---|---|---|---|---|
| | | low RH | high RH | |
| 3235 | O-H | 5.9 | 1.9 | 0.32 |
| 3000 | C-H | 4.5 | 1.4 | 0.31 |
| 1720 | C=O | 5.1 | 1.5 | 0.29 |
| 1605 | C-C of aromatic rings and conjugated C=O | 4.4 | 2.8 | 0.64 |
| 1415 | CO-H | 4.8 | 2.4 | 0.50 |
| 1180 | C-O-C, C-O and OH of COOH | 2.9 | 1.4 | 0.48 |
| 1080 | C-C-OH | 5.3 | 1.8 | 0.34 |

[a] Ratio of the intensity at high RH to that at low RH.