# Peer review of "Secondary organic aerosol formation from OH-initiated oxidation of *m*-xylene: effects of relative humidity on yield and chemical composition"

_Atmospheric Chemistry and Physics, 2019_

## Referee Comment (RC1) · Anonymous Referee #1 · 10 Mar 2019

Zhang et al. presented a chamber study that examined the effect of RH on SOA mass yields and composition. This paper is potentially useful to the SOA community. However, there are portions of the manuscripts that need to be addressed before the manuscript can be considered for publication.

1. Page 4 line 3: Clarify how $H_2O_2$ and m-xylene were introduced into the chamber. Via an injection into a glass bulb using a syringe? Using a bubbler? How did the authors determine when the chamber contained 20 ppm of $H_2O_2$? Was the concentration of gas-phase $H_2O_2$ in the chamber measured in real-time? If yes, what instrument was used?

2. Page 4 line 5: Explain the rationale behind not using any seed aerosols in this study. Seed aerosols are typically used in chamber studies to promote the condensation of SOA-forming vapors onto seed aerosol instead of the chamber walls. The mass yields reported by the authors are likely under-estimated since most of the vapors are likely lost the chamber walls in these experiments (See examples provided in Zhang et al., PNAS 2014, Nah et al., ACP 2016, 2017). Vapor wall loss is also going to affect the types of products formed in these SOA experiments since highly oxygenated and least volatile compounds are lost to the chamber walls are faster rates (See Zhang et al., ACP 2015). The authors should comment on how vapor wall loss affects their results. Can they also provide an estimation on how much their SOA mass yields are under-estimated by?

Nah, T., McVay, R. C., Zhang, X., Boyd, C. M., Seinfeld, J. H., and Ng, N. L.: Influence of seed aerosol surface area and oxidation rate on vapor wall deposition and SOA mass yields: a case study with $\alpha$-pinene ozonolysis, Atmos. Chem. Phys., 16, 9361-9379, https://doi.org/10.5194/acp-16-9361-2016, 2016.

Nah, T., McVay, R. C., Pierce, J. R., Seinfeld, J. H., and Ng, N. L.: Constraining uncertainties in particle-wall deposition correction during SOA formation in chamber experiments, Atmos. Chem. Phys., 17, 2297-2310, https://doi.org/10.5194/acp-17-2297-2017, 2017.

Zhang, X., Schwantes, R. H., McVay, R. C., Lignell, H., Coggon, M. M., Flagan, R. C., and Seinfeld, J. H.: Vapor wall deposition in Teflon chambers, Atmos. Chem. Phys., 15, 4197-4214, https://doi.org/10.5194/acp-15-4197-2015, 2015.

3. Page 4 line 11: How were the particle wall loss rates determined? In seed aerosols only experiments? Were these particle wall loss rates measured by tracking the decay of the aerosol mass or volume? How often were particle wall loss experiments conducted? Were the reported particle wall loss rates consistent with previously measured rates? Was the particle wall loss rate always faster in high RH experiments or is this

measurement within experimental uncertainty?

4. Page 4 line 14: It is not clear how the aerosol LWC was calculated. More details should be provided.

5. Page 4 line 20: The PILS only samples water-soluble species in the SOA, not the total SOA composition. Hence, the compositional results reported by the authors in this study are really the water-soluble species, and the authors should specify this in their manuscript. On a related note, why did the authors decided to collect aerosol samples with a PILS instead of on filters. Filter collection and analysis would have allowed them to analyze both the water-soluble and water-insoluble species. Do the authors know what fraction of the SOA formed is composed of water-soluble vs. water-insoluble species?

6. Page 5 line 8: Show the corresponding reaction time profile of m-xylene measured by the GC-MS that accompanied the observed SOA growth for the four experiments. This can be placed in the supplementary information. It is currently unclear how quickly the reactions took place. Perhaps the time profiles can be used to explain the differences in SOA formation in dry vs. humid conditions? For example, did m-xylene react faster in the dry experiments thus resulting in higher SOA mass yields? Ng et al., ACP 2007 previously showed that SOA formation in the m-xylene system will be faster at faster oxidation rates. From Fig. 1, it looks like peak SOA growth was not achieved at the end of the dry experiments (SOA mass looks like it may still increase). Why the authors decide to stop these dry experiments early? Won't that affect their calculated SOA mass yields?

Ng, N. L., Kroll, J. H., Chan, A. W. H., Chhabra, P. S., Flagan, R. C., and Seinfeld, J. H.: Secondary organic aerosol formation from m-xylene, toluene, and benzene, Atmos. Chem. Phys., 7, 3909-3922, https://doi.org/10.5194/acp-7-3909-2007, 2007.

7. Page 5 line 21: Regarding the authors' definition of SOA yield, did they calculate the SOA yield by dividing the SOA mass obtained at the end of the experiment by the total

reacted m-xylene at the end of the experiment? If yes, why did they decide to use this calculation? Previous chamber studies calculated the SOA mass yield by taking the ratio of the SOA mass at peak SOA mass divided by the mass of VOC reacted. Was peak SOA mass only reached at the end of each experiment (reaction time profiles of SOA mass growth with the corresponding reacted m-xylene for the four experiments will be useful; see comment 6)? Related to this point, are the authors confident that peak SOA mass have already occurred before they ended their experiment. Given that the authors are comparing their measured SOA mass yields with previous studies, they should make sure that their calculation of SOA mass yields are consistent with those of previous studies before they compare mass yields.

8. Page 5 line 23: How was LWC subtracted from the SOA measurement? How did the authors determine the amount of LWC in the aerosols? The authors should briefly describe this process even if this was previously mentioned in one of their previous paper. The sentence "It should be pointed out that... would evaporate back into the gas phase when aerosol water is removed" is confusing. The experimental section did not mention that authors removed aerosol water prior to SMPS measurement. If aerosol water was not removed prior to SMPS measurement, then this sentence seems out of place. Unless the authors are proposing a hypothetical situation?

9. Page 5 line 27: Table 1 should also state the m-xylene concentration in ug/m3 so that readers can more easily compare this study's reaction conditions with those of previous studies.

10. Page 5 line 28: Why were the temperatures in the high RH experiments higher than those in the low RH experiments?

11. page 7 line 25: A magnified view of the mass spectra shown in Fig. 3 would be more useful for comparison purposes.

12. Page 7 line 27: The sentence "It should be pointed out that the signal intensities..." is confusing. Were the mass spectra for the different experiments obtained using different MS operation conditions (e.g., ESI spray conditions, MS collision gas)?

13. Page 7 line 25 to page 8 line 11: The mass peaks discussed here do not seem to be the major peaks shown in Fig. 3. Why did the authors choose to focus their discussion only on these selected peaks? The major peaks seem to be m/z > 200. How were these products formed? The authors should include a list of all the product ions identified. Do these identified products match their proposed reaction mechanism show in Scheme 1?

14: General comment: What compounds are the -ve MS mode sensitive to? Were these compounds identified in their collected mass spectra?

15. General comment: The authors mentioned in the experimental system that they used a HPLC-MS system in their study. It is not clear from their presented results whether this was the case. Was HPLC not used to separate the products via their volatilities prior to MS analysis?

16. Page 9 line 30: The authors claimed that they used the distribution of relative intensity of SOA products with the same carbon number to investigate the potential RH effect on HOMs. The rationale behind this course of action seems to contradict their previous statement in Page 7 line 27 that signal intensities can be biased by ionization properties.

17. Scheme 1: The authors should indicate explicitly in Scheme 1 which are the products that they have identified.

18. Page 10 line 27: The sentence "Together with the previous study on toluene SOA, it is conceivable that the effect of RH on SOA yield is a common feature of SOA formation from oxidation of all OH-initiated aromatics" is too generalized and needs to be re-phrased. As discussed by the authors in their introduction, an increase RH does not necessarily cause a decrease in SOA mass yields in aromatics SOA systems. Other factors such as NOx can also alter the effect that RH has on SOA mass yields in these

systems.

---

## Referee Comment (RC2) · Anonymous Referee #2 · 11 Mar 2019

Overview:

This study explore the role of relative humidity (RH) on the m-xylene SOA formation under OH initiated no NOx condition. The results showed that the SOA yield under high RH is significantly lower than that under low RH conditions. This study provides SOA yields and particle-phase SOA products under different RH levels. The LWC was measured by the particles mass deduction in the DAASS. The authors measured the SOA compositions by using a Fourier transform infrared (FTIR) spectra and ultrahigh performance liquid chromatograph electrospray ionization-high-resolution mass spectrometer (UPLC-ESI-HRMS). The authors reported that SOA yield found to be about 7

times high in dry condition (RH∼13%) than that in wet condition (RH∼75%). Overall, the experimental data to show the impact of RH on SOA yields and products and the conclusion originating from the chamber are doubtful. The small chamber used in this study will be significantly influenced by the gas-wall processes of organic species increasing the uncertainty in data and interpretation of results. This paper in its current form is not acceptable. Please find the comments below.

Major Comments:

1. The aromatics VOCs are gas pollutants that is found to be high in urban environments where the NOx is also abundant. It is unclear why the authors chose no NOx condition to study the humidity effects on the formation of xylene SOA. Clarify this. 2. What is the effect of the wall on the loss gaseous H2O2? H2O2 is very hydrophilic and sticky to the wall. When RH is high, the water on the chamber wall becomes high forming a water film. This wat film can absorb a large amount of H2O2 and modulate the concentration of OH radicals. Please clarify how the authors measured OH radical concentrations under varying RH conditions. Why did the author use 20 ppm of H2O2 which was 40 times higher than the m-xylene concentration? What is the photolysis rate constant of H2O2 in the chamber ? 3. The size of chamber used in this study was one cubic meter and relatively very small. Thus, the wall of chemical species is very high. Additionally, the loss of chemical species to the chamber is very sensitive to humidity. The impact of RH on SOA yields can be very uncertain and doubtful. The reduction of SOA yields at the high humidity is more likely due to the chemical loss to the wet chamber wall. Thus, the conclusion made by the authors could be incorrect. Hydrophilic products and reactive chemical species can more deposit to the wall at high humidity. 4. In order to analyze the chemical compositions in gas and particles phase using a variety of aerosol, a large amount of air volume should be collected. The size of the chamber used in this study was only one cubic. It is hard to believe how the authors analyzed gas and aerosol with the air volume less than one cubic meter. Additionally, the chamber volume was getting small as the experiment progressed. The

consumption of the chamber air increased the wall effect. The authors should clarify this problem. 5. Page 5, line 4. The Master Chemical Mechanism can only provide the gas-phase reaction pathways. The yield of the products in particle phase may not directly connected to the yield of products in gas-phase. How does the author compare gas-phase oxygenated m-xylene products predicted using MCM to the measured particle-phase products from HRMS ? 6. Page 5, line 11. The value of the maximum SOA mass in Figure 1 is not consist with the values reported in the text and the Table 1. The value of SOA mass under 73.6% and 79.1% in Figure 1 is about 40 and 10 ug/m3 but the value reported in the text is only 1.9 and 0.8 ug/m3 and the value reported in Table 1 is 15.8 and 7.9 ug/m3. 7 It is not clear how much LWC was present at the end of experiments and how much SOA mass was obtained after subtracting the LWC from total aerosol mass. What is the effects of LWC on the SOA formation in this study? The author mention that LWC can explain the positive effect of RH on SOA formation under high NOx condition. What is the difference in LWC between SOA with the high NOx condition and that with the low or no NOx condition? 8. What is the particle size distribution of m-xylene SOA? Does all of the particle size smaller than 1000 nm and within the SMPS measurement range? 9. Section 3.2. The intensity of the functional groups in FTIR spectrum was correlated to the sample mass. What was the SOA mass that collected on the disk and that measured using FTIR? Or does the author use same sampling duration for both RH conditions? What was the collection efficiency of the impactor on a sampling disk as a function of the particle size? Without knowing the mass of measured SOA, it is unreasonable to compare the peak intensity of the functional group between SOA from different samples. 10. Figure 2 and Table 2. There is also peak at 3000 cm-1 which is missing in Table 2. 11. What is the measured glyoxal fraction in m-xylene SOA? Was oligomerization impacted by the RH in this study? Even though the concentration of highly oxygenated molecule (HOM) is much lower at high RH, the overall trend of the SOA mass, which is much less at high RH compare to low RH, can not be explained by solely through HOMs. As mentioned in the previous comment above, the effect of the wet wall on SOA formation can be very significant

particularly in small reaction. The time scale of the gas-wall partitioning of organic species can be significantly fast and results in the less SOA yields at higher humidity. 12 The author claimed that the increase of the C-O-C stretching was resulted from the oligomerization of carbonyls under the high RH condition. However, the IR absorption at 1080 cm-1 can be also due to the existence of C-OH group. Clarify this. 13. Fig. 2. Authors assigned the peak at 1600 cm-1 as carboxylate. The reviewer is doubt this assignment. In general, dry organic aerosol cannot produce carboxylic acid ions. Even if the organic aerosol is produce in the wet condition, the aerosol water content is not enough to product the dissociation of carboxylic acid. In general, the pKa of carboxylic acid ranges from 2 to 6. Even in the dilution in water, less than 1% of carboxylic acid is dissociable. In SOA, most carboxylic acids will be in the un-dissociated form. 14. Fig. 4. Based on Fig. 4, a large mass appeared in negative ion mode suggesting that the aerosol has a large fraction of carboxylic acid. It is contradictable compared to either MCM simulation or the conclusion by the authors in the glyoxal was abundant. In general, a large fraction of gaseous products from MCM prediction are alcohols and carbonyls, and amount of carboxylic acids are small. Please clarify this. 15. Fig. 4. Low carbon number products are generally more volatile than high carbon number products. Fig.4 showed that low carbon number products are high with the SOA with the low RH, possibly suggesting that volatile low carbon number products more likely deposited to the wall due to the gas-wall process. 16. What was the RH of the environment inside the FTRI spectrometer when FTIR spectra were obtained for Fig. 4 ? 17. What is the atmospheric implication of this study? What is the potential impact of RH on p-xylene and o-xylene as well as other aromatics? Will other aromatics also have the similar RH effects with m-xylene? What is the uncertainty of this study? Does the impaction or the PILS sampling has uncertainty? What is the duration of the experiments? 18. There are numerous grammatical problems. The manuscript needs to be approved by a native English speaker. 19. Page 3 Line 13. The author may need to provide the reason why the author set the density of m-xylene SOA is set as 1.4 g/cm. According to the citation Ng et al. 2007 gives the density of m-xylene SOA as 1.33 $\pm$

0.1 g/cm and Sato et al. 2007 provide the density of Toluene SOA as 1.42 ± 0.8 g/cm. 20. Page 3 Line 17. The author mention about the uncertainty of SMPS. However, the uncertainty of SMPS measurement for the used data in the paper was not reported. 21. Page 4 Line 6. For comparing the SOA yield between cited values and that of this study, it may need to provide the error range of the values. Additionally, numerous data of this paper need errors. 22. Pages 6 and 7 (section 3.2): The description to construct the functional group distribution using FTIR spectra is unclear. How to separate the FTIR peaks for each functional group ? The intensity of each function group varies with vibration force constant and peak broadening changes with compositions.

Minor comments: Page 2, line 21. The sentence is confused that it compares the RH effects between low NOx condition and with NOx condition. Figure 3. The y-axis scale is negative for high RH and positive for Low RH. It is better to make them as a same positive scale. Figure 5: RH scale should range from 0 to 100 (negative is incorrect). Page 2 line 15: it is better to use "have been conducted". Page 2 line 19: after "as an OH radical source," there supposed to use period instead of comma. Table 3. The form of the table is better to unify with other tables.

---

## Author Comment (AC1) · 24 May 2019

Please also see the attached file of XU-Response-RC1 as acp-2019-20-supplement.pdf because marked colors, figures and tables are missed in this plain text.

Response to Reviewer 1

We greatly appreciate the time and effort that reviewer 1 spent in reviewing our manuscript. The comments are really thoughtful and helpful to improve the quality of our paper. Reviewer 1 has provided both main comments and other specific com-

ments. Below we make a point-by-point response to these comments. According to editor's requirement, the response to the referee 1 is structured in the following sequence: (1) comments from the referee in black color, (2) our response in blue color, and (3) our changes in the revised manuscript in red color.

Zhang et al. presented a chamber study that examined the effect of RH on SOA mass yields and composition. This paper is potentially useful to the SOA community. However, there are portions of the manuscripts that need to be addressed before the manuscript can be considered for publication.

1. Page 4 line 3: Clarify how H2O2 and m-xylene were introduced into the chamber. Via an injection into a glass bulb using a syringe? Using a bubbler? How did the authors determine when the chamber contained 20 ppm of H2O2? Was the concentration of gas-phase H2O2 in the chamber measured in real-time? If yes, what instrument was used?

H2O2 and m-xylene were introduced into the reactor along with the zero air flow via an injection into a three-way tube using a syringe. The concentration of gas-phase H2O2 in the reactor was not measured but calculated. To obtain a certain concentration of H2O2, the density and mass concentration of injected H2O2 solution, and the volume of the reactor were used to calculate the volume of H2O2 solution that needed to be injected.

2. Page 4 line 5: Explain the rationale behind not using any seed aerosols in this study. Seed aerosols are typically used in chamber studies to promote the condensation of SOA-forming vapors onto seed aerosol instead of the chamber walls. The mass yields reported by the authors are likely under-estimated since most of the vapors are likely lost the chamber walls in these experiments (See examples provided in Zhang et al., PNAS 2014, Nah et al., ACP 2016, 2017). Vapor wall loss is also going to affect the types of products formed in these SOA experiments since highly oxygenated and least volatile compounds are lost to the chamber walls are faster rates (See Zhang et al.,

[Figure]

ACP 2015). The authors should comment on how vapor wall loss affects their results. Can they also provide an estimation on how much their SOA mass yields are underestimated by? Nah, T., McVay, R. C., Zhang, X., Boyd, C. M., Seinfeld, J. H., and Ng, N. L.: Influence of seed aerosol surface area and oxidation rate on vapor wall deposition and SOA mass yields: a case study with $\alpha$-pinene ozonolysis, Atmos. Chem. Phys., 16, 9361-9379, https://doi.org/10.5194/acp-16-9361-2016, 2016. Nah, T., McVay, R. C., Pierce, J. R., Seinfeld, J. H., and Ng, N. L.: Constraining uncertainties in particle-wall deposition correction during SOA formation in chamber experiments, Atmos. Chem. Phys., 17, 2297-2310, https://doi.org/10.5194/acp-17-2297-2017, 2017. Zhang, X., Schwantes, R. H., McVay, R. C., Lignell, H., Coggon, M. M., Flagan, R. C., and Seinfeld, J. H.: Vapor wall deposition in Teflon chambers, Atmos. Chem. Phys., 15, 4197-4214, https://doi.org/10.5194/acp-15-4197-2015, 2015.

We agree with the reviewer that seed aerosols can promote the condensation of SOA-forming vapors onto seed aerosol instead of the chamber walls. However, inorganic salt can both participate into the SOA formation and change the reaction environment such as providing acidic surface and aqueous environment. These properties of seed aerosols probably interfere with the RH effect on SOA formation, as the RH combined with seed aerosols complicate the m-xylene-OH system. Losses of organic vapors to the chamber wall can be substantial. The fact that seed aerosols were not artificially introduced can probably lead to the underestimation of SOA. Thus, we have added a paragraph to comment on how vapor wall loss affects our results at the end of Sec. 3.1 in the revised manuscript, but we cannot provide a factor of underestimation of SOA yields.

It should be noted that seed aerosols were not artificially introduced throughout all the experiments, which could lead to the underestimation of SOA, as SOA-forming vapors partly condense to the chamber walls instead of particles (Matsunaga and Ziemann, 2010; Zhang et al., 2014). The extent to which vapor wall deposition affects SOA mass yields depends on the specific parent hydrocarbon system (Zhang et al., 2014; Zhang

et al., 2015; Nah et al., 2016; Nah et al., 2017). Zhang et al (2014) have estimated two m-xylene systems under low NOx conditions and concluded that SOA mass yields were underestimated by factors of 1.8 (Ng et al., 2007) and 1.6 (Loza et al., 2012) under low RH conditions. In addition, the excess use of H2O2 can lead to an excess OH radicals, leading to a less underestimation of SOA formation as the losses of SOA-forming vapors can be mitigated via the use of excess oxidant concentrations (Nah et al., 2016). Thus, the underestimation of SOA formation can be limited. In fact, the wall loss of m-xylene was not taken into consideration of calculation of mass yields, which generally overestimates the mass yields.

Loza, C. L., Chhabra, P. S., Yee, L. D., Craven, J. S., Flagan, R. C., and Seinfeld, J. H.: Chemical aging of m-xylene secondary organic aerosol: laboratory chamber study, Atmos. Chem. Phys., 12, 151-167, 10.5194/acp-12-151-2012, 2012. Matsunaga, A., and Ziemann, P. J.: Gas-wall partitioning of organic compounds in a Teflon film chamber and potential effects on reaction product and aerosol yield measurements, Aerosol Sci. Technol., 44, 881-892, 10.1080/02786826.2010.501044, 2010. Nah, T., McVay, R. C., Zhang, X., Boyd, C. M., Seinfeld, J. H., and Ng, N. L.: Influence of seed aerosol surface area and oxidation rate on vapor wall deposition and SOA mass yields: a case study with $\alpha$-pinene ozonolysis, Atmos. Chem. Phys., 16, 9361-9379, 10.5194/acp-16-9361-2016, 2016. Nah, T., McVay, R. C., Pierce, J. R., Seinfeld, J. H., and Ng, N. L.: Constraining uncertainties in particle-wall deposition correction during SOA formation in chamber experiments, Atmos. Chem. Phys., 17, 2297-2310, 10.5194/acp-17-2297-2017, 2017. Ng, N. L., Kroll, J. H., Chan, A. W. H., Chhabra, P. S., Flagan, R. C., and Seinfeld, J. H.: Secondary organic aerosol formation from m-xylene, toluene, and benzene, Atmos. Chem. Phys., 7, 3909-3922, 10.5194/acp-7-3909-2007, 2007. Zhang, X., Cappa, C. D., Jathar, S. H., McVay, R. C., Ensberg, J. J., Kleeman, M. J., and Seinfeld, J. H.: Influence of vapor wall loss in laboratory chambers on yields of secondary organic aerosol, Proc. Natl. Acad. Sci. U. S. A., 111, 5802-5807, 10.1073/pnas.1404727111, 2014.

3. Page 4 line 11: How were the particle wall loss rates determined? In seed aerosols only experiments? Were these particle wall loss rates measured by tracking the decay of the aerosol mass or volume? How often were particle wall loss experiments conducted? Were the reported particle wall loss rates consistent with previously measured rates? Was the particle wall loss rate always faster in high RH experiments or is this measurement within experimental uncertainty?

Particle wall loss rates were generally measured in seed aerosol experiments by tracking the decay of the aerosol volume. For the same volume of new reactor, the wall loss rates were evaluated. We also checked the wall loss rate for the old reactor. Particle wall loss rate constant varies from $3 \times 10^{-5}$ s-1 to $6 \times 10^{-5}$ s-1 at the RH range of 5% to 90% with a trend of increase with RH, but their relationship is not statistically significant. The average particle wall loss rate constant is $(3.8 \pm 0.8) \times 10^{-5}$ s-1 at $(13 \pm 10)$% RH and $(4.2 \pm 1.8) \times 10^{-5}$ s-1 at $(79 \pm 10)$% RH, respectively. The relatively large wall loss rate at high RH and small wall loss rate at low RH are used in our correction of particle wall loss to look at RH effects in this study.

4. Page 4 line 14: It is not clear how the aerosol LWC was calculated. More details should be provided.

Taking the reviewer's advice, we have added some sentences about the details of LWC measurement at the end of the first paragraph of Sec. 2.2.

Thus, here a brief introduction is only given. After the lights were turned off in high RH experiments, the SMPS was modified to the dry mode through adding a Nafion dryer (Perma Pure MD-700-12F-3) to the sampling flow and a Nafion dryer (Perma Pure PD-200T-24MPS) to the sheath flow, leading to the reduction of RH in the sample air to 10 % and that in the sheath to 7 %. After modifying to the dry mode, the humid air in SMPS was quickly replaced by dry air through venting the sheath air at 5 L min-1, and then the dry aerosol was measured by SMPS. The LWC was determined by the difference of the particle mass concentrations before and after the modification of the

dry mode.

5. Page 4 line 20: The PILS only samples water-soluble species in the SOA, not the total SOA composition. Hence, the compositional results reported by the authors in this study are really the water-soluble species, and the authors should specify this in their manuscript. On a related note, why did the authors decided to collect aerosol samples with a PILS instead of on filters. Filter collection and analysis would have allowed them to analyze both the water-soluble and water-insoluble species. Do the authors know what fraction of the SOA formed is composed of water-soluble vs. water-insoluble species?

We agree with the reviewer that the PILS samples water-soluble species in the SOA. Nevertheless, after the FTIR measurement of SOA samples collected on ZnSe windows, the ZnSe window was washed with ultrapure water and was measured by FTIR again, no absorbance was observed on FTIR spectra. It can be believed that the SOA compositions are almost all water-soluble species and the PILS samples almost all SOA components. In addition, we agree with the reviewer that filter collection samples both water-soluble and water-insoluble species in the SOA. However, the filter-based analysis has its limitation, including adsorption of organic vapors and evaporation of semi-volatile organic compounds from the filter surface, leading to some uncertainties in the identification of SOA components. Moreover, Bateman et al. (2010) compared the off-line mass spectra of SOA samples from limonene ozonolysis collected by PILS with those collected on filters and found that the peak abundance, organic mass to organic carbon ratios, and the average O:C ratio are essentially identical. Water-soluble species account for the vast majority of SOA.

Bateman, A. P., Nizkorodov, S. A., Laskin, J., and Laskin, A.: High-resolution electrospray ionization mass spectrometry analysis of water-soluble organic aerosols collected with a particle into liquid sampler, Anal. Chem., 82, 8010-8016, 10.1021/ac1014386, 2010.

6. Page 5 line 8: Show the corresponding reaction time profile of m-xylene measured by the GC-MS that accompanied the observed SOA growth for the four experiments. This can be placed in the supplementary information. It is currently unclear how quickly the reactions took place. Perhaps the time profiles can be used to explain the differences in SOA formation in dry vs. humid conditions? For example, did m-xylene react faster in the dry experiments thus resulting in higher SOA mass yields? Ng et al., ACP 2007 previously showed that SOA formation in the m-xylene system will be faster at faster oxidation rates. From Fig. 1, it looks like peak SOA growth was not achieved at the end of the dry experiments (SOA mass looks like it may still increase). Why the authors decide to stop these dry experiments early? Won't that affect their calculated SOA mass yields? Ng, N. L., Kroll, J. H., Chan, A. W. H., Chhabra, P. S., Flagan, R. C., and Seinfeld, J. H.: Secondary organic aerosol formation from m-xylene, toluene, and benzene, Atmos. Chem. Phys., 7, 3909-3922, https://doi.org/10.5194/acp-7-3909-2007, 2007.

Taking the reviewer's advice, we have added the reaction time profile of m-xylene measured by the GC-MS that accompanied the observed SOA growth for the four experiments in the supplementary information (see Fig. S1).

Fig. S1. Reaction time profiles of m-xylene measured by the GC-MS that accompanied the observed SOA growth for the four experiments

As shown in the time profile in Fig. S1, the reacted m-xylene account for around 40% of the initial m-xylene in both high and low RH experiments. m-Xylene did not react faster in the dry experiments which may lead to the higher SOA formation.

As the reviewer pointed out, peak SOA growth was not achieved at the end of the dry experiments from Fig. 1 and SOA mass still increase. As the experiments were conducted under low NOx condition, the SOA mass will increase unless m-xylene has all reacted. The SOA mass formation and reacted m-xylene are both nearly linear. The SOA yields will be basically constant no matter when we stop the reaction. In addition,

the SOA-forming vapor could loss less if the reaction time was relatively short 4 h. Thus, we decided to stop the experiments early.

7. Page 5 line 21: Regarding the authors' definition of SOA yield, did they calculate the SOA yield by dividing the SOA mass obtained at the end of the experiment by the total reacted m-xylene at the end of the experiment? If yes, why did they decide to use this calculation? Previous chamber studies calculated the SOA mass yield by taking the ratio of the SOA mass at peak SOA mass divided by the mass of VOC reacted. Was peak SOA mass only reached at the end of each experiment (reaction time profiles of SOA mass growth with the corresponding reacted m-xylene for the four experiments will be useful; see comment 6)? Related to this point, are the authors confident that peak SOA mass have already occurred before they ended their experiment. Given that the authors are comparing their measured SOA mass yields with previous studies, they should make sure that their calculation of SOA mass yields are consistent with those of previous studies before they compare mass yields.

As the reviewer pointed out, the SOA yield in this study is defined by the ratio of the SOA mass obtained at the end of the experiment to the total reacted m-xylene at the end of the experiment. As the experiments were conducted under low NOx condition, the SOA mass would increase unless all m-xylene reacted. The experiment for 4-6 h is a ubiquitous reaction time used in many previous studies. Indeed, the SOA yield generally increases with time. If the relationship between the yield and time is extrapolated to 6 h, the yield is increased by 45% relative to that at 4 h, which can be compared with many previous studies (Cao and Jang, 2010; Hinks et al., 2018). Most importantly, as the purpose of our study is to investigate the RH effect on SOA formation, the reaction time of 4 h is sufficient to compare the SOA formation and to sample for SOA component analysis. Furthermore, a relatively short reaction time can minimize the wall loss of oxidized species and limit the further SOA mass uncertainty.

Cao, G., and Jang, M.: An SOA model for toluene oxidation in the presence of inorganic aerosols, Environ. Sci. Technol., 44, 727-733, 10.1021/es901682r, 2010. Hinks, M. L.,

Montoya-Aguilera, J., Ellison, L., Lin, P., Laskin, A., Laskin, J., Shiraiwa, M., Dabdub, D., and Nizkorodov, S. A.: Effect of relative humidity on the composition of secondary organic aerosol from the oxidation of toluene, Atmos. Chem. Phys., 18, 1643-1652, 10.5194/acp-18-1643-2018, 2018.

8. Page 5 line 23: How was LWC subtracted from the SOA measurement? How did the authors determine the amount of LWC in the aerosols? The authors should briefly describe this process even if this was previously mentioned in one of their previous paper. The sentence "It should be pointed out that…would evaporate back into the gas phase when aerosol water is removed" is confusing. The experimental section did not mention that authors removed aerosol water prior to SMPS measurement. If aerosol water was not removed prior to SMPS measurement, then this sentence seems out of place. Unless the authors are proposing a hypothetical situation?

Taking the reviewer's advice, we have added some sentences about the details of LWC measurement at the end of the first paragraph of Sec. 2.2 (same with the reply of Comment 4).

Thus, here a brief introduction is only given. After the lights were turned off in high RH experiments, the SMPS was modified to the dry mode through adding a Nafion dryer (Perma Pure MD-700-12F-3) to the sampling flow and a Nafion dryer (Perma Pure PD-200T-24MPS) to the sheath flow, leading to the reduction of RH in the sample air to 10 % and that in the sheath to 7 %. After modifying to the dry mode, the humid air in SMPS was quickly replaced by dry air through venting the sheath air at 5 L min-1, and then the dry aerosol was measured by SMPS. The LWC was determined by the difference of the particle mass concentrations before and after the modification of the dry mode.

When we measured the LWC, the aerosol water should be removed after the SMPS was modified. For clarification, we have rephrased the sentence pointed out by the reviewer, "The removal of aerosol water during the LWC measurement may cause the

dissolved species that are probably volatile/semi-volatile compounds to evaporate back into the gas phase. Thus, SOA concentrations for high RH conditions were slightly underestimated, but the underestimation is extremely low and can be negligible."

9. Page 5 line 27: Table 1 should also state the m-xylene concentration in ug/m3 so that readers can more easily compare this study's reaction conditions with those of previous studies.

Taking the reviewer's advice, we have modified the m-xylene concentration in ug/m3 in Table 1.

Table 1. Experimental conditions, SOA concentrations and yields at the end of the experiments in m-xylene-OH oxidation system. Exp. No. [m-xylene]0 ($\mu$g m-3) [m-xylene]reacted ($\mu$g m-3) RH (%) T ($^\circ$C) [SOA]e ($\mu$g m-3) SOA yield (%) 1 2287.9 1026.3 13.6 25.9 150.3 $\pm$ 15.0 14.6 $\pm$ 1.5 2 1855.5 682.0 13.7 25.3 95.5 $\pm$ 9.5 14.0 $\pm$ 1.4 3 2410.8 941.4 73.6 27.5 21.0 $\pm$ 2.1 2.2 $\pm$ 0.2 4 2029.1 946.9 79.1 27.4 7.5 $\pm$ 0.7 0.8 $\pm$ 0.1 [SOA]e indicates the mass concentration of SOA at the end of each experiment with particle wall loss corrected.

10. Page 5 line 28: Why were the temperatures in the high RH experiments higher than those in the low RH experiments?

The accuracy of temperature controller led to this fluctuation that the temperatures in the high RH experiments were higher than those in the low RH experiments. The highest difference between low and high RH experiment was 2 $^\circ$C. The temperature effect on SOA formation has been investigated in some previous studies about the m-xylene oxidation. According to previous studies about the temperature effect of SOA formation from m-xylene oxidation (Takekawa et al., 2003; Qi et al., 2010), an increase of 2 $^\circ$C can lead to a mean SOA mass decrease by 4.6%. It can be concluded that the 2 $^\circ$C higher temperature in high RH experiments cannot significantly affect the results of RH effect on SOA formation in this study.

Qi, L., Nakao, S., Tang, P. and Cocker, D. R., III: Temperature effect on physical and chemical properties of secondary organic aerosol from m-xylene photooxidation, Atmos. Chem. Phys., 10, 3847-3854, 10.5194/acp-10-3847-2010, 2010. Takekawa, H., Minoura, H. and Yamazaki, S.: Temperature dependence of secondary organic aerosol formation by photo-oxidation of hydrocarbons, Atmos. Environ., 37, 3413-3424, 10.1016/s1352-2310(03)00359-5, 2003.

11. page 7 line 25: A magnified view of the mass spectra shown in Fig. 3 would be more useful for comparison purposes.

Taking the reviewer's suggestion, we have magnified the view of the mass spectra shown in Fig. 3.

Figure 3. Selected background-subtraction HESI-Q Exactive-Orbitrap MS results of SOA in both positive and negative ion modes from the photooxidation of m-xylene-OH under both low and high RH conditions (Note that the Y-axis scales for low and high RH are largely different, 106 at low RH and 105 at high RH).

12. Page 7 line 27: The sentence "It should be pointed out that the signal intensities..." is confusing. Were the mass spectra for the different experiments obtained using different MS operation conditions (e.g., ESI spray conditions, MS collision gas)?

The sentence pointed out by the reviewer is indeed confusing and thus we have deleted this sentence in the text. The mass spectra for the different experiments were obtained using exactly same MS operation conditions. Thus, the mass spectra for different experiments were comparable.

13. Page 7 line 25 to page 8 line 11: The mass peaks discussed here do not seem to be the major peaks shown in Fig. 3. Why did the authors choose to focus their discussion only on these selected peaks? The major peaks seem to be m/z > 200. How were these products formed? The authors should include a list of all the product ions identified. Do these identified products match their proposed reaction mechanism

show in Scheme 1?

The mass peaks discussed between Page 7 line 25 to page 8 line 11 are the most abundant peaks in Fig 3, so we gave the proposed structures and discussed here. The m/z > 200 peaks are not discussed in this paragraph, but we discussed the m/z > 200 peaks and explained how these products formed in the Sec 3.4. In addition, taking the reviewer's advice, we have added a list of all the product ions identified in Table S1 in the supplementary information. These identified products that match their proposed reaction mechanism show in Scheme 1 are marked in Scheme 1.

Table S1(a). List of all the SOA product ions identified from ESI-HRMS in positive mode. Low RH High RH Formula m/z intensity m/z intensity 415.12032 1.55E+05 415.11937 1.26E+04 C18 H23 O11 413.10459 1.18E+05 413.10393 5.79E+03 C18 H21 O11 399.12542 1.30E+05 399.12470 2.03E+04 C18 H23 O10 397.10976 2.18E+05 397.10926 1.95E+04 C18 H21 O10 381.11482 2.42E+05 381.11434 2.92E+04 C18 H21 O9 379.09902 1.09E+05 379.09866 7.43E+03 C18 H19 O9 365.11992 1.17E+05 365.11944 3.41E+04 C18 H21 O8 385.10957 1.03E+05 385.10901 9.72E+03 C17 H21 O10 383.09399 1.01E+05 383.09330 6.04E+03 C17 H19 O10 369.11481 1.06E+05 369.11446 1.57E+04 C17 H21 O9 367.09911 1.15E+05 367.09892 - C17 H19 O9 355.09919 1.88E+05 355.09870 2.72E+04 C16 H19 O9 353.08352 1.31E+05 353.08319 1.08E+04 C16 H17 O9 339.10481 1.27E+05 339.10408 1.72E+04 C16 H19 O8 337.08857 1.62E+05 337.08800 2.34E+04 C16 H17 O8 321.09386 1.33E+05 321.09328 1.11E+04 C16 H17 O7 341.08370 2.11E+05 341.08322 2.89E+04 C15 H17 O9 325.08882 1.54E+05 325.08817 2.30E+04 C15 H17 O8 323.07306 1.22E+05 323.07263 1.49E+04 C15 H15 O8 309.09403 9.96E+04 309.09236 - C15 H17 O7 307.07817 1.31E+05 307.08028 - C15 H15 O7 329.08383 1.01E+05 329.08320 7.12E+03 C14 H17 O9 327.06797 1.08E+05 327.06751 6.51E+03 C14 H15 O9 313.08917 1.08E+05 313.08830 1.05E+04 C14 H17 O8 311.07309 1.62E+05 311.07260 1.15E+04 C14 H15 O8 297.09553 9.65E+04 297.09353 9.68E+03 C14 H17 O7 295.07852 1.11E+05 295.07782 1.29E+04 C14

H15 O7 281.10079 8.07E+04 281.09867 - C14 H17 O6 299.07316 1.49E+05 299.07245 1.29E+04 C13 H15 O8 297.05750 9.95E+04 297.05697 7.39E+03 C13 H13 O8 283.07818 1.43E+05 283.07558 - C13 H15 O7 281.06267 1.37E+05 281.06220 1.19E+04 C13 H13 O7 267.08471 1.06E+05 267.08299 - C13 H15 O6 265.06795 9.57E+04 265.06732 1.06E+04 C13 H13 O6 287.07312 1.36E+05 287.07266 - C12 H15 O8 285.05754 2.03E+05 285.05710 1.41E+04 C12 H13 O8 271.07849 1.33E+05 271.07786 2.30E+04 C12 H15 O7 269.06262 3.57E+05 269.06157 - C12 H13 O7 267.04693 1.32E+05 267.04613 9.73E+03 C12 H11 O7 265.02894 1.92E+05 265.02852 4.24E+04 C12 H9 O7 263.01347 1.17E+05 263.01304 2.73E+04 C12 H7 O7 255.08462 8.73E+04 255.08340 - C12 H15 O6 253.06787 1.75E+05 253.06731 - C12 H13 O6 251.05217 1.70E+05 251.05172 - C12 H11 O6 249.03409 1.97E+05 249.03366 4.15E+04 C12 H9 O6 231.02374 1.62E+05 231.02326 2.38E+04 C12 H7 O5 275.07311 1.97E+05 275.07260 - C11 H15 O8 273.05759 3.28E+05 273.05718 1.18E+04 C11 H13 O8 271.04182 1.54E+05 271.04129 6.63E+03 C11 H11 O8 259.07844 2.48E+05 259.07786 3.72E+04 C11 H15 O7 257.06264 3.15E+05 257.06224 - C11 H13 O7 255.04708 1.51E+05 255.04659 - C11 H11 O7 243.08360 2.76E+05 243.08304 5.15E+04 C11 H15 O6 239.05231 1.71E+05 239.05191 - C11 H11 O6 235.01848 9.71E+04 235.01853 1.38E+04 C11 H7 O6 225.07285 1.87E+05 225.07253 - C11 H13 O5 223.05754 8.06E+04 223.05702 - C11 H11 O5 261.05771 2.51E+05 261.05727 2.13E+04 C10 H13 O8 259.04210 1.99E+05 259.04157 8.26E+03 C10 H11 O8 245.06288 1.36E+06 245.06243 1.74E+05 C10 H13 O7 243.04720 2.19E+06 243.04677 1.46E+05 C10 H11 O7 241.03138 7.38E+05 241.02850 - C10 H9 O7 229.06802 1.21E+06 229.06751 - C10 H13 O6 227.05224 3.10E+06 227.05182 3.34E+05 C10 H11 O6 225.03667 1.16E+06 225.03624 6.95E+04 C10 H9 O6 223.01851 2.82E+05 223.01812 1.69E+04 C10 H7 O6 213.07305 1.30E+05 213.07248 - C10 H13 O5 211.05741 2.01E+06 211.05702 3.14E+05 C10 H11 O5 209.04177 3.40E+06 209.04141 8.84E+04 C10 H9 O5 193.04693 3.66E+05 193.04652 - C10 H9 O4 181.08606 5.51E+04 181.08581 - C10 H13 O3 179.07026 6.43E+04 179.06998 - C10 H11 O3 231.04717 2.73E+05

231.04688 1.32E+04 C9 H11 O7 229.03161 1.01E+05 229.03122 - C9 H9 O7 215.05216 5.96E+05 215.05182 - C9 H11 O6 213.03667 9.22E+05 213.03617 - C9 H9 O6 199.05730 3.73E+05 199.05692 - C9 H11 O5 197.04179 4.02E+05 197.04137 - C9 H9 O5 195.02624 1.68E+05 195.02591 - C9 H7 O5 193.00815 3.16E+05 193.00767 1.88E+04 C9 H5 O5 185.08084 1.38E+05 185.07750 - C9 H13 O4 183.06534 1.05E+05 183.06212 - C9 H11 O4 179.03120 8.18E+04 179.03112 - C9 H7 O4 167.07030 1.20E+05 167.06986 - C9 H11 O3 147.05008 7.10E+04 147.04975 - C9 H7 O2 201.03665 7.21E+05 201.03628 - C8 H9 O6 189.07567 1.76E+05 189.07240 - C8 H13 O5 187.06003 1.11E+06 187.05678 - C8 H11 O5 185.04165 3.88E+05 185.04139 - C8 H9 O5 183.02618 2.25E+05 183.02609 - C8 H7 O5 171.06509 1.03E+06 171.06488 - C8 H11 O4 169.04959 5.29E+05 169.04638 - C8 H9 O4 167.03115 3.59E+05 167.03077 - C8 H7 O4 155.07013 1.02E+06 155.06985 - C8 H11 O3 153.05453 7.56E+05 153.05425 - C8 H9 O3 151.03891 1.33E+05 151.03874 4.89E+03 C8 H7 O3 137.05962 1.70E+06 137.05931 - C8 H9 O2 219.01631 2.01E+05 219.01588 - C7 H7 O8 189.03670 1.66E+05 189.03673 - C7 H9 O6 171.03328 1.50E+06 171.03295 - C7 H7 O5 171.02640 2.77E+05 171.02571 - C7 H7 O5 157.04949 2.03E+05 157.04906 - C7 H9 O4 155.03117 8.61E+04 155.03096 - C7 H7 O4 141.05445 5.63E+05 141.05420 - C7 H9 O3 139.03886 8.46E+04 139.03859 - C7 H7 O3 125.05974 1.68E+06 125.05950 - C7 H9 O2 123.04411 1.34E+05 123.04389 - C7 H7 O2 109.06503 2.41E+05 109.06480 - C7 H9 O 209.02861 6.79E+04 209.02812 - C6 H9 O8 175.03264 1.17E+05 175.03264 - C6 H7 O6 175.02542 9.74E+05 175.02510 - C6 H7 O6 157.01758 8.63E+04 157.01737 - C6 H5 O5 143.03380 9.78E+04 143.03354 - C6 H7 O4 127.03897 4.85E+05 127.03868 - C6 H7 O3 113.05988 4.34E+05 113.05967 - C6 H9 O2 111.04425 1.26E+06 111.04405 - C6 H7 O2 215.03908 9.91E+04 215.03895 - C5 H11 O9 199.04426 7.18E+04 199.04416 - C5 H11 O8 115.03911 2.47E+05 115.03889 - C5 H7 O3

Table S1(b). List of all the SOA product ions identified from ESI-HRMS in negative mode. Low RH High RH Formula m/z Intensity m/z Intensity 309.17388 8.22E+04 309.17346 - C17 H25 O5 427.02033 1.43E+05 427.01930 3.35E+02 C16 H11

O14 407.11955 1.04E+05 407.11842 2.19E+02 C16 H23 O12 405.10387 1.04E+05 405.10320 2.63E+02 C16 H21 O12 391.12466 1.12E+05 391.12383 2.93E+02 C16 H23 O11 389.10906 1.96E+05 389.10828 1.48E+03 C16 H21 O11 387.09325 1.09E+05 387.09280 5.09E+02 C16 H19 O11 373.11399 1.97E+05 373.11329 1.26E+03 C16 H21 O10 371.09821 1.44E+05 371.09766 6.70E+02 C16 H19 O10 357.11903 1.69E+05 357.11856 9.95E+02 C16 H21 O9 355.10338 1.80E+05 355.10274 1.62E+03 C16 H19 O9 359.09835 1.67E+05 359.09778 1.15E+03 C15 H19 O10 343.10330 1.50E+05 343.10278 1.01E+03 C15 H19 O9 341.08758 1.21E+05 341.08664 5.45E+02 C15 H17 O9 339.20000 2.00E+05 339.19922 - C15 H31 O8 327.10844 1.12E+05 327.10799 5.27E+02 C15 H19 O8 325.09285 1.08E+05 325.09217 6.23E+02 C15 H17 O8 265.14792 3.33E+05 265.14776 - C15 H21 O4 218.03824 2.71E+05 218.03779 5.27E+03 C15 H6 O2 363.09335 1.13E+05 363.09077 9.01E+02 C14 H19 O11 347.09836 2.51E+05 347.09758 2.67E+03 C14 H19 O10 345.08263 1.80E+05 345.08202 1.35E+03 C14 H17 O10 331.10347 1.83E+05 331.10283 2.34E+03 C14 H19 O9 329.08781 2.19E+05 329.08696 1.87E+03 C14 H17 O9 327.07190 1.29E+05 327.07119 9.89E+02 C14 H15 O9 325.18438 3.85E+05 325.18366 - C14 H29 O8 313.09287 1.96E+05 313.09204 1.88E+03 C14 H17 O8 311.07715 1.52E+05 311.07670 1.01E+03 C14 H15 O8 297.09786 1.32E+05 297.09724 1.05E+03 C14 H17 O7 295.08212 1.29E+05 295.08163 9.81E+02 C14 H15 O7 333.08273 1.42E+05 333.08206 1.23E+03 C13 H17 O10 331.06692 9.75E+04 331.06613 7.28E+02 C13 H15 O10 317.08774 2.67E+05 317.08714 4.21E+03 C13 H17 O9 315.07210 1.77E+05 315.07401 - C13 H15 O9 311.16878 7.80E+05 311.16806 - C13 H27 O8 301.09273 1.82E+05 301.09215 2.77E+03 C13 H17 O8 299.07727 2.15E+05 299.07641 3.44E+03 C13 H15 O8 297.06154 1.09E+05 297.06002 1.42E+03 C13 H13 O8 285.09789 1.00E+05 285.09726 1.77E+03 C13 H17 O7 283.08221 2.00E+05 283.08162 2.54E+03 C13 H15 O7 281.06697 1.22E+05 281.06609 - C13 H13 O7 267.08726 1.26E+05 267.08657 2.19E+03 C13 H15 O6 265.07197 1.61E+05 265.07106 1.19E+03 C13 H13 O6 247.06280 1.81E+05 247.06092 3.33E+02 C13 H11 O5 231.06771 3.10E+05

231.06592 - C13 H11 O4 303.07184 1.82E+05 303.07152 3.35E+03 C12 H15 O9 301.05629 9.34E+04 301.05576 1.42E+03 C12 H13 O9 297.15292 3.35E+05 297.15230 - C12 H25 O8 287.07698 2.52E+05 287.07691 3.35E+03 C12 H15 O8 285.06133 1.57E+05 285.06109 2.27E+03 C12 H13 O8 271.08217 1.69E+05 271.08173 3.03E+03 C12 H15 O7 269.06606 2.53E+05 269.06611 2.03E+03 C12 H13 O7 267.05024 9.81E+04 267.05010 7.33E+02 C12 H11 O7 255.08719 1.17E+05 255.08664 - C12 H15 O6 253.07111 1.91E+05 253.07089 - C12 H13 O6 251.05515 1.24E+05 251.05505 5.59E+02 C12 H11 O6 237.07648 1.36E+05 237.07600 - C12 H13 O5 221.08136 8.88E+04 221.08108 - C12 H13 O4 291.07203 1.18E+05 291.07139 1.57E+03 C11 H15 O9 289.05636 9.61E+04 289.05566 2.67E+03 C11 H13 O9 275.07729 1.70E+05 275.07673 3.31E+03 C11 H15 O8 273.06161 2.00E+05 273.06105 4.47E+03 C11 H13 O8 259.08219 1.57E+05 259.08171 3.21E+03 C11 H15 O7 257.06651 2.28E+05 257.06598 4.31E+03 C11 H13 O7 255.05088 1.07E+05 255.05025 2.48E+03 C11 H11 O7 243.08714 8.57E+04 243.08664 - C11 H15 O6 241.07147 2.04E+05 241.07093 - C11 H13 O6 225.07641 1.52E+05 225.07596 - C11 H13 O5 223.06071 1.40E+05 223.06018 - C11 H11 O5 209.08133 1.02E+05 209.08102 - C11 H13 O4 207.06573 1.11E+05 207.06531 - C11 H11 O4 193.08630 7.02E+04 193.08600 - C11 H13 O3 261.06155 2.09E+05 261.05977 - C10 H13 O8 259.04588 9.74E+04 259.04511 3.26E+03 C10 H11 O8 245.06644 2.22E+05 245.06586 - C10 H13 O7 243.05084 2.08E+05 243.05037 - C10 H11 O7 229.07132 2.03E+05 229.07087 - C10 H13 O6 227.05570 2.51E+05 227.05525 - C10 H11 O6 211.06064 1.96E+05 211.06027 - C10 H11 O5 209.04512 9.50E+04 209.04452 - C10 H9 O5 195.06563 1.61E+05 195.06536 - C10 H11 O4 193.05002 8.88E+04 193.04978 - C10 H9 O4 181.08628 9.12E+04 181.08592 - C10 H13 O3 163.07558 5.18E+04 163.07520 - C10 H11 O2 249.06096 1.71E+05 249.06092 2.65E+03 C9 H13 O8 247.04556 1.43E+05 247.04500 3.44E+03 C9 H11 O8 233.06614 1.99E+05 233.06580 4.43E+03 C9 H13 O7 231.05066 1.62E+05 231.05023 - C9 H11 O7 229.03504 7.84E+04 229.03443 - C9 H9 O7 217.07129 1.69E+05 217.07080 - C9 H13 O6 215.05569 2.04E+05 215.05511 - C9 H11 O6 201.07618 2.38E+05

201.07580 - C9 H13 O5 199.06058 1.95E+05 199.06031 - C9 H11 O5 197.04500 1.59E+05 197.04464 - C9 H9 O5 183.06553 1.70E+05 183.06531 - C9 H11 O4 181.04989 1.64E+05 181.04952 - C9 H9 O4 165.05484 1.21E+05 165.05454 - C9 H9 O3 149.05981 6.31E+04 149.05947 - C9 H9 O2 237.06126 9.92E+04 237.06075 2.61E+03 C8 H13 O8 235.04566 2.21E+05 235.04511 2.88E+03 C8 H11 O8 233.03005 1.23E+05 233.02891 - C8 H9 O8 221.06623 4.82E+05 221.06579 8.46E+03 C8 H13 O7 219.05057 3.05E+06 219.05011 5.02E+04 C8 H11 O7 217.03488 3.13E+06 217.03444 7.13E+04 C8 H9 O7 205.07117 2.33E+05 205.07077 4.95E+03 C8 H13 O6 203.05550 1.74E+06 203.05513 2.87E+04 C8 H11 O6 201.03983 2.10E+06 201.03967 3.18E+04 C8 H9 O6 199.02428 2.08E+05 199.02409 - C8 H7 O6 187.06049 1.68E+06 187.06013 - C8 H11 O5 185.04483 5.49E+06 185.04449 - C8 H9 O5 183.02923 4.41E+05 183.02901 - C8 H7 O5 171.06543 5.43E+05 171.06511 1.23E+05 C8 H11 O4 169.04976 3.79E+06 169.04954 - C8 H9 O4 167.03412 7.36E+05 167.03368 - C8 H7 O4 153.05472 5.91E+05 153.05442 - C8 H9 O3 151.03908 4.09E+05 151.03878 - C8 H7 O3 137.05970 1.18E+05 137.05943 - C8 H9 O2 205.03496 2.28E+05 205.03286 - C7 H9 O7 191.05540 1.23E+06 191.05504 1.88E+04 C7 H11 O6 189.03975 2.04E+06 189.03942 2.91E+04 C7 H9 O6 187.02422 1.80E+05 187.02389 - C7 H7 O6 175.06041 1.80E+05 175.06006 - C7 H11 O5 173.04471 6.71E+05 173.04440 - C7 H9 O5 171.02908 6.75E+05 171.02879 - C7 H7 O5 169.01357 1.20E+05 169.01317 - C7 H5 O5 157.04965 9.08E+05 157.04939 - C7 H9 O4 155.03403 1.23E+06 155.03374 - C7 H7 O4 153.01828 2.32E+05 153.01805 - C7 H5 O4 141.05463 2.11E+06 141.05439 - C7 H9 O3 139.03897 1.25E+06 139.03869 - C7 H7 O3 125.05961 6.55E+05 125.05940 - C7 H9 O2 123.04397 1.33E+06 123.04376 - C7 H7 O2 229.05210 1.79E+05 229.04953 - C6 H13 O9 213.05676 6.20E+04 213.05473 - C6 H13 O8 191.01907 1.26E+05 191.01873 7.54E+03 C6 H7 O7 177.03967 2.86E+06 177.03930 4.45E+04 C6 H9 O6 175.02402 1.43E+06 175.02363 4.04E+04 C6 H7 O6 173.00836 2.54E+05 173.00529 - C6 H5 O6 161.04464 1.30E+06 161.04430 - C6 H9 O5 157.01330 3.67E+05 157.01334 - C6 H5 O5 147.06522 8.02E+04 147.06492 - C6 H11 O4 145.04957 3.71E+05

145.04929 - C6 H9 O4 143.03391 1.54E+06 143.03365 - C6 H7 O4 141.01827 7.79E+05 141.01797 - C6 H5 O4 139.00264 6.12E+04 139.00224 - C6 H3 O4 127.03890 1.83E+06 127.03866 - C6 H7 O3 125.02325 9.93E+05 125.02297 - C6 H5 O3 113.05952 7.21E+05 113.05925 - C6 H9 O2 111.04388 7.15E+05 111.04361 - C6 H7 O2 109.02824 2.35E+05 109.02805 - C6 H5 O2 201.05694 1.09E+05 201.05479 - C5 H13 O8 163.02399 7.06E+04 163.02364 1.05E+04 C5 H7 O6 147.02887 2.97E+06 147.02878 4.63E+04 C5 H7 O5 145.01320 4.46E+05 145.01289 - C5 H5 O5 131.03383 8.78E+05 131.03360 - C5 H7 O4 129.01819 2.07E+06 129.01793 - C5 H5 O4 127.00253 1.97E+05 127.00232 - C5 H3 O4 115.03882 1.08E+06 115.03858 - C5 H7 O3 113.02311 2.36E+06 113.02278 - C5 H5 O3 111.00750 4.41E+05 111.00726 - C5 H3 O3 133.01316 1.69E+05 133.01282 - C4 H5 O5 119.03381 1.48E+05 119.03355 - C4 H7 O4 117.01807 3.64E+05 117.01775 - C4 H5 O4 115.00245 3.19E+05 115.00215 - C4 H3 O4 101.02308 1.11E+06 101.02289 - C4 H5 O3

14: General comment: What compounds are the -ve MS mode sensitive to? Were these compounds identified in their collected mass spectra?

In positive mode analysis, ions are produced by protonation. Thus, groups that more readily accept a positive charge, such as carbonyls, are often observed in this mode. As listed in Table 3, the proposed compounds obtained by HRMS in positive ion mode are all with the carbanyl group. Negative mode analysis leads to formation of deprotonated ions. Thus, molecules containing functional groups that readily lose a proton, such as carboxylic acids, are frequently observed in this mode. Also, the esters compounds can be obtained in the negative ion mode (Hamilton et al., 2008; Camredon et al., 2010; Ge et al., 2017). In the MCM prediction about m-xylene-OH oxidation, many carbonyls are included. It can be deduced that many carboxylic acids can be formed via OH oxidation of these carbonyls and these carboxylic acids can be measured in the negative ion mode.

Camredon, M., Hamilton, J. F., Alam, M. S., Wyche, K. P., Carr, T., White, I. R., Monks, P. S., Rickard, A. R., and Bloss, W. J.: Distribution of gaseous and particulate organic composition during dark $\alpha$-pinene ozonolysis, Atmos. Chem. Phys., 10, 2893-2917, 10.5194/acp-10-2893-2010, 2010. Ge, S., Xu, Y. and Jia, L.: Secondary organic aerosol formation from propylene irradiations in a chamber study, Atmos. Environ., 157, 146-155, 10.1016/j.atmosenv.2017.03.019, 2017. Hamilton, J. F., Lewis, A. C., Carey, T. J., and Wenger, J. C.: Characterization of polar compounds and oligomers in secondary organic aerosol using liquid chromatography coupled to mass spectrometry, Anal. Chem., 80, 474-480, 10.1021/ac701852t, 2008.

15. General comment: The authors mentioned in the experimental system that they used a HPLC-MS system in their study. It is not clear from their presented results whether this was the case. Was HPLC not used to separate the products via their volatilities prior to MS analysis?

HPLC was used in our experiments as the injection system before HRMS analysis. We used the high resolution of mass analyzer for the separation of major SOA components instead of HPLC.

16. Page 9 line 30: The authors claimed that they used the distribution of relative intensity of SOA products with the same carbon number to investigate the potential RH effect on HOMs. The rationale behind this course of action seems to contradict their previous statement in Page 7 line 27 that signal intensities can be biased by ionization properties.

The statement was incorrect and confusing in Page 7 Line 27 and we have deleted it from the text. The mass spectra for the different experiments were obtained using same MS operation conditions. Thus, the mass spectra for different experiments were comparable (see the reply of Comment 12).

17. Scheme 1: The authors should indicate explicitly in Scheme 1 which are the products that they have identified.

Taking the reviewer's advice, we have modified Scheme 1 in which the products identified are marked with a molecular weight number below the molecular formula.

Scheme 1. The route of OH-initiated m-xylene oxidation. The red number below the molecular formula is its molecular weight, which is determined by HRMS to exist in the particle phase.

18. Page 10 line 27: The sentence "Together with the previous study on toluene SOA, it is conceivable that the effect of RH on SOA yield is a common feature of SOA formation from oxidation of all OH-initiated aromatics" is too generalized and needs to be rephrased. As discussed by the authors in their introduction, an increase RH does not necessarily cause a decrease in SOA mass yields in aromatics SOA systems. Other factors such as NOx can also alter the effect that RH has on SOA mass yields in these systems.

Taking the reviewer's advice, we have rephrased the sentence in Page 10 line 27.

Together with the previous study on toluene SOA, it is conceivable that the effect of RH on SOA yield is a common feature of SOA formation from aromatics oxidation under low NOx conditions and using H2O2 as the OH radical source.

Please also note the supplement to this comment:
https://www.atmos-chem-phys-discuss.net/acp-2019-20/acp-2019-20-AC1-supplement.pdf

———————————————————

---

## Author Comment (AC2) · 25 May 2019

Please also see the attached file of XU-Response-RC2 as acp-2019-20-supplement.pdf because marked colors, figures and tables are missed in this plain text.

Response to Reviewer 2

We greatly appreciate the time and effort that reviewer 2 spent in reviewing our manuscript. The comments are really thoughtful and helpful to improve the quality of our paper. Reviewer 2 has provided both main comments and other specific com-

ments. Below we make a point-by-point response to these comments. According to editor's requirement, the response to the reviewer 2 is structured in the following sequence: (1) comments from the reviewer in black color, (2) our response in blue color, and (3) our changes in the revised manuscript in red color.

Overview: This study explore the role of relative humidity (RH) on the m-xylene SOA formation under OH initiated no NOx condition. The results showed that the SOA yield under high RH is significantly lower than that under low RH conditions. This study provides SOA yields and particle-phase SOA products under different RH levels. The LWC was measured by the particles mass deduction in the DAASS. The authors measured the SOA compositions by using a Fourier transform infrared (FTIR) spectra and ultrahigh performance liquid chromatograph electrospray ionization-high-resolution mass spectrometer (UPLC-ESI-HRMS). The authors reported that SOA yield found to be about 7 times high in dry condition (RH_13%) than that in wet condition (RH_75%). Overall, the experimental data to show the impact of RH on SOA yields and products and the conclusion originating from the chamber are doubtful. The small chamber used in this study will be significantly influenced by the gas-wall processes of organic species increasing the uncertainty in data and interpretation of results. This paper in its current form is not acceptable. Please find the comments below.

The chamber volume for our current experiments was around 1 m3. All chambers have the wall losses of species. Though a larger volume reactor may minimize these effects, the fans are usually equipped inside this kind of reactor to make the heat generated by lights homogeneously mixed, which counteracts the decrease of wall effect by the larger volume (Carter et al., 2005; Cocker et al., 2001). In addition, relatively small reactors in the range of 0.2-3 m3 are also ubiquitously used in smog chamber studies (Chen et al., 2017; Chu et al., 2016; Díaz-de-Mera et al., 2017; Huang et al., 2017; Peng et al., 2017; Schnitzler et al., 2014; Ye et al., 2018). After careful analysis of our experiments, we believe that our results are reliable and credible. Below are the specific replies to the comments.

Carter, W., Cockeriii, D., Fitz, D., Malkina, I., Bumiller, K., Sauer, C., Pisano, J., Bufalino, C., and Song, C.: A new environmental chamber for evaluation of gas-phase chemical mechanisms and secondary aerosol formation, Atmos. Environ., 39, 7768-7788, 10.1016/j.atmosenv.2005.08.040, 2005. Chen, L., Bao, K., Li, K., Lv, B., Bao, Z., Lin, C., Wu, X., Zheng, C., Gao, X., and Cen, K.: Ozone and secondary organic aerosol formation of toluene/NOx irradiations under complex pollution scenarios, Aerosol Air Qual. Res., 17, 1760-1771, 10.4209/aaqr.2017.05.0179, 2017. Chu, B., Zhang, X., Liu, Y., He, H., Sun, Y., Jiang, J., Li, J., and Hao, J.: Synergetic formation of secondary inorganic and organic aerosol: effect of SO2 and NH3 on particle formation and growth, Atmos. Chem. Phys., 16, 14219-14230, 10.5194/acp-16-14219-2016, 2016. Cocker, D. R., 3rd, Flagan, R. C., and Seinfeld, J. H.: State-of-the-art chamber facility for studying atmospheric aerosol chemistry, Environ. Sci. Technol., 35, 2594-2601, 10.1021/es0019169, 2001. Díaz-de-Mera, Y., Aranda, A., Martínez, E., Rodríguez, A. A., Rodríguez, D., and Rodríguez, A.: Formation of secondary aerosols from the ozonolysis of styrene: Effect of SO2 and H2O, Atmos. Environ., 171, 25-31, 10.1016/j.atmosenv.2017.10.011, 2017. Huang, M., Hao, L., Cai, S., Gu, X., Zhang, W., Hu, C., Wang, Z., Fang, L., and Zhang, W.: Effects of inorganic seed aerosols on the particulate products of aged 1,3,5-trimethylbenzene secondary organic aerosol, Atmos. Environ., 152, 490-502, 10.1016/j.atmosenv.2017.01.010, 2017. Peng, J., Hu, M., Du, Z., Wang, Y., Zheng, J., Zhang, W., Yang, Y., Qin, Y., Zheng, R., Xiao, Y., Wu, Y., Lu, S., Wu, Z., Guo, S., Mao, H., and Shuai, S.: Gasoline aromatics: a critical determinant of urban secondary organic aerosol formation, Atmos. Chem. Phys., 17, 10743-10752, 10.5194/acp-17-10743-2017, 2017. Schnitzler, E. G., Dutt, A., Charbonneau, A. M., Olfert, J. S., and Jaeger, W.: Soot aggregate restructuring due to coatings of secondary organic aerosol derived from aromatic precursors, Environ. Sci. Technol., 48, 14309-14316, 10.1021/es503699b, 2014. Ye, J., Abbatt, J. P. D., and Chan, A. W. H.: Novel pathway of SO2 oxidation in the atmosphere: reactions with monoterpene ozonolysis intermediates and secondary organic aerosol, Atmos. Chem. Phys., 18, 5549-5565, 10.5194/acp-18-5549-2018, 2018.

Major Comments:

1. The aromatics VOCs are gas pollutants that is found to be high in urban environments where the NOx is also abundant. It is unclear why the authors chose no NOx condition to study the humidity effects on the formation of xylene SOA. Clarify this.

It is true that aromatics are found to be high in urban environments where NOx is also abundant. Real environment is relatively complicated. Nevertheless, the purpose of our study is to investigate the RH effect on the SOA formation from m-xylene only oxidized by the OH radicals and not interpreted by other factors. The NOx can complicate the aromatics oxidation system, since NOx conditions can provide OH and NO3 radicals, and NOx themselves can also participate in the oxidation reactions.

2. What is the effect of the wall on the loss gaseous H2O2? H2O2 is very hydrophilic and sticky to the wall. When RH is high, the water on the chamber wall becomes high forming a water film. This wet film can absorb a large amount of H2O2 and modulate the concentration of OH radicals. Please clarify how the authors measured OH radical concentrations under varying RH conditions. Why did the author use 20 ppm of H2O2 which was 40 times higher than the m-xylene concentration? What is the photolysis rate constant of H2O2 in the chamber?

We agree with the reviewer that H2O2 is very hydrophilic. According to the reviewer's comment, we estimated the wall loss of H2O2 at low and high RH at 299 K for 4 h in our study using the O3 analyzer (49C, Thermo Environmental Instruments Inc.), since H2O2 can also absorb the light at 254 nm. Thus, the output by O3 analyzer can basically represent the relative change in H2O2 concentration though it is not the real H2O2 concentration. The results for H2O2 wall loss experiments show that the numbers outputted by O3 analyzer were in the range of 22.2-23.2 at low RH (8%) and in the range of 22.1-23.3 at high RH (75%) throughout each H2O2 wall loss experiment, respectively. Results indicate that there is no significant H2O2 wall loss throughout the experiments and no obvious difference between both RHs. In other words, H2O2

[Figure]

concentrations in the chamber were roughly constant during the experiment. In this study there was no equipment for the measurement of OH radical concentrations, so we cannot directly obtain the OH radical concentration during the experiments. But it can be convinced that the OH concentrations were consistent at varying RHs, which can be realized from the similar change in concentrations of the reacted m-xylene (Fig. S1) (see below) that is added in the supplementary information according to the Comment 6 of the Reviewer 1. In addition, the MCM simulation was conducted to obtain the OH concentration of $1.6 \times 10^{-4}$ ppb, for which a photolysis rate constant of $7.56 \times 10^{-6}$ min-1 for $H_2O_2$ was used in the chamber.

Fig. S1. Reaction time profiles of m-xylene measured by the GC-MS that accompanied the observed SOA growth for the four experiments

3. The size of chamber used in this study was one cubic meter and relatively very small. Thus, the wall of chemical species is very high. Additionally, the loss of chemical species to the chamber is very sensitive to humidity. The impact of RH on SOA yields can be very uncertain and doubtful. The reduction of SOA yields at the high humidity is more likely due to the chemical loss to the wet chamber wall. Thus, the conclusion made by the authors could be incorrect. Hydrophilic products and reactive chemical species can more deposit to the wall at high humidity.

We agree with the reviewer that the reactor volume in this study is relatively small. As we mentioned in the beginning of reply to Reviewer 2, in recent years small reactors in the range of (0.2-3) m3 are ubiquitously used in smog chamber studies. Indeed, small chambers have the wall losses of species, but wall effects also exist in big reactors, since big reactor is needed to be equipped with fans inside the reactor to make the heat generated by lights homogeneously mixed. For small reactors, the particle wall loss of $5.87 \times 10^{-5}$ s-1 was measured in a 3 m3 reactor (Chen et al., 2017) and (3.21-5.57) $\times 10^{-5}$ s-1 was measured in a 2 m3 reactor (Chu et al., 2016). For big reactors, the particle wall loss of $8 \times 10^{-5}$ s-1 was measured in a 90 m3 reactor (Carter et al., 2005) and (2.5-5.0) $\times 10^{-5}$ s-1 was measured in the dual 28 m3 (Cocker et al.,

2001). It is obvious that the particle wall loss from different researchers is different, but there is no clear relation between particle wall loss and reactor volume for the reactor volume of over 1 m3. It can be speculated that wall effects for other chemical species are probably similar between small and big reactors. In addition, we agree with the reviewer that the loss of chemical species to the wall is sensitive to humidity. To clarify this, we have added a paragraph at the end of Sec. 3.1 following the newly added paragraph according to the Comment 2 of Review 1 in the revised manuscript to clarify the possible underestimation of SOA mass and the reliability of our study.

The wall loss of chemical species that is sensitive to humidity may affect the RH effect on SOA yields, as the reduction of SOA yields at the high humidity may be due to the chemical loss to the wet chamber wall. To estimate the extent of how much the wall loss of chemical species affects the SOA formation at different RHs, we take glyoxal and acetone as reference compounds. Glyoxal, a typical compound that can form SOA, can easily dissolve in the aqueous phase due to the large Henry's law constant of $4.19 \times 105$ M atm-1 (Ip et al., 2009), very sensitive to humidity. Loza et al. (2010) found that the wall loss of glyoxal was minimal at 5% RH, with $kW = 9.6 \times 10\text{-}7$ s-1, whereas kW was $4.7 \times 10\text{-}5$ s-1 at 61% RH. We assume that kW linearly increases with RH, and the kW value is estimated to be $6.1 \times 10\text{-}5$ s-1 at 80% and $7.4 \times 10\text{-}6$ at 13% RH, with the difference being 8.2 times. According to the wall loss of glyoxal, glyoxal only decreased by 10% at the end of our experiment at low RH, while glyoxal decreased by 59% at high RH. Acetone can hardly dissolve in the aqueous phase due to the small Henry's law constant of 29 M atm-1 (Poulain et al., 2010), which is 4 orders of magnitude less than that of glyoxal. Ge et al. (2017) obtained that the wall loss of acetone was $5.0 \times 10\text{-}6$ s-1 at 87% RH and $3.3 \times 10\text{-}6$ s-1 at 5% RH, with a factor of 1.5. The difference of wall loss between glyoxal and acetone at low RH is about 2 times, while it becomes about 12 times at high RH. Thus, it can be considered that the wall loss among different species at low RH is less affected by the Henry's law constant, but it is greatly affected at high RH. In our study glycolaldehyde (See the Sec. 3.3) is found to be an important SOA precursor that can form a large fraction

of oligomers in our experiments, but the wall loss of glycolaldehyde is not available. The Henry's law constant of glycolaldehyde was obtained to be $4.14 \times 10^4$ M atm$^{-1}$ (Betterton and Hoffmann, 1988), an order of magnitude lower than glyoxal, indicating that glycolaldehyde is less sensitive to humidity than glyoxal but much more sensitive to humidity than acetone. Based on the data of these two reference species, the wall loss of glycolaldehyde at low RH is taken to be $5 \times 10^{-6}$ s$^{-1}$, and the difference in wall loss between high and low RHs is about 6 times. Then, the wall loss of glycolaldehyde at high RH can be $3 \times 10^{-5}$ s$^{-1}$. Then, it is estimated that glycolaldehyde would decrease by 7% at low RH and by 35% at high RH at the end of our experiment, respectively. This means that SOA yield would be underestimated by 35% at high RH and by 7% at low RH if glycolaldehyde lost to the wall was completely transformed to SOA. If this wall effect of SOA precursors was taken into consideration, the SOA yields at high (Exp. 3) and low (Exp. 2) RHs would be 3.4% and 15.1%, respectively. Alternatively, the SOA yield at high RH was underestimated to be 42% relative to that at low RH. Even the sensitivity of the wall loss to RH was taken to be 8 times, the SOA yield at high RH would be underestimated to be 62% compared to that at low RH. In fact, there were many different SOA precursors from the m-xylene oxidation system that probably have much smaller Henry's law constant relative to that of glycolaldehyde. Thus, it is concluded that the RH effect on SOA formation from m-xylene oxidation by H2O2 without NOx is negative.

Betterton, E. A. and Hoffmann, M. R.: Henry's law constants of some environmentally important aldehydes, Environ. Sci. Technol., 12, 1415-1418, 10.1021/es00177a004, 1988. Carter, W., Cockeriii, D., Fitz, D., Malkina, I., Bumiller, K., Sauer, C., Pisano, J., Bufalino, C., and Song, C.: A new environmental chamber for evaluation of gas-phase chemical mechanisms and secondary aerosol formation, Atmos. Environ., 39, 7768-7788, 10.1016/j.atmosenv.2005.08.040, 2005. Chen, L., Bao, K., Li, K., Lv, B., Bao, Z., Lin, C., Wu, X., Zheng, C., Gao, X., and Cen, K.: Ozone and secondary organic aerosol formation of toluene/NOx irradiations under complex pollution scenarios, Aerosol Air Qual. Res., 17, 1760-1771, 10.4209/aaqr.2017.05.0179, 2017. Chu, B.,

[Figure]

Zhang, X., Liu, Y., He, H., Sun, Y., Jiang, J., Li, J., and Hao, J.: Synergetic formation of secondary inorganic and organic aerosol: effect of SO2 and NH3 on particle formation and growth, Atmos. Chem. Phys., 16, 14219-14230, 10.5194/acp-16-14219-2016, 2016. Cocker, D. R., 3rd, Flagan, R. C., and Seinfeld, J. H.: State-of-the-art chamber facility for studying atmospheric aerosol chemistry, Environ. Sci. Technol., 35, 2594-2601, 10.1021/es0019169, 2001. Ge, S., Xu, Y., and Jia, L.: Effects of inorganic seeds on secondary organic aerosol formation from photochemical oxidation of acetone in a chamber, Atmos. Environ., 170, 205-215, 10.1016/j.atmosenv.2017.09.036, 2017. Ip, H. S. S., Huang, X. H. H., and Yu, J. Z.: Effective Henry's law constants of glyoxal, glyoxylic acid, and glycolic acid, Geophys. Res. Lett., 36, L01802, 10.1029/2008gl036212, 2009. Loza, C. L., Chan, A. W., Galloway, M. M., Keutsch, F. N., Flagan, R. C., and Seinfeld, J. H.: Characterization of vapor wall loss in laboratory chambers, Environ. Sci. Technol., 13, 5074-5078, 10.1021/es100727v, 2010. Poulain, L., Katrib, Y., Isikli, E., Liu, Y., Wortham, H., Mirabel, P., Le Calve, S., and Monod, A.: In-cloud multiphase behavior of acetone in the troposphere: gas uptake, Henry's law equilibrium and aqueous phase photooxidation, Chemosphere, 81, 312-320, 10.1016/j.chemosphere.2010.07.032, 2010.

4. In order to analyze the chemical compositions in gas and particles phase using a variety of aerosol, a large amount of air volume should be collected. The size of the chamber used in this study was only one cubic. It is hard to believe how the authors analyzed gas and aerosol with the air volume less than one cubic meter. Additionally, the chamber volume was getting small as the experiment progressed. The consumption of the chamber air increased the wall effect. The authors should clarify this problem.

It is true that the chamber volume was getting small as the experiment progressed, but the change was not significant. The measurement of m-xylene concentration was conducted once every 30 min by a one-liter summa canister, so the total sampling volume for m-xylene measurement was only 9 L throughout each experiment. We also monitored the concentrations of ozone and NOx once an hour by sampling 5 min,

so the total sampling volume for ozone and NOx measurement was only 30 L. The SMPS was on-line analysis, for which the flow rate was 0.3 L min-1, and thus the total sampling volume was 72 L. Thus, before the SOA sampling, when the reaction stopped, the total volume for sampling was only around 110 L, approximately 10% of the size of reactor. Thus, the wall effect for gases and particles would not be significantly changed. After the reaction was finished, the DLPI and PILS simultaneously sampled for FTIR and HRMS analysis of chemical compositions of SOA, respectively. The DLPI sampled 150 L and the PILS needed 100 L. Thus, after the experiment was completely finished, the reactor still contained more than 700 L of air.

5. Page 5, line 4. The Master Chemical Mechanism can only provide the gas-phase reaction pathways. The yield of the products in particle phase may not directly connected to the yield of products in gas-phase. How does the author compare gas-phase oxygenated m-xylene products predicted using MCM to the measured particle-phase products from HRMS?

We agree with the reviewer that the MCM only provides the gas-phase reaction pathways. We went through the products in the MCM, and put the structure of these products in the Mass Frontier program, which can simulate the breakage of bonds made by the MS/MS analysis of HRMS. Meanwhile, we found 5 products in the gas phase by MCM those can match the HRMS analysis. These 5 products are considered to likely partition into the particle phase from the gas phase. Though we only find 5 products in the particle phase identified by HRMS that are also predicted in the gas phase by MCM, the MCM prediction can provide the formation pathway of RO2 radicals, which are helpful with the prediction of RO2 autoxidation.

6. Page 5, line 11. The value of the maximum SOA mass in Figure 1 is not consist with the values reported in the text and the Table 1. The value of SOA mass under 73.6% and 79.1% in Figure 1 is about 40 and 10 ug/m3 but the value reported in the text is only 1.9 and 0.8 ug/m3 and the value reported in Table 1 is 15.8 and 7.9 ug/m3.

There are indeed some mistakes in the text and in Fig. 1. Together with the reply to the Comment 3 of Reviewer 1, the SOA mass was re-corrected by the particle wall loss rates. The values of the maximum SOA mass under 73.6% and 79.1% RH are 21.0 and 7.5 ug/m3, respectively. We have modified in Page 5, line 11 in the text and in Fig 1. A sentence has been added in the revised manuscript: "The maximum mass concentrations fitted are 150.3 and 95.5 $\mu$g m-3 at low RHs, whereas they are 21.0 and 7.5 $\mu$g m-3 at high RHs, ..."

Figure 1. SOA mass concentrations as a function of irradiation time (corrected by particle wall loss and subtracted by LWC).

7 It is not clear how much LWC was present at the end of experiments and how much SOA mass was obtained after subtracting the LWC from total aerosol mass. What is the effects of LWC on the SOA formation in this study? The author mention that LWC can explain the positive effect of RH on SOA formation under high NOx condition. What is the difference in LWC between SOA with the high NOx condition and that with the low or no NOx condition?

At the end of experiments LWC volume concentration accounts for 34% and 45% in the Exps. 3-4 of the volume concentrations of wet particles, which are 5.1 and 2.4 $\mu$g m-3, respectively. SOA mass obtained after subtracting the LWC from total aerosol mass were 21.0 and 7.5 ïA∎g/m3 under 73.6% and 79.1% RH, respectively. The LWC has a negative effect on SOA formation in our study. LWC can generally promote SOA formation under high NOx condition, which was reported in many previous studies (Healy et al., 2009; Kamens et al., 2011; Zhou et al., 2011; Jia and Xu, 2014, 2018; Wang et al., 2016), but the recent study by Hinks et al. (2018) indicates the negative effect of RH on SOA. Although we did not conduct any experiments of m-xylene under high NOx condition, in previous studies, the LWC volume concentration was found to be 22% (78% RH, Jia and Xu, 2018) and 17% (85% RH, Prenni et al., 2007) of wet SOA volume concentration from toluene photoxidation under NOx concentrations of 300 ppb. This indicates that LWC is larger under the high NOx condition than under

the low or no NOx condition.

Healy, R. M., Temime, B., Kuprovskyte, K., and Wenger, J. C.: Effect of relative humidity on gas/particle partitioning and aerosol mass yield in the photooxidation of p-xylene, Environ. Sci. Technol., 43, 1884-1889, 10.1021/es802404z, 2009. Hinks, M. L., Montoya-Aguilera, J., Ellison, L., Lin, P., Laskin, A., Laskin, J., Shiraiwa, M., Dabdub, D., and Nizkorodov, S. A.: Effect of relative humidity on the composition of secondary organic aerosol from the oxidation of toluene, Atmos. Chem. Phys., 18, 1643-1652, 10.5194/acp-18-1643-2018, 2018. Jia, L., and Xu, Y.: Effects of relative humidity on ozone and secondary organic aerosol formation from the photooxidation of benzene and ethylbenzene, Aerosol Sci. Technol., 48, 1-12, 10.1080/02786826.2013.847269, 2014. Jia, L., and Xu, Y.: Different roles of water in secondary organic aerosol formation from toluene and isoprene, Atmos. Chem. Phys., 18, 8137-8154, 10.5194/acp-18-8137-2018, 2018. Kamens, R. M., Zhang, H., Chen, E. H., Zhou, Y., Parikh, H. M., Wilson, R. L., Galloway, K. E., and Rosen, E. P.: Secondary organic aerosol formation from toluene in an atmospheric hydrocarbon mixture: Water and particle seed effects, Atmos. Environ., 45, 2324-2334, 10.1016/j.atmosenv.2010.11.007, 2011. Prenni, A. J., Petters, M. D., Kreidenweis, S. M., DeMott, P. J., and Ziemann, P. J.: Cloud droplet activation of secondary organic aerosol, J. Geophys. Res., 112, D10223, 10.1029/2006JD007963, 2007. Wang, Y., Luo, H., Jia, L., and Ge, S.: Effect of particle water on ozone and secondary organic aerosol formation from benzene-NO2-NaCl irradiations, Atmos. Environ., 140, 386-394, 10.1016/j.atmosenv.2016.06.022, 2016. Zhou, Y., Zhang, H., Parikh, H. M., Chen, E. H., Rattanavaraha, W., Rosen, E. P., Wang, W., and Kamens, R. M.: Secondary organic aerosol formation from xylenes and mixtures of toluene and xylenes in an atmospheric urban hydrocarbon mixture: Water and particle seed effects (II), Atmos. Environ., 45, 3882-3890, 10.1016/j.atmosenv.2010.12.048, 2011.

8. What is the particle size distribution of m-xylene SOA? Does all of the particle size smaller than 1000 nm and within the SMPS measurement range?

According to the reviewer's comment, we have added the particle size distribution of m-xylene SOA for the four experiments in the supplementary information (see Fig. S2). As shown in Fig. S2, all of the particle size is smaller than 1000 nm and within the SMPS measurement range.

Fig. S2 Variations of particle size distribution of number and mass concentrations at the 2-h time point and at the end of the experiment for the four experiments.

9. Section 3.2. The intensity of the functional groups in FTIR spectrum was correlated to the sample mass. What was the SOA mass that collected on the disk and that measured using FTIR? Or does the author use same sampling duration for both RH conditions? What was the collection efficiency of the impactor on a sampling disk as a function of the particle size? Without knowing the mass of measured SOA, it is unreasonable to compare the peak intensity of the functional group between SOA from different samples.

We did not measure the mass of SOA collected on the disk, as the weight of ZnSe window is several grams, much larger than the SOA mass that is only several micrograms. We used the same sampling duration for both RH conditions and the FTIR spectra under different RH conditions can be comparable. The DLPI sampling flow rate was 10 L min-1, and the sampling duration was 15 min. The total collection efficiency of the DLPI was 87%, and the efficiency varies for different impaction stages (Durand et al., 2014). DLPI has 13 stages. When we sampled using DLPI, the four plates for stages 4-7 were removed, so that particles in the range of 108-650 nm were collected on the third plate. As shown in Fig. S2 that is newly added in the reply to Comment 8, the particles in the range of 108-650 nm can represent the total SOA from m-xylene oxidation in this study. The mean collection efficiency of the DLPI for stages 4-7 is around 83% (Durand et al., 2014). SOA mass was obtained by the calculation based on the SMPS measurement and the DLPI collection efficiency, 10.3 and 3.0 $\mu$g at low RH (Exp. 2) and high RH (Exp. 3). The ratio of the SOA mass collected on disk at high RH to that at low RH is 0.29. The relative intensities of most functional groups in Table 2 match this SOA mass

ratio. According to the reviewer's comment, we have added some sentences about the SOA mass collected on the ZnSe window after the first sentence of Sec. 3.2.

The DLPI sample flow rate was 10 L min-1, and the sampling duration was 15 min. We used same sampling flow rate and duration for both RH conditions. DLPI has 13 stages, and it can collect particles in the size range of 30 nm - 10 mm. When we sampled using DLPI, the four plates for stages 4-7 were removed, so that particles in the range of 108-650 nm were collected on the third plate. As shown in Fig. S2 in the supplementary information, the particles in the range of 108-650 nm can represent the total SOA from m-xylene oxidation in this study. The mean collection efficiency of the DLPI was 83% for stages 4-7 (Durand et al., 2014). Thus, the SOA mass collected on the ZnSe window was 10.3 and 3.0 $\mu$g at low RH (Exp. 2) and high RH (Exp. 3), based on the SMPS measurement and the DLPI collection efficiency.

Durand, T., Bau, S., Morele, Y., Matera, V., Bémer, D., and Rousset, D.: Quantification of low pressure impactor wall deposits during zinc nanoparticle sampling, Aerosol Air Qual. Res., 14, 1812-1821, 10.4209/aaqr.2013.10.0304, 2014.

10. Figure 2 and Table 2. There is also peak at 3000 cm-1 which is missing in Table 2.

Taking the reviewer's suggestion, we have added the peak at 3000 cm-1 in Table 2.

Table 2. Absorbance positions of functional groups and the intensities at low and high RHs. Absorption frequencies Functionality Intensity ($\times$ 10-3) Ratio a low RH high RH 3235 O-H 5.9 1.9 0.32 3000 C-H 4.5 1.4 0.31 1720 C=O 5.1 1.5 0.29 1415 CO-H 4.8 2.4 0.50 1180 C-O-C, C-O and OH of COOH 2.9 1.4 0.48 1080 C-C-OH 5.3 1.8 0.34 a Ratio of the intensity at high RH to that at low RH.

11. What is the measured glyoxal fraction in m-xylene SOA? Was oligomerization impacted by the RH in this study? Even though the concentration of highly oxygenated molecule (HOM) is much lower at high RH, the overall trend of the SOA mass, which is much less at high RH compare to low RH, cannot be explained by solely through

HOMs. As mentioned in the previous comment above, the effect of the wet wall on SOA formation can be very significant particularly in small reaction. The time scale of the gas-wall partitioning of organic species can be significantly fast and results in the less SOA yields at higher humidity.

Glyoxal in SOA was not observed in our study. Instead, we obtained the glycolaldehyde ($C_2H_4O_2$) fraction in SOA from MS/MS analysis which has been observed previously in the oxidation of m-xylene (Cocker et al., 2001). The RH suppresses oligomerization in this study as can be obviously observed in Fig. 4. In oligomerization reaction of glycolaldehyde with carbonyls by aldol condensation reactions, water is involved as a by-product, leading to the suppression of the oligomerization by high RH. Indeed, we agree with the reviewer that it cannot be solely explained through HOMs that SOA mass at high RH is much less than that at low RH. To further explain the large discrepancy of SOA mass at low and high RHs, we have added a paragraph at the end of the Sec. 3.4.

The wall process of the reactor enlarges the difference of SOA mass between low and high RH. The wall loss of some chemical species is faster at high RH, which leads to the reduction of SOA yield. In addition, the difference of SOA mass can be also enhanced based on the gas to particle partitioning rule (Li et al, 2018).

Li, K., Li, J., Wang, W., Li, J., Peng, C., Wang, D., and Ge, M.: Effects of gas-particle partitioning on refractive index and chemical composition of m-xylene secondary organic aerosol, J. Phys. Chem. A, 122, 3250-3260, 10.1021/acs.jpca.7b12792, 2018.

12 The author claimed that the increase of the C-O-C stretching was resulted from the oligomerization of carbonyls under the high RH condition. However, the IR absorption at 1080 cm-1 can be also due to the existence of C-OH group. Clarify this.

The statement in the text was incorrect. The relative intensity of the C-O-C group is higher than that of the C=O group, indicating that C-O-C group is more than other functional groups at high RH, which does not mean that the oligomerization is higher

at high RH. We have deleted the corresponding sentence in the last second sentence of the Sec. 3.2.

13. Fig. 2. Authors assigned the peak at 1600 cm-1 as carboxylate. The reviewer doubt this assignment. In general, dry organic aerosol cannot produce carboxylic acid ions. Even if the organic aerosol is produce in the wet condition, the aerosol water content is not enough to product the dissociation of carboxylic acid. In general, the pKa of carboxylic acid ranges from 2 to 6. Even in the dilution in water, less than 1% of carboxylic acid is dissociable. In SOA, most carboxylic acids will be in the un-dissociated form.

We agree with the reviewer's comment that the peak in 1605 cm-1 cannot be assigned to dissociation of carboxylic acid. Thus, we have deleted the peak assignment in Table 2 and deleted the corresponding sentence in the second and third paragraph of the Sec. 3.2.

Table 2. Absorbance positions of functional groups and the intensities at low and high RHs. Absorption frequencies Functionality Intensity ($\times$ 10-3) Ratio a low RH high RH 3235 O-H 5.9 1.9 0.32 3000 C-H 4.5 1.4 0.31 1720 C=O 5.1 1.5 0.29 1415 CO-H 4.8 2.4 0.50 1180 C-O-C, C-O and OH of COOH 2.9 1.4 0.48 1080 C-C-OH 5.3 1.8 0.34 a Ratio of the intensity at high RH to that at low RH.

14. Fig. 4. Based on Fig. 4, a large mass appeared in negative ion mode suggesting that the aerosol has a large fraction of carboxylic acid. It is contradictable compared to either MCM simulation or the conclusion by the authors in the glyoxal was abundant. In general, a large fraction of gaseous products from MCM prediction are alcohols and carbonyls, and amount of carboxylic acids are small. Please clarify this.

Indeed, a large fraction of gaseous products from MCM prediction are alcohols and carbonyls and a few compounds are organic acids. Glyoxal is one of the relatively abundant gas-phase products in MCM simulation (about 20 ppb at the end of 4 h experiment). Carbonyls in SOA account for a large fraction of compounds that have

been identified by HRMS in the positive mode (Fig. 4), some of which are in agreement with those in the gas phase simulated by MCM. In addition, we think that the large number of carboxylic acids observed in HRMS are also produced from the oxidation of organic species in the particle phase. In our original manuscript about FTIR analysis, we mentioned that the peak at 1080 cm-1 was assigned to be the C-C-OH, which could be considered as the glyoxal hydrate. This description may not be very accurate, since FTIR has limitation in identification of compounds. In further identification, the glyoxal fragment was not identified in HRMS analysis. For clarification of description of glyoxal in the text, some sentences started in Page 7 Line 7 have been modified.

The ratios of O-H, C-H, C=O and C-C-OH groups are 0.29 to 0.34, which is close to the ratio of SOA mass at high RH to that at low RH collected on the ZnSe disk, whereas the ratios of CO-H, C-O-C, C-O-H in COOH are above 0.48. The relative intensity of the C-O-C group is significantly higher than the C=O group, which can be explained by more oligomerization with the formation of C-O-C than other reactions at high RH. Nevertheless, the FTIR results cannot provide further information to well explain the differences of SOA yields between low and high RH, which will be further discussed in terms of mass spectra of SOA in the next section.

15. Fig. 4. Low carbon number products are generally more volatile than high carbon number products. Fig.4 showed that low carbon number products are high with the SOA with the low RH, possibly suggesting that volatile low carbon number products more likely deposited to the wall due to the gas-wall process.

Indeed, a fraction of low carbon number products at high RH are likely deposited to the wall. As we discussed in the reply to Comment 3, wall loss effects of organic vapors likely reduce formation of SOA, which is more obvious at high RH. However, it is considered that the decrease of low carbon number products in SOA are mainly due to the chemical reaction process at high RH.

16. What was the RH of the environment inside the FTIR spectrometer when FTIR

spectra were obtained for Fig. 4?

The sample compartment of the FTIR spectrometer was purged by dry air desiccated from FTIR purge gas generator (Model 75-45-12VDC, Balston, Parker) ahead of the FTIR measurement. The dew point temperature of the dry gas from this generator is as low as -65 °C. So the RH of the environment inside the FTIR spectrometer was extremely low, close to zero when FTIR spectra were obtained for Fig. 4.

17. What is the atmospheric implication of this study? What is the potential impact of RH on p-xylene and o-xylene as well as other aromatics? Will other aromatics also have the similar RH effects with m-xylene? What is the uncertainty of this study? Does the impaction or the PILS sampling has uncertainty? What is the duration of the experiments?

The atmospheric implication of this study is that the production of SOA from aromatics in low-NOx environments can be strongly modulated by the ambient RH probably due to the influence of H2O on the formation of HOMs and oligomers. We proposed that the clear pathway of the influence of H2O on the formation of HOMs needs to be further studied in the future. Negative RH effect on the SOA yield is a common feature of the monocyclic aromatics oxidation under low NOx conditions and using H2O2 as the OH radical source from the previous study on toluene SOA and this study on m-xylene SOA. The uncertainty of this study is mainly from the wall loss of chemical species at different RHs. In a previous study (Sorooshian et al., 2006), the PILS sampling uncertainty was reported to be 6%. The uncertainty of DLPI sampling was around 20% (Durand et al., 2014). The duration of the experiments was 4 h. Based on the reviewer's comment, we have added a sentence at the end of the Sec. 2.1 "The experiments were conducted for 4 h." and we have modified the Sec. 4 for clarification of the atmospheric implication.

Durand, T., Bau, S., Morele, Y., Matera, V., Bémer, D., and Rousset, D.: Quantification of low pressure impactor wall deposits during zinc nanoparticle sampling, Aerosol Air

Qual. Res., 14, 1812-1821, 10.4209/aaqr.2013.10.0304, 2014. Sorooshian, A., Brechtel, F. J., Ma, Y., Weber, R. J., Corless, A., Flagan, R. C., Seinfeld, J. H.: Modeling and characterization of a particle-into-liquid sampler (PILS), Aerosol Sci. Technol., 40, 396-409, 10.1080/02786820600632282, 2006.

The current study investigates the effect of RH on SOA formation from the oxidation of m-xylene under low NOx conditions in the absence of seed particles. The elevated RH can significantly obstruct the SOA formation from the m-xylene-OH system, so that the SOA yield decrease from 13.8% at low RH to 0.8% at high RH, with a significant discrepancy of higher than one order of magnitude. The FTIR results of functional groups show the relative increase of the C-O-C group at high RH as compared with low RH, indicating that the oligomers from carbonyl compounds cannot well explain the suppression of SOA yield. HOMs were observed to be suppressed in the HRMS spectra. The chemical mechanism for explaining the obvious difference of RH effects on SOA formation from m-xylene-OH system has been proposed based on the analysis of both FTIR and HRMS measurements as well as MCM simulations. The reduced SOA at high RH is mainly ascribed to the less formation of oligomers and the suppression of RO2 autoxidation. Together with the previous study on toluene SOA, it is conceivable that the negative RH effect on the SOA yield is a common feature of the monocyclic aromatics oxidation under low NOx conditions and using H2O2 as the OH radical source. Our results obviously indicate that the production of SOA from aromatics in low-NOx environments can be strongly modulated by the ambient RH probably due to the influence of H2O on the formation of HOMs and oligomers. Our study highlights the role of water in the SOA formation, which is particularly related to chemical mechanisms used to explain observed air quality and to predict chemistry in air quality models and climate models. The clear pathway of the influence of H2O on the formation of HOMs needs to be further studied in the future.

18. There are numerous grammatical problems. The manuscript needs to be approved by a native English speaker.

According to the advice of the reviewer, we have read the manuscript carefully and modified some sentences to correct grammatical errors.

19. Page 3 Line 13. The author may need to provide the reason why the author set the density of m-xylene SOA is set as 1.4 g/cm. According to the citation Ng et al. 2007 gives the density of m-xylene SOA as 1.33 _ 0.1 g/cm and Sato et al. 2007 provide the density of Toluene SOA as 1.42 _ 0.8 g/cm.

We agree with the reviewer's comment that the density we cited is not consistent with this study. We made a mistake when we cited references. Thus, we have corrected this mistake in the text, "To obtain the particle mass concentrations and SOA yield, an SOA density of 1.4 g cm-3 was used (Song et al., 2007)."

Song, C., Na, K., Warren, B., Malloy, Q., and Cocker, D. R., III: Secondary organic aerosol formation from m-xylene in the absence of NOx, Environ. Sci. Technol., 41, 7409-7416, 10.1021/es070429r, 2007.

20. Page 3 Line 17. The author mention about the uncertainty of SMPS. However, the uncertainty of SMPS measurement for the used data in the paper was not reported.

As we mentioned in the text, SMPS measurement uncertainty is mainly dominated by size-dependent aerosol charging efficiency uncertainties and CPC sampling flow rate variability. The size-dependent aerosol charging efficiency is typically characterized by an accuracy of $\pm$ 10% (Jiang et al., 2014), which was used in our study to calculate the uncertainties of SOA mass concentration on the premise that the uncertainty of organic vapor wall loss was not included. The standard error of linear regression of the m-xylene concentration and peak area obtained by GC-MS was 0.013, which was extremely low compared with the m-xylene concentration and can be negligible. Thus, the uncertainty of 10% was used in SOA yield. In the revised manuscript, we have modified the Table 1 with the addition of uncertainty of SOA mass concentration.

Table 1. Experimental conditions, SOA concentrations and yields at the end of the

experiments in m-xylene-OH oxidation system. Exp. No. [m-xylene]0 ($\mu$g m-3) [m-xylene]reacted ($\mu$g m-3) RH (%) T ($^\circ$C) [SOA]e ($\mu$g m-3) SOA yield (%) 1 2287.9 1026.3 13.6 25.9 150.3 $\pm$ 15.0 14.6 $\pm$ 1.5 2 1855.5 682.0 13.7 25.3 95.5 $\pm$ 9.5 14.0 $\pm$ 1.4 3 2410.8 941.4 73.6 27.5 21.0 $\pm$ 2.1 2.2 $\pm$ 0.2 4 2029.1 946.9 79.1 27.4 7.5 $\pm$ 0.7 0.8 $\pm$ 0.1 [SOA]e indicates the mass concentration of SOA at the end of each experiment with particle wall loss corrected.

21. Page 4 Line 6. For comparing the SOA yield between cited values and that of this study, it may need to provide the error range of the values. Additionally, numerous data of this paper need errors.

We used the uncertainty of 10% to calculate SOA mass concentration from SMPS measurement. As we explained in Comment 21, the uncertainty of 10% was used in SOA yield (see Table 1 above).

22. Pages 6 and 7 (section 3.2): The description to construct the functional group distribution using FTIR spectra is unclear. How to separate the FTIR peaks for each functional group? The intensity of each function group varies with vibration force constant and peak broadening changes with compositions.

We used the peak height to represent the functional group distribution. The separation of FTIR peaks for each functional group was conducted by the peak valleys between two peaks.

Minor comments:

Page 2, line 21. The sentence is confused that it compares the RH effects between low NOx condition and with NOx condition.

Taking the reviewer's suggestion, we have corrected the sentence.

However, under low NOx level, it has been found that, in the study on toluene SOA formation, moderate RH level (48%) leads to a lower SOA yield than low RH level (17-18%).

Figure 3. The y-axis scale is negative for high RH and positive for Low RH. It is better to make them as a same positive scale.

Taking the reviewer's advice, we have modified the y-axis scale for high RH in Fig 3.

Figure 3. Selected background-subtraction HESI-Q Exactive-Orbitrap MS results of SOA in both positive and negative ion modes from the photooxidation of m-xylene-OH under both low and high RH conditions (Note that the Y-axis scales for low and high RH are largely different, 106 at low RH and 105 at high RH).

Figure 5: RH scale should range from 0 to 100 (negative is incorrect).

Taking the reviewer's advice, we have modified the RH scale and corrected the negative scale in Fig 5.

Figure 5. Mass spectra of SOA from m-xylene at both low (red) and high (blue) RH in the positive (+) and negative (-) ion modes, grouped with the same number of carbon atoms (from nC =8 to 16). On (n = 2, 3, ......, 12) means the number of oxygen atoms in the formula of the peak.

Page 2 line 15: it is better to use "have been conducted".

Taking the reviewer's advice, we have corrected this mistake, "Investigations of RH effects on aromatics SOA have been conducted in many previous works."

Page 2 line 19: after "as an OH radical source," there supposed to use period instead of comma.

In this sentence, "no NOx were introduced artificially and photolysis of H2O2 was as an OH radical source" is an appositive clause to explain the "condition", so the part before the comma is an adverbial modifier and thus we use a comma.

Table 3. The form of the table is better to unify with other tables.

Taking the reviewer's advice, we have unified the Table 3 with other tables.

Table 3. Plausibility of different types of compounds with elemental formulae measured by HRMS in the positive ion mode. Low RH High RH Ion formula Proposed structure Measured (m/z) Intensity Error (mDa) Measured (m/z) Intensity Error (mDa) 137.0596 $1.7 \times 10^6$ 0.6 137.0593 $1.4 \times 10^5$ 1 [C8H9O2]+

141.0545 $5.6 \times 10^6$ 1.3 141.0542 - 1.5 [C7H9O3]+

155.0701 $1.0 \times 10^6$ 1.2 155.0699 - 1.5 [C8H11O3]+

171.0651 $1.0 \times 10^6$ 1.2 171.0649 - 1.4 [C8H11O4]+

187.06 $1.1 \times 10^6$ 1.2 187.0568 - 4.4 [C8H11O5]+

Please also note the supplement to this comment:
https://www.atmos-chem-phys-discuss.net/acp-2019-20/acp-2019-20-AC2-supplement.pdf

---

## Author Response (AR2)

**Response to Referee #1**

We greatly appreciate the time and effort that referee 1 spent in reviewing our manuscript. Below we make a point-by-point response to these comments. According to editor's requirement, the response to the referee 1 is structured in the following sequence: (1) comments from the referee in black color, (2) our response in blue color, and (3) our changes in the revised manuscript in red color.

Although I find the authors' explanations reasonable, I feel that they should add a sentence or two in the manuscript to clarify:
1. why they decide to use a PILS

Taking the referee's suggestion, we have added two sentences in the third paragraph in Sec. 2.2 in the revised manuscript.

The PILS samples water-soluble species in particles. As the SOA compositions are almost all water-soluble species, it is reasonable and reliable to use PILS to sample SOA for analysis of chemical composition.

2. that the HPLC was only used as an injection system

Taking the referee's suggestion, we have added a sentence in last paragraph in Sec. 2.2 in the revised manuscript.

20    In this study, the UPLC was only used as the injection system of HRMS.

**Response to Referee #3**

We greatly appreciate the time and effort that referee 3 spent in reviewing our manuscript. The comments are really thoughtful and helpful to improve the quality of our paper. Referee 3 has provided both main comments and other specific comments. Below we make a point-by-point response to these comments. According to editor's requirement, the response to the referee 3 is structured in the following sequence: (1) comments from the referee in black color, (2) our response in blue color, and (3) our changes in the revised manuscript in red color.

The study proposed by Zhang et al, presents new results related to the effect of relative humidity on the formation and composition of SOA generated from the OH-initiated oxidation of xylene. While it is an interesting topic, this study cannot be published as it is, as different aspects/conclusions of the paper are not well constrained or speculative (see below). I suggest that the authors have a close look at the existing literature and provide strong(er) evidence (e.g., MS2 spectra, ...) to support their statements.

1. Page 2, lines 33-34: Please provide references and/or results.

In fact, the references have been given at the end of the mentioned sentence (Jia and Xu, 2014, 2018) in the revised manuscript. Taking the referee's suggestion, we have added two sentences about the results following the mentioned sentence in the revised manuscript.

In these studies, O-H, C=O, C-O, and C-OH were found to be the main functional groups, intensities of which largely increased with increasing RH. Compounds in SOA with the O-H group mainly contributed to the increasement of SOA, such as polyalcohols formed from aqueous reactions.

2. Page 3. Lines 4-15. This section is not useful/needed, especially as it is disconnected to the rest of the introduction. The authors mainly focused on particle phase processes. Different aspects are actually missing including acidity effect (directly linked to LWC) and/or viscosity/phase effect on particle-phase reactions and can be added to the introduction.

Taking the referee's suggestion, we have deleted this section in the revised manuscript. We have also added several sentences about the acidity effect and viscosity/phase effect on particle-phase reaction at the end of the third paragraph in the introduction section in the revised manuscript.

Acid-catalysis of heterogeneous reactions of atmospheric organic carbonyl species in particle phase can lead to a large increase of SOA mass, while this process can be suppressed by the lower acidity at high RH (Czoschke et al., 2003). In

addition, RH can change the viscosity of SOA and further affect the chemical processes of SOA formation (Kidd et al., 2014; Liu et al., 2017).

Czoschke, N. M., Jang, M., and Kamens, R. M.: Effect of acidic seed on biogenic secondary organic aerosol growth, Atmos. Environ., 37, 4287-4299, 10.1016/s1352-2310(03)00511-9, 2003.

Kidd, C., Perraud, V., Wingen, L. M., and Finlayson-Pitts, B. J.: Integrating phase and composition of secondary organic aerosol from the ozonolysis of alpha-pinene, Proc. Natl. Acad. Sci. U. S. A., 111, 7552-7557, 10.1073/pnas.1322558111, 2014.

Liu, Y., Wu, Z., and Hu, M.: Advances in the phase state of secondary organic aerosol (in Chinese), China Environ. Sci., 37, 1637-1645, 2017.

3. Paragraph 3.1. This section should be revised and better organized. As it is, it is very difficult to follow. In addition, the authors should discuss the impact of reacted VOC on SOA yields when comparing different studies.

Taking the referee's advice, we have revised and reorganized the Sec. 3.1 for the clear understanding. The referee 4 suggested us to add some experiments at different RHs, of which results have been included in the revised Sec. 3.1. The impact of reacted m-xylene on SOA yields has been also included in the revised manuscript based on the newly added Fig. 2 according to the comment 7.

[revised manuscript text omitted]

4. Page 5, line 20. Why did the authors stop the experiments after 4h, while the SOA mass was still increasing? Usually, SOA yields are calculated when the oxidation of the precursor is over. Was H2O2 still present in the chamber?

As the referee pointed out, the SOA mass was still increasing after 4 h. The experiment for 4-6 h is a ubiquitous reaction time used in many previous studies. The SOA yield indeed generally increases with time. $H_2O_2$ was still present in the chamber after 4 h. If the relationship between the yield and time is extrapolated to 6 h, the yield will be increased by 45% relative to that at 4 h (Exp. 1), which can be compared with many previous studies (Cao and Jang, 2010; Hinks et al., 2018). Most importantly, as the purpose of our study is to investigate the RH effect on SOA formation, the reaction time of 4 h is sufficient to compare the SOA formation and to sample for SOA component analysis. Furthermore, a relatively short reaction time can minimize the wall loss of oxidized species and limit the further SOA mass uncertainty.

Cao, G., and Jang, M.: An SOA model for toluene oxidation in the presence of inorganic aerosols, Environ. Sci. Technol., 44, 727-733, 10.1021/es901682r, 2010.

Hinks, M. L., Montoya-Aguilera, J., Ellison, L., Lin, P., Laskin, A., Laskin, J., Shiraiwa, M., Dabdub, D., and Nizkorodov, S. A.: Effect of relative humidity on the composition of secondary organic aerosol from the oxidation of toluene, Atmos. Chem. Phys., 18, 1643-1652, 10.5194/acp-18-1643-2018, 2018.

5. Page 5, lines 25-27; 30-32. I do not think it is a correct explanation. Indeed, as reported in many laboratory studies, SMPS measurements are reliable and do not exhibit such a large variability (15-20%). The authors should better constrain the evaluation of the uncertainties and provide references supporting their hypothesis.

We agree with the referee that in many studies SMPS measurements do not exhibit the variability as large as in our study. However, there are also some studies with such a large variability of SMPS measurement. For example, Sakamoto et al. (2013) found a large uncertainty (~20% or even more, the ratio of standard deviation to mean value) within 2.5 min when measuring particulate volume concentration. Technically, according to the instruction manual of the ultrafine condensation particle counter (Model 3776), the particle concentration accuracy is $\pm$ 10% at $< 3 \times 10^5$ particles cm$^{-3}$. The number concentrations at the end of each experiment in this study were below $5 \times 10^3$ particles cm$^{-3}$, so in this study the particle concentration error caused by CPC alone was $\pm$ 10%. As can be seen in Fig. S2, the particle distribution falls into two particle diameter ranges. Here take Exp.1 as an example to explain, 130.0-371.8 nm and 461.4-661.2 nm. For evaluation of the uncertainty of the mass concentration, we assume that if the 10% uncertainty of number concentration exists in the particle diameter range of 461.4-661.2 nm (Exp. 1), the uncertainty of mass concentration would be 17%. This is more likely to happen in this study as the number concentration is extremely lower than that in many other studies ($10^4$-$10^5$ particles cm$^{-3}$ order of magnitudes), in which the seed particle was used or the background air was not clean enough with the presence of SO$_2$. In addition, size-dependent aerosol charging efficiency uncertainties and CPC sampling flow rate variability also dominate the SMPS measurement uncertainty. The combination of various uncertainties, including SMPS measurement, sampling and even conversion of mass concentration from number concentration leads to the fairly large scatter in Fig. 1. Based on the reply above, we have rewritten the explanation in the revised manuscript.

Sakamoto, Y., Inomata, S., and Hirokawa, J.: Oligomerization reaction of the Criegee intermediate leads to secondary organic aerosol formation in ethylene ozonolysis, J. Phys. Chem. A, 117, 12912-12921, 10.1021/jp408672m, 2013.

The interval of SOA data sampled by SMPS was 5 minutes, for which the sampling frequency was relatively low. Technically, according to the instruction manual of the CPC (Model 3776), the particle concentration accuracy is $\pm$ 10% at $< 3 \times 10^5$ particles cm$^{-3}$. The number concentrations at the end of each experiment in this study were below $5 \times 10^3$ particles cm$^{-3}$, so in this study the particle concentration error caused by CPC alone was $\pm$ 10%. In addition, size-dependent aerosol

charging efficiency uncertainties and CPC sampling flow rate variability also dominate the SMPS measurement uncertainty. The combination of various uncertainties, including SMPS measurement, sampling and even conversion of mass concentration from number concentration leads to the fairly large scatter in Fig. 1.

5    6. Page 6, line 5. How did the authors conclude that the underestimation was negligible? Please provide a clear explanation.

The underestimation comes from the dissolved species that are probably volatile/semi-volatile compounds to evaporate back into the gas phase when aerosol water is removed. Glyoxal is a typical semi-volatile compound that is involved in SOA formation in m-xylene-OH system of our study. We will take glyoxal as the example to interpret why the underestimation

10    was negligible. The LWC after 4 h from the start of reaction in Exp. 7 which was under the highest RH condition was 4.4 μg m$^{-3}$ as shown in Table 1. According to the MCM prediction, the final concentration of glyoxal during the 4 h of experiment time is 17 ppb. The Henry's law constant of glyoxal in pure water is $4.19 \times 10^5$ M atm$^{-1}$ at 298 K (Ip et al., 2009). The glyoxal dissolved in the LWC can be calculated to be less than 1 ppt. Thus, the evaporation of the volatile/semi-volatile compounds back to the gas phase can be negligible. In addition, growth factor (GF) was determined to be 1.2 in Exp. 7 of

15    our study. Aklilu and Mozurkewich (2004) gave a GF range of 1.05–1.12 for atmospheric organic particles (79% RH). Cocker et al (2001) reported the GF of 1.14 at 85% RH for SOA formed from m-xylene. In general, our results of GF are in good agreement with previous studies, indicating that the LWC measured by our modified SMPS is reliable. Though the removal of aerosol water may cause the dissolved species that are probably volatile/semi-volatile compounds to evaporate back into the gas phase, which may lead to the underestimation of SOA, it can be negligible. Thus, we have added some

20    sentences in the first paragraph in the Sec. 3.1 in the revised manuscript.

Aklilu, Y. A. and Mozurkewich, M.: Determination of external and internal mixing of organic and inorganic aerosol components from hygroscopic properties of submicrometer particles during a field study in the lower fraser valley, Aerosol Sci. Tech., 38, 140-154, 10.1080/02786820490251367, 2004.

25    Cocker, D. R., Mader, B. T., Kalberer, M., Flagan, R. C., and Seinfeld, J. H.: The effect of water on gas-particle partitioning of secondary organic aerosol: II. m-xylene and 1,3,5-trimethylbenzene photooxidation systems, Atmos. Environ., 35, 6073-6085, 10.1016/s1352-2310(01)00405-8, 2001.
Engelhart, G. J., Hildebrandt, L., Kostenidou, E., Mihalopoulos, N., Donahue, N. M., and Pandis, S. N.: Water content of aged aerosol, Atmos. Chem. Phys., 11, 911-920, 10.5194/acp-11-911-2011, 2011.

30    Ip, H. S. S., Huang, X. H. H., and Yu, J. Z.: Effective Henry's law constants of glyoxal, glyoxylic acid, and glycolic acid, Geophys. Res. Lett., 36, L01802, 10.1029/2008gl036212, 2009.

Glyoxal is a typical semi-volatile compound with high Henry's law constant, which is involved in SOA formation in *m*-xylene-OH system of our study. The Henry's law constant of glyoxal in pure water is as high as $4.19 \times 10^5$ M atm$^{-1}$ at 298 K

(Ip et al., 2009). Only one in ten thousand of glyoxal can dissolve in the LWC whose concentration was obtained in our study.

Ip, H. S. S., Huang, X. H. H., and Yu, J. Z.: Effective Henry's law constants of glyoxal, glyoxylic acid, and glycolic acid,
5   Geophys. Res. Lett., 36, L01802, 10.1029/2008gl036212, 2009.

7. Page 6, lines 20-28. To help the reader and underline the differences, the authors should make a figure presenting the SOA yields as a function of RH for the different aromatic systems (toluene, xylene,...). It would help picturing the effect of RH. In addition, they can use the size of the markers as the function of reacted VOC as the concentration of reacted VOC has a great
10   impact on SOA yields.

Taking the referee's suggestion, we have made the figure and added it to the revised manuscript as Figure 2. Also, we have added some sentences about this figure in the last paragraph of the Sec. 3.1 in the revised manuscript which can be seen in the reply of comment 3.

[Figure]

Figure 2. SOA yields as a function of RH for the different aromatic (toluene and *m*-xylene) oxidation under low NO$_x$ conditions with the photolysis of H$_2$O$_2$ as the OH source. The hollow circles represent that no seed particles were introduced and the circles with a cross represent that seed particles were introduced. The size of markers indicates the magnitude of
20   amount of reacted VOC.

8. Page 8, lines 23-25. Please provide references to support the assignments. if the band at 1605 cm-1 corresponds to liquid water, why is the absorption lower in humid conditions when the LWC is expected to be larger?

Taking the referee's suggestion, we have added references to support the assignments in the second paragraph of Sec. 3.2 in the revised manuscript. In addition, we corrected the mistake about the assignment of the band at 1605 $cm^{-1}$ in the revised manuscript. In fact, the sample compartment of the FTIR spectrometer was purged by dry air desiccated from FTIR purge gas generator (Model 75-45-12VDC, Balston, Parker) ahead of the FTIR measurement. The dew point temperature of the dry gas from this generator is as low as -65 °C, so the RH of the environment inside the FTIR spectrometer was extremely low, close to zero when FTIR spectra were obtained. The band at 1605 $cm^{-1}$ should be C-C stretching of aromatic rings and the C=O stretching of conjugated carbonyl groups.

The broad absorption at 3600-2400 $cm^{-1}$ is O-H stretching vibration in phenol, hydroxyl and carboxyl groups (Stevenson and Goh, 1971; Santos and Duarte, 1998; Duarte et al., 2005). The band at 3000 $cm^{-1}$ is C-H stretching vibration (Stevenson and Goh, 1971; Santos and Duarte, 1998; Duarte et al., 2005). The sharp absorption at 1720 $cm^{-1}$ is the C=O stretching vibration in carboxylic acids, formate esters, aldehydes and ketones (Stevenson and Goh, 1971; Santos and Duarte, 1998; Duarte et al., 2005). The absorptions at 1605 $cm^{-1}$ match C-C stretching of aromatic rings and the C=O stretching of conjugated carbonyl groups. The absorptions at 1415 $cm^{-1}$ match the deformation of CO-H, phenolic O-H and C-O (Coury and Dillner, 2008; Ofner et al., 2011). The absorptions at 1180 $cm^{-1}$ match the C-O-C stretching of polymers, C-O and OH of COOH groups (Jang and Kamens, 2001; Jang et al., 2002; Duarte et al., 2005). The absorptions at 1080 $cm^{-1}$ match the C-C-OH stretching of alcohols (Jang and Kamens, 2001; Jang et al., 2002).

Coury, C., and Dillner, A. M.: A method to quantify organic functional groups and inorganic compounds in ambient aerosols using attenuated total reflectance FTIR spectroscopy and multivariate chemometric techniques, Atmos. Environ., 42, 5923-5932, 10.1016/j.atmosenv.2008.03.026, 2008.

Duarte, R. M. B. O., Pio, C. A., and Duarte, A. C.: Spectroscopic study of the water-soluble organic matter isolated from atmospheric aerosols collected under different atmospheric conditions, Analytica Chimica Acta, 530, 7-14, 10.1016/j.aca.2004.08.049, 2005.

Jang, M., Czoschke, N. M., Lee, S., and Kamens, R. M.: Heterogeneous atmospheric aerosol production by acid-catalyzed particle-phase reactions, Science, 298, 814-817, 10.1126/science.1075798, 2002.

Jang, M. S., and Kamens, R. M.: Characterization of secondary aerosol from the photooxidation of toluene in the presence of $NO_x$ and 1-propene, Environ. Sci. Technol., 35, 3626-3639, 10.1021/es010676+, 2001.

Ofner, J., Kruger, H.-U., H., G., Schmitt-Kopplin, P., Whitmore, K., and Zetzsch, C.: Physico-chemical characterization of SOA derived from catechol and guaiacol-a model substance for the aromatic fraction of atmospheric HULIS, Atmos. Chem. Phys., 11, 1-15, 10.5194/acp-11-1-2011, 2011.

Santos, E. B. H., and Duarte, A. C.: The influence of pulp and paper mill effluents on the composition of the humic fraction of aquatic organic matter, Wat. Res., 32, 597-608, 10.1016/S0043-1354(97)00301-1, 1998.

Stevenson, F. J., and Goh, K. M.: Infrared spectra of humic acids and related substances, Geochim. Cosmochim. Acta, 35, 471-483, 10.1016/0016-7037(71)90044-5, 1971.

9. Page 8, lines 29-30. Are the ratio calculated for 1 experiment or for all the experiments?

The ratio was not calculated for all experiments but using the results from Exps. 2 and 6. According to the comment 1 of referee 4, we have added three experiments and we renew the serial number of each experiment as shown in Table 1. For clear statement of how we calculated the ratio, we have added the serial numbers of low and high experiments to the first sentence in paragraph 2 and the second sentence in paragraph 3 of Sec. 3.2, as well as the title of Table 2.

The assignment and the intensity of the FTIR absorption frequencies at low (Exp. 2) and high (Exp. 6) RHs is summarized in Table 2.

As well, Table 2 gives the ratio of intensities at high RH (Exp. 6) to those at low RH (Exp. 2) to compare the difference of relative intensities of functional groups.

Table 2. Absorbance positions of functional groups and the intensities at low (Exp. 2) and high (Exp. 6) RHs.

10. Page 9, lines 15-16. Please provide the MS2 spectra in the SI to support the discussion and the structure assignments.

Taking the referee's suggestion, we have added the MS$^2$ spectra in Figure S3 and the breakage mechanism in Fig. S4 of the Supporting Information.

[Figure]

Figure S3. MS$^2$ spectrum of the parent ion at m/z = 137 (A), m/z = 141 (B), m/z = 155 (C), m/z = 171 (D) and m/z = 187 (E) in Table 3. The red columns are the main fragments those are matched with the fragments proposed by the Mass Frontier program.

**(A) m/z = 137.0603**

m/z 137.06

m/z 137.06

m/z 137.06

m/z 109.06

m/z 137.06

m/z 137.06

m/z 137.06

m/z 137.06

m/z 95.05

m/z 137.06

m/z 91.05

m/z 141.05

m/z 123.04

m/z 141.05

m/z 141.05

m/z 115.04

m/z 115.04

m/z 113.02

m/z 141.05

m/z 141.05

m/z 111.04

m/z 141.05

m/z 141.05

m/z 141.05

m/z 99.04

m/z 141.05

m/z 97.03

**(C) m/z = 155.0714**

[Figure]

**(D) m/z = 171.0663**

m/z 171.07      m/z 153.05

m/z 171.07      m/z 153.05      m/z 153.05      m/z 125.06

m/z 171.07      m/z 171.07      m/z 171.07      m/z 154.06      m/z 111.04

m/z 171.07      m/z 153.05      m/z 153.05      m/z 127.04      m/z 99.04

**(E) m/z = 187.0612**

Figure S4. Breakage mechanisms of the parent ion at m/z = 137 (A), m/z = 141 (B), m/z = 155 (C),m/z = 171 (D) and m/z = 187 (E) in HRMS, proposed in the Mass Frontier program. The process symbols of Lib, i and rH represent the pathway from the HighChem Fragmentation Library, the pathway of inductive cleavage, and the charge site rearrangement.

11. Page 9, line 24. It would be very surprising to have such reactive species (e.g., 155, 187) within the particle phase. Please provide much deeper MS2 analyses to further confirm the presence of such species and add references.

$MS^2$ spectra of m/z=155 and 187 have been added in Fig. S3 in the Supporting Information. The formula of m/z = 155 and 187 in positive ion mode are identified to be $C_8H_{10}O_3$ and $C_8H_{10}O_5$, respectively. The rate constants of reaction of $C_8H_{10}O_3$

and $C_8H_{10}O_5$ with OH are $8 \times 10^{-11}$ and $4 \times 10^{-11} \, cm^3$ molecule$^{-1}$ s$^{-1}$ in the MCM, with a factor of 4 and 2 compared with that of *m*-xylene with OH, respectively. According to the MCM prediction, the concentration of these two compounds in the gas phase can reach 20 and 8 ppb respectively during the reaction process. In other words, though these two compounds are relatively reactive, they can still partition in the particle phase. In addition, the formation pathways of the MS$^2$ fragment of m/z=155 and 187 proposed by the Mass Frontier program are shown in the Fig. S4 in the supporting information, as can be seen in the reply of comment 10.

12. Page 10, lines 1-5. The most intense peak corresponds to C10 compounds, so according to the author's definition, those compounds are oligomers. It is surprising that oligomers (dry conditions, no seed) contribute to a large fraction of SOA mass. Indeed, larger SOA mass would result in a larger aerosol surface area and lead to a greater condensation sink for SVOC and LVOC. In other words, it can be expected that larger SOA formation leads to a greater concentration of SVOC (i.e., monomer) within the particle phase. Please provide a deeper analysis of gas-particle partitioning in your experiments: what's the surface area of the aerosols under dry vs wet conditions and compare it with the surface area of the wall. Under dry conditions the aerosol would act as a main sink, while under wet conditions the chamber walls would be the main sink.

Indeed, as can be seen in Fig. 5, the compounds with lower nC (nC=4-7) in the particle phase were observed at low RH but not at high RH. In other words, more compounds with high volatility are condensed in the particle phase at low RH. The surface area of the reactor wall was 6.6 m$^2$. At 4 h of Exps. 1 and 7, the surface concentrations of the aerosols were $1.01 \times 10^9$ and $1.25 \times 10^8 \, nm^2 \, cm^{-3}$ at low and high RHs, respectively. The ratio of the aerosol surface area to the surface area of the wall was $1.5 \times 10^{-4}$ and $1.9 \times 10^{-5}$ at low and high RHs, respectively. If the chamber wall were the main sink at high RH, particles at low RH could be also lost on the chamber wall. Thus, some chemical processes must dominate the negative RH effect on SOA formation other than gas to particle partitioning. In addition, the comment about the C10 compounds is closely related to the next comment, so please see the reply of the next comment.

13. Page 10, line 15. What should be the production of glycolaldehyde to explain the formation/concentration of such high molecular weight compounds? It is confusing to not see any dimers and/or lower oligomers (nC = 3,4,5,... monomers). In other words, the authors are suggesting that from gaseous glycolaldehyde, such aldol condensation reaction will form only/mainly oligomers with 8 monomers. If such a process takes place the authors should be able to find the distribution of the different oligomers. Is it the case? The authors should look at the MS/MS spectra of the compounds from C4 to C10 to look for the presence of oligomers (nC = 3,4,5,..). The MS/MS spectra should be provided. Is there any evidence of oligomerization between a xylene-monomer and a carbonyl (e.g., glycolaldehyde)? Finally, such aldol processes are generally considered to be too slow to be observed, especially without the presence of an acidic solution (Herrmann et al., 2015).

We went through the MS$^2$ spectra of C4-C9 compounds and did not find any $C_2H_5O_2$ fragment that is generally identified as glycolaldehyde. We agree with the referee that the formation of oligomers with glycolaldehyde part would not only be in C10 compounds. The C10 compounds may not be from the oligomerization of glycolaldehyde and monomers. So we have deleted the explanation of C10 compounds in the revised manuscript.

14. Have the authors considered the reaction of glycolaldehyde with C7-C8 monomers?

We considered the reaction of glycolaldehyde with C8 monomers as the Comment 13 mentioned, but we did not consider the reaction of glycolaldehyde with C7 monomers, since we did not find any $C_2H_5O_2$ fragment in C9 compounds in MS$^2$ spectra,

10    as we replied the Comment 13. Thus, the $C_2H_5O_2$ fragment in C10 compounds cannot be considered as glycolaldehyde.

15. Paragraph 3.4. This section is incorrect and should be revised according to the existing literature on HOM. The authors characterized the distribution of oxygenated species in particle phase as a function of RH. According to Bianchi et al., 2019, HOM refer to gas-phase highly oxygenated molecules formed from autoxidation. So it is speculative to claim that RH

15    impacts HOM formation as the authors did not measure such species. In addition, it is contradicting a recent study proposed by Li et al. 2019 (doi.org/10.5194/acp-19-1555-2019) showing that RH does not impact the formation of HOM. It is possible that HOM undergo further particle-phase reactions as it has been suggested in a few studies (i.e., Bianchi et al., 2019).

Taking the referee's advice, we have modified Sec. 3.4 in the revised manuscript.

[revised manuscript text omitted]

We greatly appreciate the time and effort that referee 4 spent in reviewing our manuscript. The comments are really thoughtful and helpful to improve the quality of our paper. Referee 4 has provided both main comments and other specific comments. Below we make a point-by-point response to these comments. According to editor's requirement, the response to the referee 4 is structured in the following sequence: (1) comments from the referee in black color, (2) our response in blue color, and (3) our changes in the revised manuscript in red color.

The effect of relative humidity on yield and chemical composition of secondary organic aerosol originated from the photooxidation of m-xylene with OH radicals was investigated. Two analytical techniques were used to investigate the SOA chemical composition: Fourier transform infrared spectrometer (FTIR) for functional groups analysis and ultrahigh performance liquid chromatograph-electrospray ionization-high-resolution mass spectrometer (UPLC-ESI-HRMS) for structural elucidation. Two Relative humidity were investigated in this study. A chemical mechanism was proposed to account for the RH effect on SOA formation.

The experimental results show a significant yield increase as well as changes in chemical composition when the RH was increased from ~14% to ~75%. These results are consistent with most literature data either for the same system or for other aromatics (e.g. toluene). These data are important to atmospheric scientists interested in local/regional and global modeling because of the high-water content in the atmosphere. The role of water in atmospheric chemistry is still not well understood.

1. The authors investigated the effect of RH only at two points. I do feel that additional RH (for example at dry conditions <5%, ~30 and 45 %) will benefit the paper and the conclusion associated with the chemical composition changes as the RH changes.

Taking the reviewer's advice, we have added a few experiments at the RH values between 14% and 74%, as shown in Exps. 3-5 of Table 1. We have also replotted Fig. 1 for the addition of these three new experiments. The addition of these three experiments reconfirms the result of negative effect of RH on SOA formation. According to this comment and the comment 3 of referee 3, we have revised the Sec. 3.1. Please also see the reply of comment 3 of referee 3.

Table 1. Experimental conditions, SOA concentrations and yields at 4 h of experiments in m-xylene-OH oxidation system.

| Exp. No. | $[m\text{-xylene}]_0$ ($\mu g\ m^{-3}$) | $[H_2O_2]_0{}^a$ (ppm) | RH (%) | T (°C) | $[m\text{-xylene}]_{reacted}$ ($\mu g\ m^{-3}$) | $[LWC]_{4h}{}^b$ ($\mu g\ m^{-3}$) | $[SOA]_{4h}{}^b$ ($\mu g\ m^{-3}$) | SOA yield (%) |
|---|---|---|---|---|---|---|---|---|
| 1 | 2287.9 | 20 | 13.6 | 25.9 | 1026.3 | - | $150.3 \pm 15.0$ | $14.6 \pm 1.5$ |

| 2 | 1855.5 | 20 | 13.7 | 25.3 | 682.0 | - | 95.5 ± 9.5 | 14.0 ±1.4 |
| 3 | 2157.1 | 20 | 34.0 | 26.0 | 922.9 | 3.5 ± 0.3 | 61.1 ± 6.1 | 6.6 ± 0.7 |
| 4 | 2041.9 | 20 | 50.5 | 25.5 | 837.2 | 2.9 ± 0.3 | 33.3 ± 3.3 | 4.0 ± 0.4 |
| 5 | 2233.3 | 20 | 63.0 | 25.9 | 722.5 | 5.4 ± 0.5 | 25.0 ± 2.5 | 3.5 ± 0.3 |
| 6 | 2410.8 | 20 | 73.6 | 27.5 | 841.4 | 7.7 ± 0.8 | 21.0 ± 2.1 | 2.5 ± 0.2 |
| 7 | 2029.1 | 20 | 79.1 | 27.4 | 946.9 | 4.4 ± 0.4 | 7.5 ± 0.7 | 0.8 ± 0.1 |

[a]Calculated using the density and mass concentration of injected $H_2O_2$ solution, and the volume of the reactor.

[b]The mass concentration at 4 h of reaction time with particle wall loss corrected.

[Figure]

5     Figure 1. SOA mass concentrations as a function of irradiation time (corrected by particle wall loss).

2. The volume of the reactor (although "smog chamber" was used: see my comment below) is 1-m3 and seems to me low. It is better to use the term Teflon bag instead of smog chamber (the experimental section is not clearly described). The use of small chamber volume is prone to high dilution rate when sampling is underway (SMPS, PILS…) and the authors did not

10     correct the dilution in their yield calculations! The author needs to describe the experimental section more clearly: the sampling time (when the samples were taken either for the PILS or for FTIR, SMPS) should be provided. The authors mention only the samples were taken at the end of the experiment. What the end of experiment refers too here?

Taking the referee's suggestion, we have changed the term smog chamber to the Teflon bag in the context of revised

15     manuscript. Our Teflon bag is air-tight. When sampling is underway, there is no dilution. Indeed, the reactor volume is getting small when sampling but the change was not significant, as the total volume for sampling was only around 110 L during the reaction time, approximately 10% of the size of reactor. The durations of PILS for HRMS and DLPI for FTIR are 5 min and 15 min, respectively, and these two sampling were taken just after 4 h of reaction. Taking the referee's advice, we

have added more content that the reactor is air-tight at the beginning of Sec. 2.1 and the sampling time of DLPI and PILS in the paragraphs 2 and 3 of the Sec. 2.2 in the revised manuscript. The experiments lasted for 4 h, so the end of experiment refers to 4 h of reaction from the start of the experiment, which has been addressed in the Sec. 2.1. In addition, we have added a sentence to state what is the end of experiment referred to in our study at the end of the Sec. 2.1 in the revised

5    manuscript.

Experiments of *m*-xylene photooxidation were performed in a 1 m$^3$ air-tight Teflon FEP film reactor…

The duration of DLPI for FTIR was 15 min, and this sampling was taken just after 4 h of each experiment.

The duration of PILS was 5 min, and this sampling was taken just after 4 h of each experiment.

Thus, the "end" of the experiment in this study refers to the experiment at 4 h of reaction time.

15    3. The yield was calculated by subtracting the LWC from the SOA mass. Two methods were used to measure (estimate) LWC at low and high RH. LWC measurements should be described in the manuscript clearly.

Taking the referee's suggestion, we have rewritten the description of LWC measurement for clarification.

20    The size distribution and concentrations of particles were monitored with a scanning mobility particle sizer (SMPS, Model 3936, TSI, USA). The particle wall loss constant has been determined to be $3.0 \times 10^{-5}$ s$^{-1}$ and $6.0 \times 10^{-5}$ s$^{-1}$ at low RH and moist conditions, respectively. In experiments under moist conditions, particles measured by SMPS consisted of liquid water content (LWC) and SOA. In low RH experiments, as SOA hardly absorbs aerosol water, LWC can be negligible. Thus, the SOA mass can be directly measured by SMPS in low RH experiments. To obtain the SOA mass in high RH experiments,

25    LWC should be excluded from total particle mass. The method for the measurement of LWC has been already described in the previous study (Jia and Xu, 2018), so here a brief introduction is only provided. During each high RH experiment, the SMPS measured the humid particles. After 4 h from the start of oxidation reaction in each high RH experiment, the SMPS was modified to the dry mode. In the dry mode, a Nafion dryer (Perma Pure MD-700-12F-3) was added to the sampling flow and a Nafion dryer (Perma Pure PD-200T-24MPS) was added to the sheath flow. After the modification of SMPS, the humid

30    air in SMPS was quickly replaced by dry air through venting the sheath air at 5 L min$^{-1}$, so that the RH in the sheath air can decrease to 7%. Then, SMPS at this dry mode measured dry particle concentrations as the RH in the sample air decreased to 10% at this time. The LWC was determined by the difference of the particle mass concentrations before and after the SMPS modification to the dry mode.

4. The use of the term "HOM" is confusing to me. It was used in section 3.4 2nd paragraph to describe products associated with OH abstraction and addition to m-xylene. I feel that compounds mentioned in the first paragraph of this section (3.4) should also be called "HOM". The authors should define exactly the term "HOM" in this paper.

Taking the reviewer's advice and the comment 15 of Referee 3, we have revised the content in Sec. 3.4. The HOM is referred to the highly oxygenated organic molecules that have been provided in the revised Sec. 3.4.

5. Deducing structures from HRMS mass spectra only can be misleading due to issues that can arise from LCMS instruments (e.g. solvents, unpredictable clusters formations etc..). The authors should be careful reporting these data (structures) without the benefit of reporting and interpreting MS/MS spectra associated with each compound tentatively identified in this study (see Table 3). The paper lack of providing and interpreting MS/MS spectra that could make the identification straightforward and robust. For example, MS/MS of reaction products (or selected products) in Scheme 1 identified by HRMS should be presented in the manuscript or in supplementary information.

Taking the referee's suggestion, we have added the $MS^2$ spectra of compounds mentioned in Table 3 in the supporting information, as can be seen in the reply of Comment 10 of referee 3. We agree with the referee that the structures in Scheme 1 should be carefully assigned. So we have deleted some assignments in the Scheme 1 in the revised manuscript.

6. In the abstract and in page 5 (lines 3-5), the authors report using the MCM mechanism as a tool to explain the RH effect and analyse products measured by HRMS. The following sentence is confusing to me "The reaction pathways and products of m-xylene-OH photooxidation in Master Chemical Mechanism (MCM v3.3.1, the website at http://mcm.leeds.ac.uk/MCM; last accessed October 16, 2017) was used for analysis of the products measured by HRMS (Jenkin et al., 2003; Jia and Xu, 2014)." Throughout the manuscript, the MCM mechanism was not used and these statements are not consistent with the data provided in this paper. Although, adding a section to the manuscript focusing on linking products observed in this work with the MCM mechanism will be beneficial and make the paper stronger.

In the manuscript, we listed a Table 3 to show the peaks that were measured by HRMS, whose structures can be proposed according to the gas-phase chemical mechanism of *m*-xylene-OH photooxidation included in MCM. Also, the content focusing on the linking products observed in HRMS with the MCM mechanism is already in the second paragraph of Sec. 3.3.

Other comments

1. Abstract. The sentence "The relative increase of C-O-C at high RH from the FTIR analysis of functional groups indicates that the oligomers from carbonyl compounds cannot well explain the suppression of SOA yield." Is not consistent with the data shown in Figure 2. The absorptions at 1180 cm-1 associated to C-O-C are higher under high RH than low RH?? Comments from the authors.

Yes, it is consistent with data in Fig. 2. In fact, the absorptions at 1180 cm$^{-1}$ associated to C-O-C are lower under high RH than low RH as shown in Fig. 2. However, the ratio of the SOA mass collected on disk at high RH to that at low RH is 0.29 while the ratio of peak intensities at high RH to that at low RH is 0.48, so we used the "relative increase" of C-O-C in the Abstract. It means under different RH conditions the proportion of oligomers is different. For clarification we have rephased this sentence in the Abstract.

The FTIR analysis shows that the proportion of oligomers with C-O-C groups from carbonyl compounds in SOA at high RH is higher than that at low RH, but further information cannot be provided by the FTIR results to well explain the negative RH effect on SOA formation.

2. The experimental section should be described clearly. I'm confused with the sentence describing the chamber "…Teflon FEP film reactor (…) in an indoor smog chamber". What the authors refers to indoor smog chamber in this contest? Are all reactants were introduced initially and then the light was turned on and the reaction starts. Please explain how the experiment was conducted? How H2O2 was evaporated to the reactor? Initial H2O2 concentrations should be provided in table 1. The experimental section should be described then readers can clearly understand how the experiment was conducted?

The indoor smog chamber we refer to is relative to the outdoor smog chamber, which indicates whole experiment system, including Teflon bag, light source, etc. In this study, all reactants were introduced initially and then the light was turned on and the reaction starts. Taking the referee's suggestion, we have added a sentence for clear description of the experiment and rewritten the sentences about the description of experimental procedure in the last paragraph of Sec. 2.1 in the revised manuscript. Also, we have added the $H_2O_2$ concentrations in Table 1 which can be seen in the reply of comment 1.

Hydrogen peroxide was introduced into the reactor along with the zero air flow over a period of 30 min via an injection of $H_2O_2$ solution (30 wt %) into a three-way tube using a syringe to the desired concentration of 20 ppm…All reactants were introduced initially and then the light was turned on and the reaction starts…

3. Page 4, line 13. How LWC was measured at high RH? This is very important since yield are derived from (total aerosol mass – LWC)? Why the method used for low RH was not used for high RH to deduct LWC?

Please see the reply of the major comment 3 above which is similar with this comment.

4. Is the density the same at high and low RH conditions?

Yes, the density for SOA is the same for high and low RH conditions, but it is different for particles. A value of 1.4 g cm$^{-3}$ for SOA is used in our work. Song et al. (2007) investigated the density of SOA formed from the oxidation of m-xylene under dried conditions in the absence of NOx and gave the density of 1.4 g cm$^{-3}$.

5. Page 4, lines 15-16. This sentence is not clear. Are the authors mean: "The chemical composition of SOA originated from m-xylene-OH irradiation was investigated using Fourier transform infrared spectrometer (FTIR)." instead of "For the analysis of functional groups of the chemical composition in SOA from m-xylene-OH irradiation, the SOA samples were collected and determined by FTIR (Fourier transform infrared spectrometer)."

Taking the referee's suggestion, we have rewritten the sentence in the revised manuscript.

The chemical composition of SOA originated from *m*-xylene-OH irradiation was investigated using Fourier transform infrared spectrometer (FTIR), which can provide the information of functional groups.

6. Page 4, line 17 and Table 1. What the end of the experiment refers too? Please state the exact time when samples were taken and for how long as well as the reaction time for data reported in Table 1. Are the disks were analysed directly after the sampling?

The end of the experiment refers to 4 h after the light was turned on. Sampling was made just after the end of experiments. The reaction time for data reported in Table 1 was 4 h. The disk was analysed directly after sampling. Taking the referee's advice, the above content has been added or rewritten in the manuscript for clear description, which can be seen in the reply of the major comment 2.

7. Table 1 should incorporate the LWC and reacted m-xylene and may be m-xylene if it was measure at several reaction time? Is all m-xylene was reacted at the time when samples were taken? The authors may show in Figure 1 the sampling time associated with each RH!

Taking the referee's suggestion, we have added the LWC and reacted m-xylene in Table 1, which can be seen in the reply of Comment 1. The m-xylene was measured at several reaction time points as shown in Fig. S1 in the supporting information. When the samples for particle analysis were taken, the m-xylene was not all reacted. As shown in Fig. S1, ~40% m-xylene

was reacted for each experiment. These two samples used for HRMS and for FTIR were taken just after 4 h of reaction, and the sampling durations of PILS for HRMS and DLPI for FTIR were 5 min and 15 min, respectively, which has been added in the Sec. 2.2 in the revised manuscript and can be seen in the reply of the major comment 2.

8. Are the SOA masses reported in Table 1 includes LWC?

The SOA masses reported in Table 1 do not include LWC.

9. Figure 1. Is the temperature being constant throughout the experiment? In general, the initial RH should decrease as the reaction progress and fluctuation of the temperature should be observed as the light is turned on. The RH provided in Table 1 is for the "end" of the experiment or the initial RH? I would prefer to see RH, temperature also included in Figure 1 as the reaction progresses. If the RH provided in figure 1 and Table 1 are at the "end" of the experiment (again is confusing to use the end of the experiment) then what was the initial RH and temperature at t=0 min? The term "end" is misleading and prefer to state the exact time.

Throughout the experiment a temperature probe was installed outside and close to the Teflon bag to monitor the temperature in real-time. At t = 0 and t = 4 h of an experiment a portable hygrometer with higher accuracy was used to obtain the temperature and RH inside the Teflon bag which is given in Table 1. Before lights were turned on, the temperature was controlled using air-conditioners to be 26°C measured by the temperature probe. After lights were turned on, we adjusted the setting temperature of the air-conditioners to maintain the temperature probe value to be 26°C. Indeed, there was a slight fluctuation of temperature at the first half hour. Then, the temperature was steady within a fluctuation of 0.5°C obtained by the temperature probe. As the temperature and RH were steady during most of the reaction time, we used the temperature and RH measured by the portable hygrometer to present, as shown in Table 1. The temperature and RH were not continuously recorded throughout the experiment, so we cannot provide the results with RH and temperature in Fig. 1 as the reaction progressed. Taking the referee's suggestion, we have changed the term "end" to the exact time in Table 1 and the context in the revised manuscript.

10. Does Figure 5 is necessary? I suggest reporting MS/MS spectra instead of this figure. I feel Figure 4 provides enough information?

Taking the referee's suggestion, we have deleted Fig. 5. The MS/MS spectra have been added as Fig. S3 in the supporting information based on the comment 10 from referee 3.

[revised manuscript text omitted]

---

## Author Response (AR3)

**Response to Editor**

We greatly appreciate the time and effort that Editor spent in reviewing our manuscript. The editor has pointed our several important issues in our manuscript. Based on the advices by the editor, we think we have greatly improved our manuscript. Below we make a point-by-point response to these comments. According to the requirement, the response to the editor is structured in the following sequence: (1) comments from the editor in black colour, (2) our response in blue colour, and (3) our changes in the revised manuscript in red colour.

My main issue with the manuscript is the very poor quality of SMPS measurements shown in Figures S1 and S2. Additionally, the large scatter in Figure 1 strongly suggests that something is wrong with your SMPS system - the data cannot be so noisy at these reasonably high mass concentrations of particles. I am hesitant to let a manuscript that focuses on SOA yields but relies on such noisy data to go through from ACPD to ACP.

What we understand the poor quality in Figs. S1 and S2 is the discontinuity of the particle size distribution. We found that the quality of SMPS results is related to the particle number concentration. As shown in Fig. R1, when the total number concentration decreases, the "quality" of particle size distribution goes "poorer". The uncorrected total number concentration of Exp.1 at 4 h is only $1.9 \times 10^3$ cm$^{-3}$. Thus, the quality in Figs. S1 and S2 looks poor but is reasonable.

[Figure]

Figure R1. Number concentration distributions of particles at about 4-h time point from recent m-xylene photooxidations conducted by our research group, in which the total number concentration is $4.6 \times 10^4$ cm$^{-3}$ (black) and $3.3 \times 10^3$ cm$^{-3}$ (red).

Based on the manual of CPC, if the number of particles counted in each time interval is very small, the uncertainty in the count is large. The uncertainty is controlled by the number concentration instead of the particle mass. In fact, the particle mass is sufficiently large but the number concentration is very low in our experimental system. In Fig. R2, the particle mass is similar to that in Exp. 2, but the number concentration is larger than $10^4$ cm$^{-3}$, so large scatter is absent.

[Figure]

Figure R2. Mass (black) and number (red) concentration time profiles of SOA from a recent m-xylene photooxidation system.

Recently, we found that SMPS results can be improved through increasing the scanning time of SMPS when the particle number concentration is small. The SMPS result of the four newly conducted photooxidations of m-xylene-$H_2O_2$ at low and high RH is shown in Figure 1 below, which is used to replace the previous Fig. 1. The quality of the new SMPS results has been greatly improved, which does not affect the original conclusion of RH effect on the SOA yield. This is because we increase the scanning time of SMPS. At the beginning of each experiment, we increased the scanning time of SMPS from 180s to 300s. Meanwhile, the scanning particle size range was set to be 17-1000 nm. As the reaction proceeded, the particle size increased and we decreased the scanning particle size range to 50-1000 nm. The scanning time of each size interval is enlarged to reduce the uncertainty caused by the low number concentration. Low and high RH experiments were newly conducted and reproduced to confirm the reason of poor quality of previous experiments. The large scatter in previous Fig. 1 is because we lack the SMPS measurement experience when the number concentration in the experimental system is low. As the scatters in the four newly-conducted experiments were quite small, we have deleted the corresponding discussion in the Sec. 3.1 in the revised manuscript. In addition, variations of particle size distribution of number concentrations at the 2-h time point and at the end of the experiment for the newly conducted low and high RH experiments have been shown in Fig. S1 in the revised supporting information.

[Figure]

Figure 1. Mass concentration time profiles of SOA from *m*-xylene-$H_2O_2$ photooxidation at low (Exps. 3-4) and high (Exps. 10-11) RH (corrected by particle wall loss and for the amount of LWC in particles).

[Figure]

Figure S1 Variations of particle size distribution of number concentrations at the 2-h time point and at the end of the experiment for the low RH experiment (Exp. 4) and the high RH experiment (Exp. 11).

The quality of the mass spectra shown in Figure 4 is also very poor, and generally below the threshold for the quality of
10 publishable high-resolution mass spectra. For example, in the negative ion mode spectra, the peaks above m/z 230 look like noise to me (because they all have a similar peak abundance), and only a few peaks even appear in the high RH data set. The positive ion more spectra are of slightly higher quality, but it is quite unexpected that high RH peaks would completely

disappear at lower m/z values. Finally, the peak abundance distribution shown in Figure 5a suggests that data are likely contaminated by a C10 impurity.

The quality of mass spectra looks poor because the peaks are selected of which intensities are larger than $10^5$ under the low RH condition and corresponding peaks under the high RH condition, followed by the subtraction of blank mass spectra, which is stated in the first paragraph of Sec. 3.3. This statement is not included in the title or annotation, which may confuse the editor and may make the editor think our mass spectra are of poor quality. Here we present the intact mass spectra in Fig. 4 in the revised manuscript. As shown in Fig. 4, in the negative ion mode, the peaks above m/z 230 do not have the similar peak abundance, and many peaks appear in the high RH data set. In the positive ion mode, the peaks at lower m/z values are shown in the intact mass spectra. The impurity peaks have been excluded as the mass spectra used in our study are background subtraction. The intensities of the C10 peaks in the sample are several times those of corresponding peaks in the blank. There are 15 compounds with nC=10 in Fig. 5a. It is impossible that all 15 compounds are impurities.

[Figure]

[Figure]

(b) Negative ion mode

Figure 4 Background-subtraction HESI-Q Exactive-Orbitrap MS results of SOA in both positive (a) and negative (b) ion modes from the photooxidation of m-xylene-OH under both low (Exp. 2) and high (Exp. 8) conditions (Note that the Y-axis scales for low and high RH are largely different, $10^6$ at low RH and $10^5$ at high RH).

My second issue is that I no longer think that the explanation of the suppressed particle yield in the Hinks et al. (2018) study, which you discuss extensively, was correct. After reading more papers on the subject of wall losses in chamber experiments, I am now convinced that the apparent yield suppression in Hinks et al. (2018) was simply an effect of SVOC escaping to the walls under humid conditions. Your discussion on page 7 focuses on compounds that are not supposed to partition in SOA particles (such as glyoxal and acetone). It is the SVOCs that we need to worry about under these circumstances. Please take a look at the following paper [Huang, Y. L.; Zhao, R.; Charan, S. M.; Kenseth, C. M.; Zhang, X.; Seinfeld, J. H., Unified Theory of Vapor-Wall Mass Transport in Teflon-Walled Environmental Chambers. Environ. Sci. Technol. 2018, 52, 2134-2142] and see if it can change your thinking and/or of you can improve on the discussion on page 7. The large difference between the seed vs no seed yields is a direct proof that you are losing most of condensable material on wall the surface. I wish this Huang's paper was out when we were writing up the Hinks et al. (2018) paper – we would have written a very different story.

The editor is right that the wall effect of these VOCs that can form SOA or directly partition in the particle phase under humid conditions greatly affects the RH effect of SOA yields. Nevertheless, we still consider that some mechanisms play an important

role in addition to SVOC wall effect. We have rewritten the discussion about the SVOC (or VOC) decay effect on SOA formation in the revised manuscript.

[revised manuscript text omitted]

Finally, the readability of the paper is still quite poor for the following reasons. a). Some of the paragraphs in the paper are very long and need to be split into smaller, logically coherent paragraphs. b). The flow of logic in your writing is not smooth, with frequent jumps from one idea to the next within the same paragraph or even sentence. c). There is a number of spelling and grammatical mistakes, some (but not all) of which are mentioned below.

I am afraid that I am prepared to reject the manuscript at this point based on my assessment above, and based on the recommendation of the reviewers, who rated the paper very poorly. I will select "Reconsider after major revisions" for now in order to not make this decision final, but I think a rejection is the correct course of action in this case. If you agree with me that that paper should not be published given the poor quality of your data, you do not have to do anything, and the submission will remain in the ACPD state. If you disagree with my decision, you are welcome to try to provide a rebuttal and a next revised version but I personally cannot see how you can achieve a publishable state with the data quality you currently have. It might be easier to start over at this point and submit a higher quality data set as a new paper somewhere else.

In case you elect to continue editing this manuscript, in addition to addressing my major concerns above, I would fix the following.

P2, L7: anthropogenic source -> anthropogenic sources

We have corrected it in the revised manuscript.

P2, L8: delete the 2nd instance of the "in the urban troposphere"

We have corrected it in the revised manuscript.

P2, L21: processed -> processes

We have corrected it in the revised manuscript.

P2, L26: condition that no -> condition, wherein no

We have corrected it in the revised manuscript.

P2, L34: this sentence does not make any to me, first it says it observed an RH effect but then is says it did not focus on it. Please rephrase.

We have rephrased this sentence in the revised manuscript.

In a study on chemical oxidative potential of SOA (Tuet et al., 2017) under low $NO_x$ conditions, it was observed that the mass concentration of SOA from *m*-xylene irradiation under the dry condition was much larger than that under the humid condition, whereas the study did not focus on the mechanism of the RH effect on *m*-xylene SOA formation.

P3, L5: I would split this sentence in two, one stating the limited RH range of the previous study and one discussing why GC may not be the optimal technique for analysis.

We have split this sentence in two in the revised manuscript.

Nevertheless, this study was only performed at a limited RH range of 15-25%. GC-MS in this study may not be the optimal technique for analysis of SOA components as high temperature at GC injection ports can easily decompose some low-volatile substances in SOA.

P3, L12: it is unclear to me what you mean by "Although the information of chemical compositions in SOA has been given". Compositions should be singular, composition.

We have rephased this sentence for clarification in the revised manuscript.

Although some chemical composition in SOA has been identified, the analysis and the mechanism of RH effects still need to be further studied.

P3, L21: Experiments of -> Experiments on

We have corrected it in the revised manuscript.

P3, L22: delete "So only a brief introduction is presented here" because the description that follows is not brief

We have deleted it in the revised manuscript.

P3, L25: "To remove the electric charge on the surface of the FEP reactor" - this is very interesting, do you have evidence that it helps? (Perhaps you can cite a previous study which shows that it helps.)

We have cited a reference in the revised manuscript.

McMurry, P. H., and Rader, D. J.: Aerosol wall losses in electrically charged chambers, Aerosol Sci. Technol., 4, 249-268, 10.1080/02786828508959054, 2007.

P4, L6: light -> lights

We have corrected it in the revised manuscript.

P4, L13: Was this determined by observing the total mass or total number of particles? Usually the particle wall constants are size dependent, so I would add statement that you approximate the wall loss as size independent.

The particle wall loss constant was determined by observing the total mass of particles. Taking the editor's advice, we have added statement that we approximate the wall loss as size independent in the revised manuscript.

Though the particle wall loss constant is size dependent, it is not a strong function of particle size for the relatively narrow size distributions in smog chamber experiments (Park et al., 2001). Here we approximate the wall loss as size independent.

Park, S. H., Kim, H. O., Han, Y. T., Kwon, S. B., and Lee, K. W.: Wall loss rate of polydispersed aerosols, Aerosol Sci. Technol., 35, 710-717, 10.1080/02786820152546752, 2001.

P4, L34: As the SOA compositions are almost all water-soluble species -> The low-NOx xylene SOA is water soluble (provide a reference or explain why you think so)

After the FTIR measurement of SOA samples collected on ZnSe windows, the ZnSe window was washed with ultrapure water and then measured by FTIR again, no absorbance was observed on FTIR spectra. Thus, it can be believed that the SOA components are almost all water-soluble species.

P5: I would insert a statement explaining that your sampling instruments consumed X% of reactor volume during the experiment, and that you have not used make up air to dilute the bag (since this was brought up by one of the reviewers)

We have added a statement in the first paragraph of the Sec. 2.2 in the revised manuscript.

In our enclosed Teflon bag, our sampling instruments consumed 10% of reactor volume during the photooxidation and we did not use make up air to dilute the bag.

P6, L4: Delete "Thus, the LWC can be obtained by VGF." to avoid repetition

We have deleted it in the revised manuscript.

P6, L4: As shown in Fig. 1 -> In Fig. 1. This sentence would fit better at the beginning of the 2nd paragraph – it does not fit well in the flow of logic of the 1st paragraph.

We have corrected it in the revised manuscript.

P6, L7: Good references to cite here are:

(1)     El-Sayed, M. M. H.; Amenumey, D.; Hennigan, C. J., Drying-Induced Evaporation of Secondary Organic Aerosol during Summer. Environ. Sci. Technol. 2016, 50, 3626-3633.

(2)     El-Sayed, M. M. H.; Wang, Y.; Hennigan, C. J., Direct atmospheric evidence for the irreversible formation of aqueous secondary organic aerosol. Geophys. Res. Lett. 2015, 42, 5577-5586.

We have added these references in the revised manuscript.

P6, L11: negligible -> neglected.

We have corrected it in the revised manuscript.

P7, L2: "the excess use of H2O2 can lead to an excess OH radicals" - this statement is incorrect. H2O2 is the primary reactant for OH under high H2O2 conditions, so OH steady state concentration is independent of H2O2 in this limit.

We have rephased this sentence in the revised manuscript.

In addition, the excess OH radicals in our experimental system lead to a less underestimation of SOA formation as the losses of SOA-forming vapours can be mitigated via the use of excess oxidant concentrations.

P7, 2nd paragraph: see comment above

Please see the response above.

P8: The paragraph on page 8 needs to be split.

We have split it in the revised manuscript.

P9, L2: Figure 2 -> Figure 3

We have corrected it in the revised manuscript.

P9, L1o: both two RH conditions -> both RH conditions.

We have corrected it in the revised manuscript.

P10, L3: deduction -> subtraction

We have corrected it in the revised manuscript.

P10, L3: blank-deducting -> blank-corrected

We have corrected it in the revised manuscript.

P10, L4: Figure3 -> Figure 4 (and fix numbering for the rest of the figures)

We have corrected it in the revised manuscript.

P10: use "peak abundance" instead of "intensity" when discussing heights of peaks in the mass spectrum

We have corrected it in the revised manuscript.

P10 and Table 3: what evidence do you have for the proposed epoxide structures? I am also quite concerned that the 2,6-dimethyl-1,4-benzoquinone compound listed in table 3 is too volatile, and should not be in the particle phase. Is it possible that you are looking at the results of fragmentation in the ion source?

The editor is right. The vapour pressure of 2,6-dimethyl-1,4-benzoquinone is $0.3 \pm 0.4$ torr at 25°C and its boiling point is 200.9 ± 15.0 °C (www.chemspider.com), which is volatile and should not be in the particle phase. From the identification of this compounds, the parent ion whose fragments are matched with those proposed by the Mass Frontier program may be proposed incorrectly, while our proposed structures in Table 3 are all based on the analytical results from this program, so we have decided to delete the discussion of Table 3 and the relevant Figures S3 and S4 in the supporting information.

P10, L29: Figure 4 -> Figure 5 (and adjust the rest of the figure numbers)

We have corrected it in the revised manuscript.

P11, L5: is proposed -> suggests

We have corrected it in the revised manuscript.

P11, L9: what method did you use to identify C2H2O as one of the most frequent mass differences?

We selected the peaks on the sample mass spectra with peak abundance of larger than $10^5$ in the low RH experiment and identified the formula of the selected mass. Then, we counted the frequency of the appearance of their mass difference.

P12, L7: differential -> different

We have corrected it in the revised manuscript.

P10, L13: "are much more oxygenated than those in the positive ion mode" - your signal-to-noise ratio is measuring O/C is not high enough to make these conclusions

The abundance is obtained by blank-subtraction. Though the signal to noise ratio is not high, the mass exists in the SOA sample.

P11-P12: the discussion of the mechanism of RH influence on the chemical composition is not supported by the presented data in my opinion

We have added some sentences in Sec. 3.4 in the revised manuscript.

SVOCs tend to escape to the wet reactor wall as we discussed in the Sec. 3.1, which interprets a certain proportion of SOA reduction at high RH. The wall process of the reactor enlarges the difference of SOA mass between low and high RH.

P20: the caption of Figure 1 should mention that SOA concentration was corrected for the amount of LWC in particles.

We have corrected it in the revised manuscript.

P23: Replace "Intensity" by "Abundance" and Nagetive by Negative in Figure 4

We have corrected it in the revised manuscript.

Supporting information

Figure S1 is not referred to in the text. Also, its caption says that it shows GC-MS data, which is obviously incorrect.

We have deleted it in the revised manuscript.

Figure S2: it is not clear how Figure S2 is different from S1. They seem to show slightly different data.

We have corrected it in the revised manuscript.

Figure S3: this figure is not at all discussed in the text. I understand the reviewers wanted to see MS/MS spectra, so those should be discussed properly.

As the response to the above comment mentioned, we have deleted the figure S3. Thus, it will not be discussed in the text.

[revised manuscript text omitted]

---

## Author Response (AR4)

**Response to Editor**

We greatly appreciate the time and effort that Editor spent in reviewing our manuscript. Below we make a point-by-point response to these comments. According to the requirement, the response to the editor is structured in the following sequence: (1) comments from the editor in black colour, (2) our response in blue colour, and (3) our changes in the revised manuscript in red colour.

Abstract: Please sentence stating that some of the reduction in the apparent yield may be due to the faster wall loss of semi-volatile products of oxidation at higher RH. (This reflects discussion on page 7)

Taking the editor's advice, we have added the sentence in the abstract in the revised manuscript.

Some of the reduction in the apparent yield may be due to the faster wall loss of semi-volatile products of oxidation at higher RH.

P6 L13: specify actual RH values (14% and 76%) instead of simply saying "low" and high". I would do this throughout the text, especially at the beginning of new paragraphs. For example, Exp. 1 (low RH) -> Exp. 1 (14% RH). This way the readers will not have to go to Table 1 all the time.

Taking the editor's advice, we have specified the actual RH values throughout the text in the revised manuscript.

P6 L18: nearly seven times larger -> nearly an order of magnitude larger

Taking the editor's advice, we have made the change in the revised manuscript.

P7: Please remove acetone from this discussion. Contrary to what you state on line 4, acetone has not been shown to form SOA under any reasonable conditions. Discussion of its wall loss is therefore irrelevant.

Taking the editor's advice, we have removed acetone from the discussion in the revised manuscript.

Figure 1: image or caption – specify the actual RH values instead of saying "low" and high". Sam goes for the rest of the figure captions.

Taking the editor's advice, we have specified the actual RH values in the images and captions in the revised manuscript.

Figures 1 and 2: I suggest starting the Y=axis at zero. The offset used in the current versions of the figures makes small values appear larger than they are.

5   Taking the editor's advice, we have corrected the Y axis at zero in the revised manuscript.

All figures: the resolution of the images may need to be improved. The figures appear grainy at the moment (because of a grey background used in the figures?)

10   Taking the editor's advice, we have improved the resolution of the images and the removed the grainy of the figures in the revised manuscript.

[revised manuscript text omitted]